# Efficacy, safety and single-cell analysis of neoadjuvant immunochemotherapy in locally advanced oral squamous cell carcinoma: a phase II trial

Zhongzheng Xiang[1,5], Xiaoyuan Wei [1,5], Zhuoyuan Zhang [2,5], Yueyang Tang [3,5], Linyan Chen [4,5], Chenfeng Tan [1], Yuanyuan Zeng[1], Jun Wang[1], Guile Zhao [2], Zelei Dai[1], Mingmin He[1], Ningyue Xu[1], Chunjie Li [2] ✉, Yi Li [2] ✉ & Lei Liu [1] ✉

The clinical activity of neoadjuvant immunochemotherapy (NAIC) for treating locally advanced oral squamous cell carcinoma (LA-OSCC) remains uncertain. This single-arm, phase II trial (ChiCTR2200066119) tested 2 cycles of NAIC with camrelizumab plus nab-paclitaxel and cisplatin in LA-OSCC patients. For primary endpoint, the major pathological response (MPR) rate was 69.0% (95% confidence interval (CI): 49.2%-84.7%). The treatment was well-tolerated, with only 2 patients (6.45%) having grade 3 or 4 treatment-related adverse events during neoadjuvant treatment. For secondary endpoints, the pathological complete response rate was 41.4% (95%CI: 23.5%-61.1%) and the objective response rate was 82.8% (24/29, 95%CI: 64.2%-94.2%). The 18-month overall survival and disease-free survival probabilities were 96.77% (95%CI: 79.23%-99.54%) and 85.71% (95%CI: 53.95%-96.22%), respectively. Exploratory analysis showed that patients with MPR exhibited higher density of baseline CD4_Tfh_CXCL13 cells, and increased density of tertiary lymphoid structures after NAIC. Baseline CD4_Tfh_CXCL13 cells might be potential predictive biomarker of efficacy. The interaction between CXCL13 on CD4_Tfh_CXCL13 cells and CXCR5 on B cells may play a role in treatment response. These findings suggest the potential of NAIC as a promising treatment for LA-OSCC and offer preliminary insights into responsive biomarkers.

Oral cavity squamous cell carcinoma (OSCC) is the most common and challenging subtype of non-nasopharyngeal head and neck squamous cell carcinoma (HNSCC)[1]. Approximately 60% of OSCC patients are diagnosed at locally advanced stage[2]. The current standard treatments for resectable locally advanced OSCC (LA-OSCC) include radical surgery resection followed by radiotherapy or cisplatin-based concurrent chemoradiotherapy[3]. Despite this, the 5-year overall survival rate remains below 50%[4]. Moreover, phase III trials to date have failed to demonstrate

[1]Department of Head and Neck Oncology, Cancer Center & State Key Laboratory of Biotherapy, West China Hospital, Sichuan University, Chengdu, China. [2]Department of Head and Neck Oncology & State Key Laboratory of Oral Diseases & National Clinical Research Center for Oral Diseases, West China Hospital of Stomatology, Sichuan University, Chengdu, China. [3]Department of Oral Pathology & State Key Laboratory of Oral Diseases & National Clinical Research Center for Oral Diseases, West China Hospital of Stomatology, Sichuan University, Chengdu, China. [4]Department of Biotherapy, Cancer Center & State Key Laboratory of Biotherapy, West China Hospital, Sichuan University, Chengdu, China. [5]These authors contributed equally: Zhongzheng Xiang, Xiaoyuan Wei, Zhuoyuan Zhang, Yueyang Tang, Linyan Chen. ✉e-mail: lichunjie@scu.edu.cn; liyi1012@163.com; liuleihx@gmail.com

survival benefits from neoadjuvant chemotherapy in LA-OSCC[5,6]. These underscore the urgent need for new treatment strategies.

The KEYNOTE 048 trial has established immune checkpoint inhibitors (ICIs), alone or combined with chemotherapy, as first-line treatments for unresectable recurrent/metastatic HNSCC[7]. This has prompted further investigation into such strategies for resectable HNSCC, including LA-OSCC. The ongoing KEYNOTE 689 trial is evaluating 2 cycles of neoadjuvant mono-ICIs in locally advanced HNSCC, with final results pending[8]. However, in LA-OSCC, neoadjuvant mono-ICI therapy showed a major pathological response (MPR) rate of only 7.1%[9].

Recent phase I/II trials have shown potential survival benefits from ICIs-based combination therapies in locally advanced HNSCC (including LA-OSCC), such as dual ICIs, ICIs plus targeted therapy with or without chemotherapy, ICIs plus radiotherapy with or without chemotherapy, and ICIs plus chemotherapy[10–27]. Among these, neoadjuvant immunochemotherapy (NAIC) has shown excellent pathological responses. Preclinical studies suggest that adding chemotherapy to ICIs could enhance the antitumor response[28,29]. NAIC has already been established as a standard treatment for resectable non-small cell lung cancer[30]. However, there are few clinical trials investigating NAIC specifically in LA-OSCC patients.

To address this gap, we conducted a single-arm, phase II trial to assess the efficacy and safety of NAIC with 2 cycles of camrelizumab plus nab-paclitaxel and cisplatin in patients with resectable LA-OSCC. We also explored potential biomarkers and underlying mechanisms for NAIC response in LA-OSCC.

## Results

### Patients and treatment

Between February 1, 2023 and May 31, 2024, 35 patients were screened and 33 patients were enrolled. Two patients withdrew consent before starting NAIC. Thus, a total of 31 patients received NAIC. The mean age was 54 ± 11 years, and 27 patients (87.1%) were male. Most patients (58.1%, 18/31) had clinical stage IVA disease. Baseline characteristics are shown in Table 1.

All 31 patients completed 2 cycles of NAIC. Among them, 2 patients refused surgical resection and imaging examination due to personal reasons. Consequently, 29 patients (93.5%) underwent radical surgery. All these 29 patients subsequently received postoperative radiotherapy or chemoradiotherapy and immunotherapy as maintenance therapy. The treatment procedure, study flowchart, and follow-up status of each patient are shown in Fig. 1a–c.

### Efficacy

A total of 29 patients who underwent surgery were evaluable for pathological response and radiographic response. The MPR rate was 69.0% (20/29, 95%CI: 49.2–84.7%), and the pathological complete response (pCR) rate was 41.4% (12/29, 95%CI: 23.5–61.1%) (Fig. 2a). In addition, all 3 patients with baseline stage IVB disease achieved MPR in both the primary tumor and cervical lymph nodes. Among the 26 patients with baseline stage III/IVA disease, 17 achieved MPR in the primary tumor (including 11 pCR), and 14 achieved MPR in the cervical lymph nodes (including 8 pCR) (Supplementary Table 1). All MPR patients had pathological downstage for the primary tumor after NAIC (Supplementary Table 1). The ORR per RECIST v1.1 was 82.8% (24/29, 95%CI: 64.2–94.2%). After NAIC, 7 patients achieved complete response, 17 had partial response, and 5 had stable disease, with no patients experiencing progressive disease (Fig. 2b). Representative radiographic response and pathological response images before and after NAIC are shown in Fig. 2c. A significant correlation between radiographic response and the percent of residual viable tumor (%RVT) was observed (Spearman's $r = -0.7318$, $p < 0.0001$) (Fig. 2d).

As of the data cutoff on November 30, 2024, the median follow-up duration for 31 treated patients was 18.6 months (range: 12.6–22.2). For all 31 treated patients, the 18-month OS probability was 96.77% (95%CI:

**Table 1 | . Baseline characteristics**

| Characteristics | All patients $n = 31$ |
|---|---|
| Age (years) | |
| Mean ± SD (range) | 54 ± 11 (30–75) |
| < 65 | 26 (83.87) |
| ≥ 65 | 5 (16.13) |
| Gender | |
| Male | 27 (87.10) |
| Female | 4 (12.90) |
| ECOG PS | |
| 0 | 31 (100.00) |
| 1 | 0 (0) |
| Alcohol use | |
| Yes | 17 (54.84) |
| No | 14 (45.16) |
| Smoking | |
| Yes | 18 (58.06) |
| No | 13 (41.94) |
| Primary tumor site | |
| Oral tongue | 15 (48.39) |
| Floor of mouth | 10 (32.26) |
| Buccal mucosa | 4 (12.90) |
| Retromolar trigone | 1 (3.23) |
| Gingiva | 1 (3.23) |
| Pre-neoadjuvant clinical T stage | |
| T1 | 0 (0) |
| T2 | 1 (3.23) |
| T3 | 18 (58.06) |
| T4 | 12 (38.71) |
| Pre-neoadjuvant clinical N stage | |
| N0 | 4 (12.90) |
| N1 | 7 (22.58) |
| N2a | 9 (29.03) |
| N2b | 3 (9.7) |
| N2c | 5 (16.12) |
| N3a | 1 (3.22) |
| N3b | 2 (6.45) |
| Pre-neoadjuvant clinical M stage | |
| M0 | 31 (100) |
| M1 | 0 (0) |
| Pre-neoadjuvant clinical TNM stage | |
| III | 8 (25.81) |
| IVA | 18 (58.06) |
| IVB | 5 (16.13) |
| Combined positive score (CPS) | |
| CPS < 1 | 6 (19.35) |
| 1 ≤ CPS < 10 | 10 (32.26) |
| 10 ≤ CPS < 20 | 4 (12.90) |
| CPS ≥ 20 | 11 (35.5) |

Data are mean ± standard deviation (range) or n (%).

79.23–99.54%) (Supplementary Fig. 1). For the two patients who refused surgery after NAIC, one (cT4bN2aM0) did not receive any subsequent treatment and died 12.6 months after the initial study treatment. The other patient (cT4bN1M0) underwent MRI reexamination at a local hospital after NAIC, suggesting a complete response. The patient then received radiotherapy at the local hospital but suffered neck lymph node metastasis 14 months after the initial

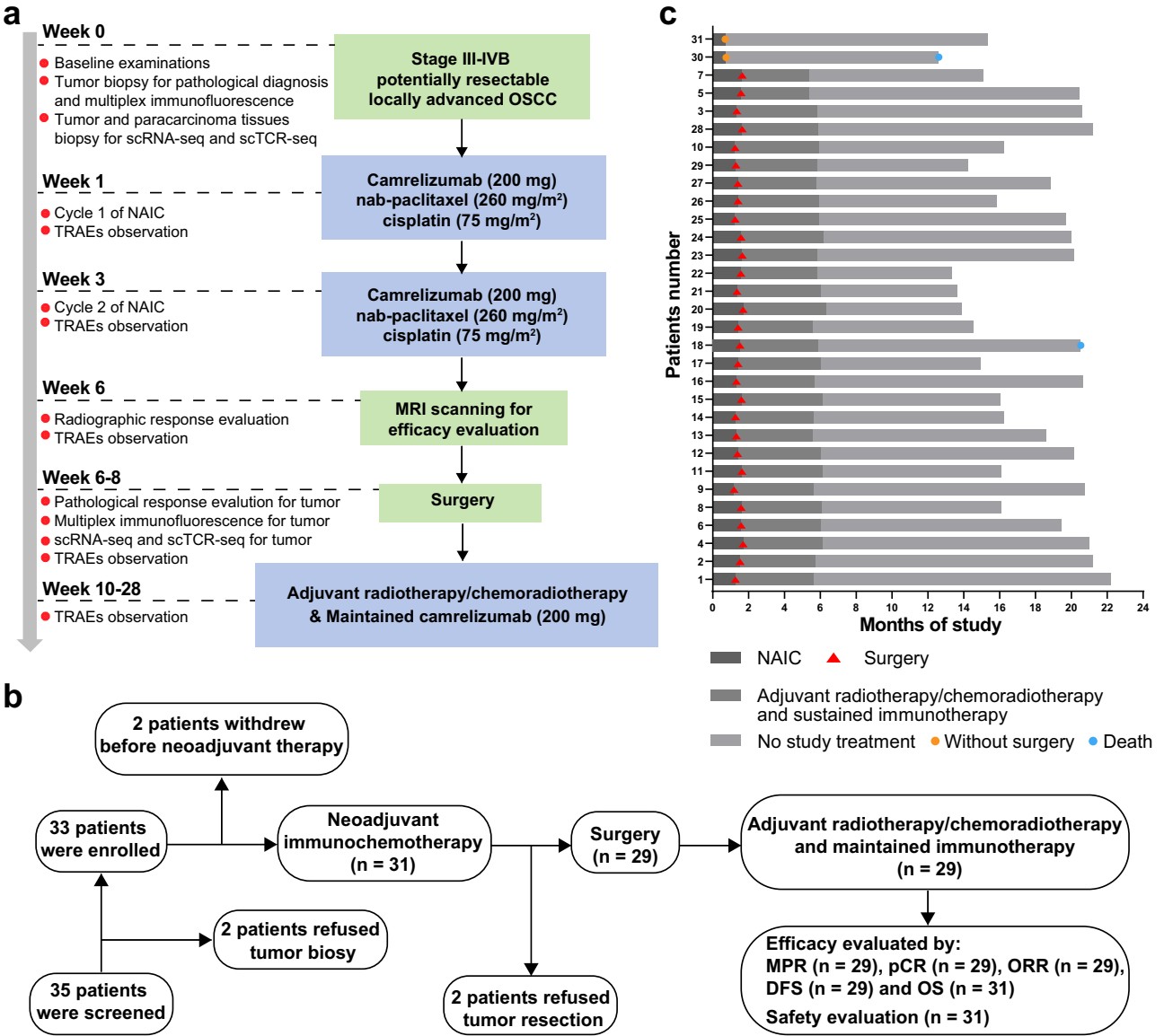

**Fig. 1 | Study design. a** Trial schema. Eligible patients received 2 cycles of NAIC with camrelizumab plus nab-paclitaxel and cisplatin (every 3 weeks). Surgery was performed within 2–4 weeks after NAIC. Radiotherapy was performed within 6 weeks after surgery. Maintenance immunotherapy with 6 cycles of camrelizumab (every 3 weeks) was started simultaneously with radiotherapy. **b** Patient flowchart. **c** Treatment and follow-up status for each patient. OSCC, oral squamous cell carcinoma; NAIC, neoadjuvant immunochemotherapy; MPR, major pathologic response; NMPR, non-major pathologic response. Source data are presented as a Source Data file. Note, P30 and P31 refused surgical resection and imaging examination for personal reasons after receiving 2 cycles of NAIC, thus the pathological response (%RVT) and radiographic response of these two patients were unavailable. Created in Adobe Illustrator 2024.

study treatment. For 29 patients who underwent surgery, the 18-month disease-free survival probability was 85.71% (95%CI: 53.95–96.22%) (Supplementary Fig. 2).

## Safety

During the NAIC phase, all 31 patients (100%) had treatment-related adverse events (TRAEs) of any grade (Supplementary Table 2). Grade 3 or 4 TRAEs occurred in 6.5% (2/31) of patients (grade 4 neutropenia and grade 3 thrombocytopenia), leading to dose reduction of nab-paclitaxel and cisplatin in the second cycle of NAIC. No grade 5 TRAEs were reported. The most frequent immune-related adverse event (irAE) was grade 1 reactive cutaneous capillary endothelial proliferation (5 [16.1%]). No grade ≥ 2 irAE occurred.

The median time from the completion of NAIC to surgery was 21.2 days (range: 14–28). No delays in surgery occurred due to NAIC-related toxicity. Of the 29 patients who underwent surgery, 13 (44.8%) experienced surgical complications, all of which were mild (grade 1 or 2) (Supplementary Table 3). The most common surgical complications were grade 1 pain (4 [13.8%]). All surgical complications resolved within 2-4 weeks with symptomatic treatment. During the adjuvant and maintenance treatment phase, no grade ≥ 3 TRAEs or unexpected safety signals were reported (Supplementary Table 4).

## Association between programmed cell death-ligand 1 (PD-L1) expression and pathological response

Of the 29 evaluable patients, 79.3% (n = 23) were PD-L1 positive (combined positive score (CPS) ≥ 1), and 20.7% (n = 6) were PD-L1 negative (CPS < 1). Patients with PD-L1 positivity had a higher MPR rate (82.6%, 19/23) than those with PD-L1 negativity (16.7%, 1/6). All (100%) patients with pCR (the percent residual viable tumor (%RVT) = 0) were PD-L1 positive (Fig. 2a). We found no significant correlation between PD-L1 expression (CPS) and %RVT (Supplementary Fig. 3a).

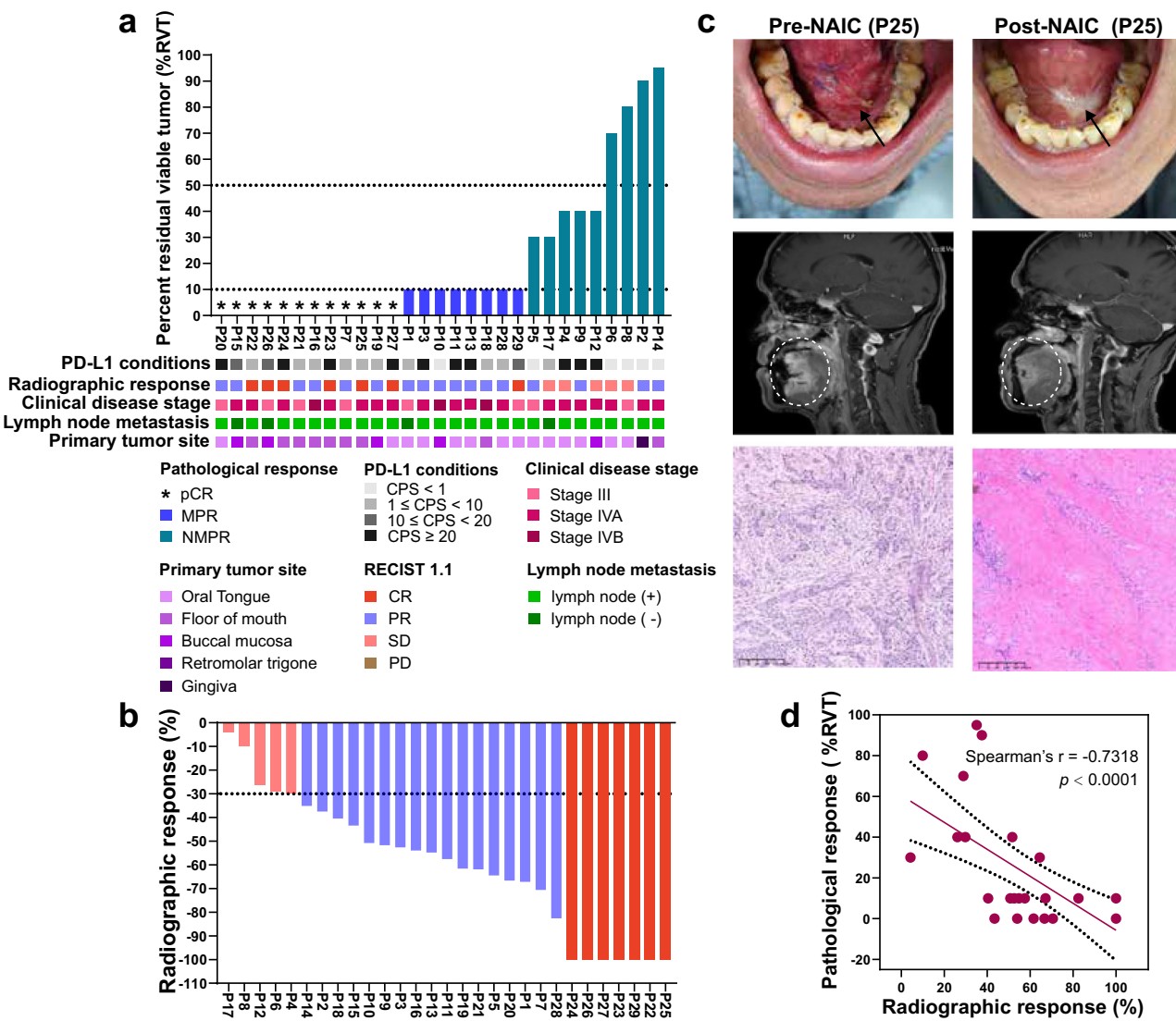

**Fig. 2 | Clinical responses to NAIC. a** Clinical characteristics and pathologic response. **b** Radiographic response per RECIST v1.1 criteria. **c** Representative image of patients who achieved MPR pre- and post-NAIC (*n* = 20), and this image was from patient P25. **d** Spearman correlation analysis between radiographic response and %RVT (Spearman *r* = −0.7318, *p* < 0.0001). The dotted lines represent a 95% confidence interval. The dot represents an individual data point. A two-sided test was used for statistical analysis. NAIC, neoadjuvant immunochemotherapy; MPR, major pathologic response; NMPR, non-major pathologic response; CPS, combined positive score; CR, complete response; PR, partial response; SD, stable disease; PD, progressive disease; %RVT, percent residual viable tumor. Source data are presented as a Source Data file.

## Pathological evaluation of tumor microenvironment changes after NAIC

H&E staining revealed a notable increase in tertiary lymphoid structures (TLS)-like features in MPR patients after NAIC (Fig. 3a). To further evaluate TLS and immune cell changes upon treatment, we performed 7-color multiplex immunofluorescence (mIF) on tumor tissue samples before and after NAIC. Post-NAIC, TLS density increased significantly only in MPR patients (*p* < 0.0001) (Fig. 3b−d) and showed a negative correlation with %RVT (Supplementary Fig. 3b). In addition, the enrichment of CD8+ T cells, CD20+ B cells, CD56+ natural killer cells and CD11c+ dendritic cells, and decreased infiltration of CD68+CD163+ tumor-associated macrophages, were only observed in MPR patients after NAIC (Fig. 3e−j). The infiltration levels of CD8+ T cells and CD20+ B cells were negatively correlated with %RVT (Supplementary Fig. 3c, d). A summary of immune cell changes during NAIC is shown in Fig. 3k.

## Genomic features

Whole exome sequencing (WES) analysis revealed that the most frequently mutated gene was *TP53* (72.4%), followed by *KMT2C* (34.5%)

and *FAT1* (31.0%) (Supplementary Fig. 4a). Genomic mutation profiles showed no significant differences between MPR and NMPR patients. We also found no significant differences in %RVT between *TP53* (+) and *TP53* (-) patients (Supplementary Fig. 4b). However, MPR patients had higher TMB levels than NMPR patients (Supplementary Fig. 4c).

## Tumor microenvironment features by single-cell RNA sequencing (scRNA-seq)

To evaluate the cellular composition and functional phenotypes of the tumor microenvironment (TME), scRNA-seq was performed on tumor and paracarcinoma tissue before and after NAIC (Fig. 4a). A total of 321,610 single cells met the quality control criteria. Using UMAP and canonical lineage markers, 7 primary clusters were identified (Fig. 4b). In both MPR and NMPR patients, the proportions of T cells, mono-cytes/macrophages, and mast cells were higher in tumor than in paracarcinoma tissue (Fig. 4c). In pre-NAIC tumor tissue, MPR patients exhibited higher proportions of T cells, B cells, and monocytes/mac-rophages, whereas NMPR patients showed higher proportions of epithelial/malignant cells, mast cells, and fibroblasts (Fig. 4d, e).

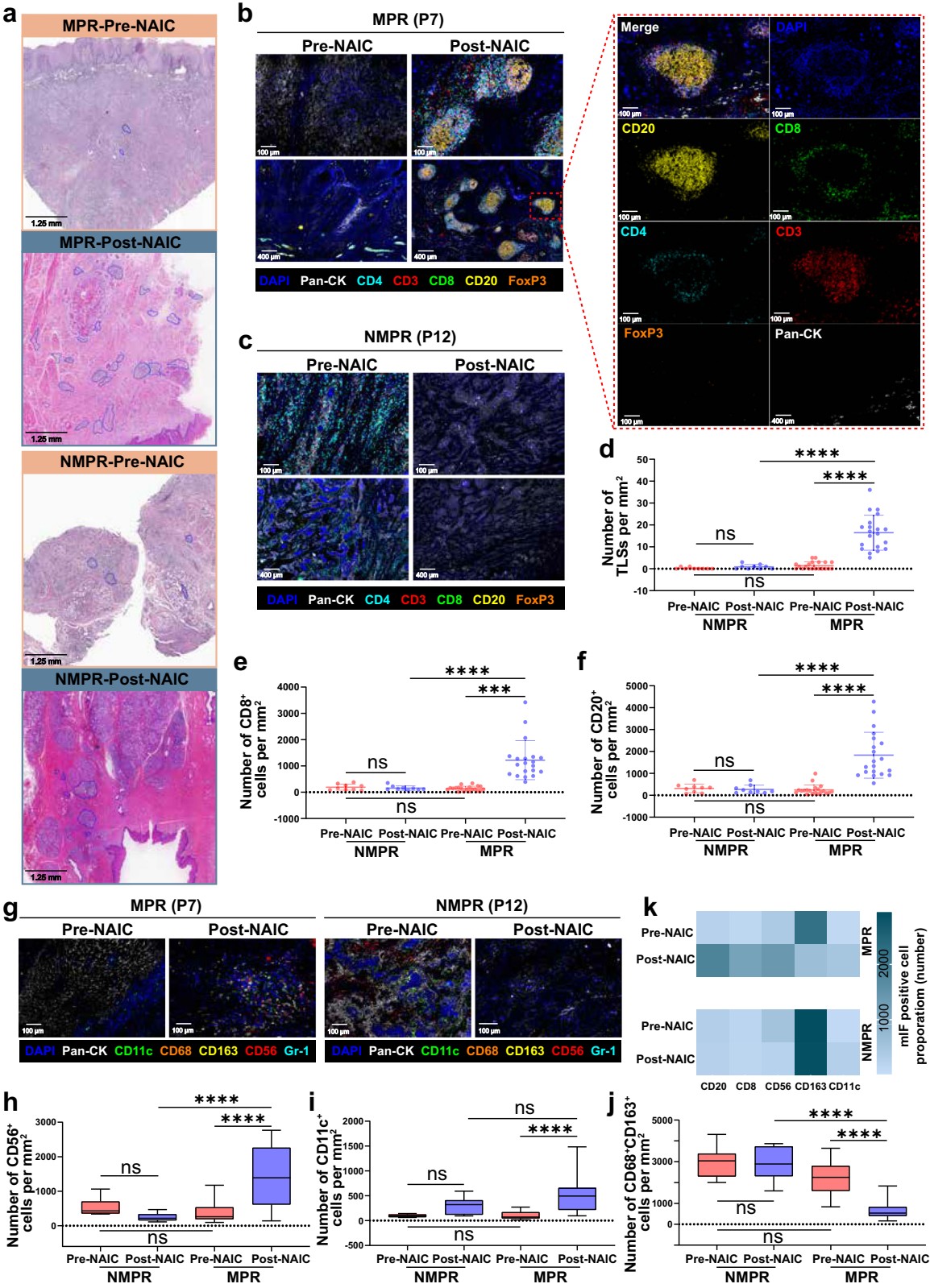

Post-NAIC, MPR patients (P18, P20, and P24) exhibited decreased proportions of epithelial/malignant cells and increased proportions of T cells, B cells, and fibroblasts (Fig. 4f).

## CD4_Tfh_CXCL13 cells correlated with clinical efficacy of NAIC

Using a graph-based clustering algorithm and classic marker genes, we identified a distinct CD4+ T follicular helper cell subset

(CD4_Tfh_CXCL13) cells (Fig. 5a and Supplementary Fig. 5a). Pre-NAIC, the effector T cell (CD4_Tact, CD4_CTL, CD8_Teff, CD8_Tact and CD4_Tfh_CXCL13, etc) were more prevalent in tumor tissue than in paracarcinoma tissue (Fig. 5b and Supplementary Fig. 5b). In pre-NAIC tumor tissue, the proportions of CD4_Tfh_CXCL13 cells in MPR patients were higher than that in NMPR patients (Fig. 5c, d and Supplementary Fig. 5c, d). The expression level of *CXCL13* was prominently higher in

**Fig. 3 | Tertiary lymphoid structures and TME features in MPR and NMPR patients before and after NAIC. a** H&E staining, TLSs were marked by blue circles. **b, c** Multiplex immunofluorescence (mIF) staining for TLSs and TLS-related markers. **d–f** Quantification of TLSs (**d**), CD8⁺ T cells (**e**), and CD20⁺ B cells (**f**) based on mIF staining analysis in MPR (n = 20) and NMPR (n = 9) patients before and after NAIC in the tumor. Data are presented as mean values with standard deviation (SD), the dot represents an individual data point. Ordinary one-way ANOVA with Tukey's multiple comparisons test was used for statistical analysis. **g** Representative mIF images showing dendritic cells (CD11c⁺), natural killer cells (CD56⁺), and M2-type tumor-associated macrophages (CD68⁺CD163⁺). **h–j** Quantification of CD56⁺ (**h**),

CD11c⁺ (**i**), and CD68⁺CD163⁺ cells (**j**) based on mIF staining analysis in MPR (n = 20) and NMPR (n = 9) patients before and after NAIC. Box plots show the distribution, the center line represents the median, the box indicates the upper and lower quartiles, and the whiskers represent minima and maxima. Ordinary one-way ANOVA with Tukey's multiple comparisons test was used for statistical analysis. **k** Heatmaps illustrating the proportions of the above immune cells. TLSs, tertiary lymphoid structures; NAIC, neoadjuvant immunochemotherapy; MPR, major pathologic response; NMPR, non-major pathologic response. ns, not significant; ***P < 0.001, ****P < 0.0001. Source data and p-values are presented as a Source Data file.

MPR_Pre-NAIC_Tumor group, especially in CD4_Tfh_CXCL13 cells subtype (Fig. 5e). Pre- and post-NAIC, paired MPR samples (P18, P20 and P24) exhibited increased proportions of CD4_Tfh_CXCL13, CD8_Teff and CD8_Trm_ex cells (Fig. 5f).

We further performed mIF on tumor tissue samples before and after NAIC to detect the dynamic changes of CD4⁺CXCL13⁺ cells upon treatment (Fig. 5g, h). The density of CD4⁺CXCL13⁺ cells was higher in MPR than in NMPR patients, both before and after NAIC. Notably, only MPR patients showed increased density of CD4⁺CXCL13⁺ cells after NAIC. A negative correlation between the density of tumor-infiltrated CD4⁺CXCL13⁺ cells pre-NAIC and %RVT (spearman's $r = -0.6412$, $p = 0.0002$) was observed, and this correlation was more pronounced in post-NAIC (spearman's $r = -0.7691$, $p < 0.0001$) (Supplementary Fig. 5e, f). The pre-NAIC density of CD4⁺CXCL13⁺ cells showed high predictive accuracy for %RVT (AUC = 0.9251, Fig. 5i). Functional enrichment analysis was performed to evaluate the gene expression patterns of all groups. In the MPR_Pre-NAIC_Tumor and MPR_Post-NAIC_Tumor groups, CD4⁺ T cells were enriched with the CXCL13 signaling pathway and T cell-induced cell death process, whereas the Tregs differentiation process was weakly enriched in the MPR_Post-NAIC_Tumor. In the MPR_Pre-NAIC_Tumor group, CD8⁺ T cells were enriched with antigen presentation of the HLA class I pathway. Additionally, T cell-mediated cytotoxicity and T cell-induced cell death processes were enriched in the MPR_Post-NAIC_Tumor group (Fig. 5j). In addition, TCGA data showed that head and neck carcinoma patients with high CXCL13 expression had improved overall survival and progression-free survival than those with low CXCL13 expression (Supplementary Fig. 6a, b). In these context, CD4⁺CXCL13⁺ cells at baseline played an important role for NAIC treatment in LA-OSCC.

**Characterization of T cell receptor (TCR) repertoire after NAIC**
The antitumor immune response is largely dependent on the proliferative expansion of T cells after the binding of TCR to the HLA-antigen complex[31]. Using single-cell TCR sequencing, we assessed NAIC-related clonal expansion of T cells and identified significant hyperexpansion in effector CD4⁺ and CD8⁺ T cells (CD4_Tfh_CXCL13, CD4_CTL, CD8_Teff, CD8_Tm, CD8_Tact, etc) (Fig. 6a). There was substantial TCR clonotype sharing between CD8_Tact and CD8_Teff after NAIC (Fig. 6b). Higher clonal ratio was mainly observed in CD4_Tfh_CXCL13 and CD8_Teff cells for MPR_Pre-NAIC_Tumor samples (Fig. 6c–e). In contrast, in the NMPR_Pre-NAIC_Paracarcinoma samples, greater clonal expansion was observed in CD4_Tfh_CXCL13 and CD8_Teff cells. Although the expansion of CD4_Treg cells was observed, unexpanded CD4_Treg clonotype was also noted in MPR patients compared with NMPR patients Pre-NAIC. Furthermore, paired MPR samples (p18, p20, p24) showed novel TCR clone mainly in CD4_Tfh_CXCL13, CD8_Teff, and CD8_Trm_ex cells (Fig. 6f, g), and the TCR clonal expansion index increased in all cells after NAIC (Fig. 6h).

**Increased infiltration of germinal center B cells after NAIC**
The formation and function of TLS mainly depend on the interaction between T and B cells, which are the dominant cellular components of TLS[32,33]. We identified 9 B cell subclusters (Fig. 7a). Several B cell

subclusters were more abundant in tumor tissue than in paracarcinoma tissue, including B_naive, germinal center B cells in dark zone (GCB_DZ), and unswitch GCB in light zone (GCB_LZ_G2M_unswitch) (Fig. 7b). Stress-related genes, *HSPB1* and *BAG3*, were expressed at higher levels in both tumor and paracarcinoma tissues in the NMPR_Pre-NAIC group compared to the MPR_Pre-NAIC group (Fig. 7c). Pseudotime trajectory analysis of these subclusters revealed 9 cell states (S1–S9) and 2 primary differentiation trajectories. B_naive cells and B_naive_active cells dominated S3 in the early stage, whereas GBC_DZ cells dominated S7 in the latest stage, and other cells dispersed throughout all states (Fig. 7d, e). Differentially expressed genes (DEGs) between the MPR_Pre-NAIC and NMPR_Pre-NAIC groups, mainly enriched in B_naive_active, GCB_LZ_HSP, and GCB_LZ cells (Fig. 7f). Paired MPR samples (P18, P20, P24) exhibited decreased percentages of B_naive and GCB_LZ_HSP cells and increased percentages of other B cell subtypes after NAIC (Fig. 7g, h). GO analysis of B cells revealed several key immune processes: antigen presentation and immune memory processes were enriched in the MPR_Pre-NAIC_Tumor group; tumor-migration process was enriched in the NMPR_Pre-NAIC_Tumor group; immune response and activation of T cell-induced cell death-related processes were enriched in MPR_Post-NAIC_Tumor group (Fig. 7i).

**Association between CD4_Tfh_CXCL13 cells and TLS**
Cell-cell communication analysis revealed that CXCL13 (ligand) on CD4_Tfh _CXCL13 cells interacted with CXCR5 (receptor) on B_naïve and CXCR3 (receptor) on B_naïve_active cells (Fig. 8a). The activated naive B cells promoted reciprocal signaling with CD8 T cell subsets (CD8_Teff, CD8_Tm, CD8_Trm_ex) via MHC I pathway (Fig. 8b). These results were supported by mIF analysis, which showed that post-NAIC, MPR patients had an increased number of TLS with mature phenotypes (reticulate FDCs and germinal center) and widespread distribution of CXCL13 and CXCR5 positive cells (Fig. 8c and Supplementary Fig. 7). Moreover, in germinal center, CD4⁺CXCL13⁺ cells were obviously adjacent to CD20⁺CXCR5⁺ cells on multiplex immunofluorescence staining slides (Fig. 8d), further spatial analysis by G-cross function and Jaccard index analyses showed that CD4⁺CXCL13⁺ cells were most frequently co-localized with CD20⁺CXCR5⁺ cells in the TLS, whereas other CD4⁺CXCL13⁻ cells prsented with a poor co-localization to CD20⁺CXCR5⁺ cells (Fig. 8e–g). These data suggest that interaction between CD4_Tfh_CXCL13 cells and B_naïve cells might be a potential mechanism for the immune response during NAIC.

**Chemotherapy enhanced the anti-tumor immune response to ICIs in vivo**
To identify which drug in this immunochemotherapy regimen functioned as an immunostimulatory or immunosuppressive role, we performed in vivo experiments on SCC7-bearing mice. Mice were treated with saline, cisplatin, nab-paclitaxel, nab-paclitaxel + cisplatin, anti-PD1, or nab-paclitaxel + cisplatin+anti-PD1 when the tumor volume reached about 80 mm³ (Supplementary Fig. 8a). We found anti-PD1+chemotherapy significantly suppressed tumor growth and prolonged mice survival compared to other groups (Supplementary Fig. 8b–e). Meanwhile, flow cytometry analyses illustrated that the

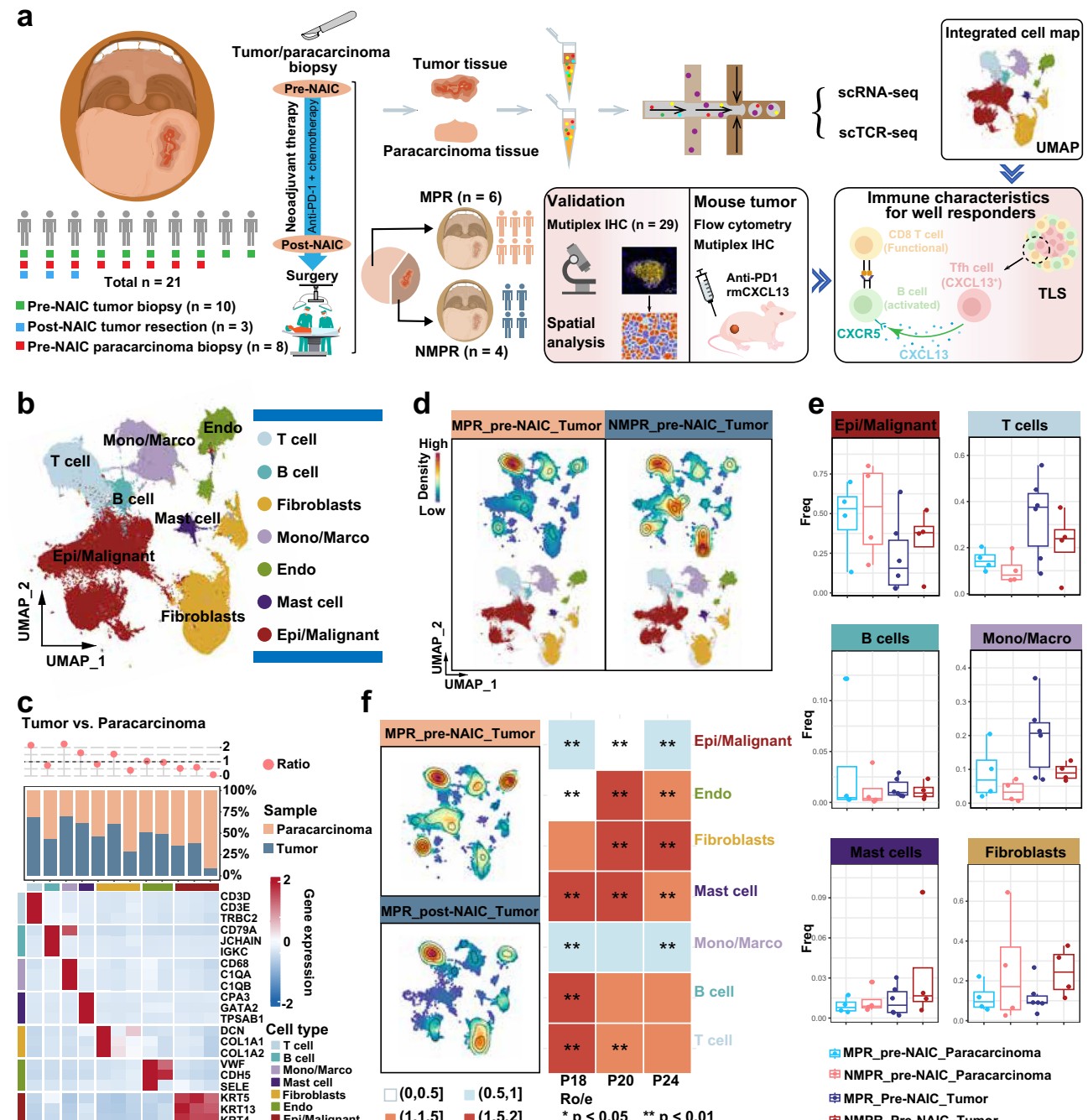

**Fig. 4 | Single-cell RNA-seq analyses of tumor and paracarcinoma tissues.**
**a** Study workflow (Created in Adobe Illustrator 2024). **b** UMAP illustrations of all cell clusters. **c** Representative marker genes for each cell cluster and cell distribution characteristics in paracarcinoma and tumor tissues. **d** UMAP illustrations of the density of cell subtypes in MPR_pre-NAIC_Tumor and NMPR_pre-NAIC_Tumor samples. **e** Proportions of cell clusters among different groups, MPR_pre-NAIC_Paracarcinoma ($n = 4$), NMPR_pre-NAIC_Paracarcinoma ($n = 4$), MPR_Pre-NAIC_Tumor ($n = 6$), NMPR_Pre-NAIC_Tumor ($n = 4$). Box plots show the

distribution (box and whiskers), the center line represents the median, the lower and upper box limit represents the interquartile range (IQR), the whiskers represent the 1.5 × IQR. **f** UMAP illustrations of changes in the density of all cell subtypes and cell numbers comparison in paired pre- and post-NAIC tumor tissues (P18, P20, P24). A two-sided Chi-square test was used for statistical analysis. MPR, major pathologic response; NMPR, non-major pathologic response; Ro/e, the ratio of observed over expected cell numbers. *p*-values are presented as a Source Data file.

percentages of CD4⁺CXCL13⁺, CD3⁻CD19⁺, and CD3⁺CD8⁺ cells were significantly increased in the anti-PD1 and nab-paclitaxel+cisplatin +anti-PD1 groups compared to other groups (Supplementary Fig. 8f–k). These results suggest that, among those agents, anti-PD1 showed a more favorable immune activation effect, and anti-PD1 combined with chemotherapy further enhanced the anti-tumor immunity.

## CXCL13 augmented the anti-tumor effect of immunochemotherapy in vivo

Considering the association between CXCL13 and pathological response, we explored the role of CXCL13 in the anti-tumor effect of immunochemotherapy in vivo. SCC7-bearing mice were divided into saline, rmCXCL13, immunochemotherapy, and rmCXCL13 plus immunochemotherapy (combination) groups (Fig. 9a). Compared with the

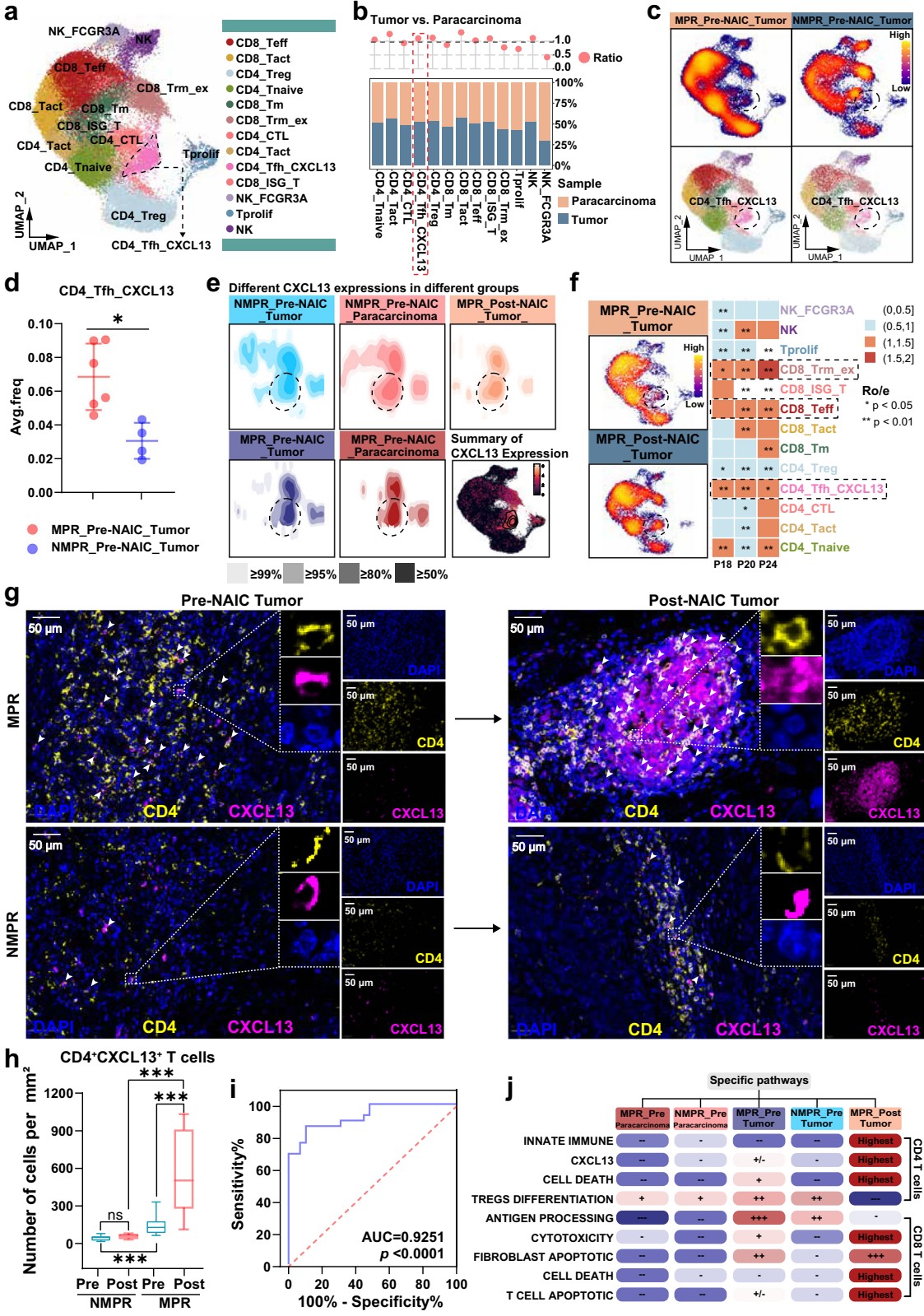

saline and rmCXCL13 groups, the combination group showed a greater inhibitory effect on tumor growth and longer survival duration. The inhibitory effect on tumor growth was more obvious in the combination group than in the immunochemotherapy group, and two of five tumors in the combination group completely regressed (Fig. 9b–e). Flow cytometry analyses revealed that the percentages of

CD4+CXCL13+, CD3−CD19+, and CD3+CD8+ cells were significantly increased in the rmCXCL13 group compared with the saline group, and ratios of these cells in the combination group were all higher than single rmCXCL13 and immunochemotherapy group (Fig. 9f–k). Meanwhile, mIF confirmed the increase percentages of CD4+, CD8+, and CD20+ cells in the combination group (Fig. 9l). These results

**Fig. 5 | Identification of CD4_Tfh_CXCL13 cells by single-cell RNA-seq and multiplex immunofluorescence analysis. a** Sub-clustering of T/NK cells, colored and labeled by subtypes. **b** Distributions of cell subtypes in paracarcinoma and tumor tissues. **c** UMAP illustrations of changes in the number of T/NK cell subtypes in MPR- and NMPR tumor tissues before NAIC. **d** Frequencies comparison of CD4_Tfh_CXCL13 cells in MPR_pre-NAIC_Tumor (*n* = 6) and NMPR_pre-NAIC_Tumor (*n* = 4) samples. Data are presented as mean values with SD, the dot represents an individual data point. The two-sided Mann-Whitney U test was used for statistical analysis. **e** Different CXCL13 expressions in different groups. **f** UMAP illustrations of changes in the number of T/NK cell subtypes and cell density comparison in paired pre- and post-NAIC tumor tissues (P18, P20, P24). A two-sided Chi-square test was used for statistical analysis. **g** Representative mIF staining images of CD4⁺CXCL13⁺ cells in pre- and post-NAIC tumor tissues in MPR and NMPR patients (*n* = 29). White

arrows represent CD4⁺CXCL13⁺ cells. **h** Comparison of CD4⁺CXCL13⁺ cells based on mIF staining analyses in NMPR_Pre-NAIC_Tumor (*n* = 9), NMPR_Post-NAIC_Tumor (*n* = 9), MPR_Pre-NAIC_Tumor (*n* = 20), MPR_Post-NAIC_Tumor (*n* = 20) samples. Box plots show the distribution, the center line represents the median, the box indicates the upper and lower quartiles, and the whiskers represent minima and maxima. The two-sided Mann-Whitney U test was used for statistical analysis. **i** Receiver operating characteristic curve analysis for baseline CD4⁺CXCL13⁺ cell density to predict the percentage of residual viable tumor (%RVT) after NAIC by Wilson/Brown method. **j** Specific pathways related to CD4⁺ and CD8⁺ T cells in different groups. MPR, major pathologic response; NMPR, non-major pathologic response; mIF, multiplex immunofluorescence; Ro/e, the ratio of observed over expected cell numbers. *\**P* < 0.05, \*\*\**P* < 0.001. Source data and *p*-values are presented as a Source Data file.

indicated the potential of CXCL13 in improving the anti-tumor effect of immunochemotherapy.

## Discussion

In this single-arm, phase II trial, NAIC with camrelizumab plus nab-paclitaxel and cisplatin demonstrated a favorable pathological response and manageable safety profile in patients with LA-OSCC. Through scRNA-seq, scTCR-seq, mIF analyses, and experimental validation, we identified baseline CD4_Tfh_CXCL13 cells as a potential predictive biomarker for NAIC response.

This study represents the largest-scale trial to date (*n* = 29) evaluating the efficacy of ICIs-based neoadjuvant treatments in LA-OSCC. Previous studies reported the following pathological responses in locally advanced HNSCC. For instance, the neoadjuvant mono/dual ICB (MPR rate: 5.9–35%; pCR rate: 0–10%), ICB combined with targeted therapy ± chemotherapy (MPR rate: 40–60%; pCR rate: 5–40%), ICB combined with radiotherapy ± chemotherapy (MPR rate: 43–60%; pCR rate: 20–57%; including p16 positive patients), and ICB combined with chemotherapy (MPR rate: 27.8–74.1%; pCR rate: 16.7–55.6%), etc.[9–27]. Compared with these results, the regimen of camrelizumab plus nab-paclitaxel and cisplatin in this study appeared to demonstrate a comparable pathological response rate, with an MPR rate of 69% and a pCR rate of 41.4%. Furthermore, 2 cycles of NAIC in this study achieved a MPR rate (69%) comparable to that of 3 cycles of the same regimen (63%) in locally advanced HNSCC[21]. The pCR rate in this study was relatively lower (41.4% vs. 55.6%), possibly due to the reduced number of NAIC cycles in this study and the inclusion of HPV-positive patients (18.8% [9/48]) in the comparative study, as HPV-positive HNSCC is known to be more sensitive to chemotherapy than HPV-negative HNSCC[34]. These results suggest that additional cycles of NAIC might further enhance the pCR rate of our regimen. Furthermore, the varied disease types between this study (29 OSCC patients) and the comparative study (16 OSCC and 32 non-OSCC HNSCC patients) require cautious interpretation of the comparisons due to potential biases.

In this study, the neoadjuvant regimen of camrelizumab plus nab-paclitaxel and cisplatin was well-tolerated, with no unexpected TRAEs occurred during NAIC. Only 2 patients experienced grade ≥3 TRAEs during NAIC, all of which resolved after symptomatic treatment. Notably, the incidence of reactive cutaneous capillary endothelial proliferation observed with the 2-cycle NAIC in this study was lower than that observed with 3-cycle NAIC using the same regimen (16.1% vs. 29.2%)[21]. In addition, the incidence of grade ≥3 TRAEs in this study was lower than that reported in the Illuminate trial (6.4% vs. 15%)[27]. These results highlight the need to further investigate the optimal number of NAIC cycles to achieve a balance between reduced toxicity and favorable efficacy in patients with LA-OSCC. In addition, surgical complications, which is a critical concern for oral and maxillofacial surgeons, were similar to those observed with conventional standard radical resection[5].

Identifying predictive biomarkers for ICIs-based neoadjuvant treatments in LA-OSCC remains challenging. While PD-L1 expression is

recommended as a treatment indicator for immunotherapy in recurrent and metastatic HNSCC[7,35], this study observed no significant correlation between PD-L1 expression and %RVT. This discrepancy may be due to the limited accuracy of PD-L1 evaluation in pre-NAIC biopsy samples. In addition, we found no significant differences in gene expression between MPR and NMPR patients, whereas higher TMB levels were associated with an improved pathological response to NAIC.

TLS are lymphoid organs that arise in non-lymphoid tissues under continuous antigen stimulation within chronic inflammatory microenvironments[36]. Recent studies have identified TLS as an intratumoral loci for initiating antitumor immune response[32,37]. TLS have been reported as a potential predictive biomarkers in several solid tumors[38], including esophageal cancer[39], clear cell renal cell cancer[40], and non-small cell lung cancer[41]. Our study revealed that there was no significantly statistical difference of the density of TLS between MPR and NMPR tissues before NAIC, and an obvious enrichment of TLS was only observed in the MPR tissue after NAIC. Thus, the accumulation of TLS after NAIC might be closely related with the tumor regression. Therefore, TLS might be the cause of the tumor regression, which needs to be further exploration. Moreover, we found that the density of CD4_Tfh_CXCL13 cells were higher in MPR than NMPR samples both before and after NAIC, indicating CD4_Tfh_CXCL13 cell could be potential predictive biomarker for NAIC in treating LA-OSCC. CD4_Tfh_CXCL13 cells have also been identified as predictive biomarkers in melanoma[42] and triple-negative breast cancer[43]. Meanwhile, higher CD4_Tfh_CXCL13 cell density in MPR than NMPR samples before NAIC, as confirmed by scRNA-seq and mIF analyses, suggesting that it might play a pivotal role in TLS formation and TLS-triggered antitumor response, consisting with previous studies[44,45]. This process may be mediated by the interaction between CXCL13 on CD4_Tfh cells and CXCR5 on B_naive cells, which was also observed in nasopharyngeal cancer[46]. Further, in vivo experiments on SCC7-bearing mice revealed that chemotherapy alone could not induce robust anti-tumor immunity in vivo, but could significantly enhance the anti-tumor immunity triggered by ICIs. These findings are consistent with prior research conducted by Yuanyuan Zhang et al.[43]. Moreover, CXCL13 appears to enhance the anti-tumor effects of immunochemotherapy by fostering a favorable tumor immune microenvironment. In the treatment of esophageal squamous cell carcinoma, CXCL13 was also identified that could potentiate the efficacy of anti-PD1 in vivo[47]. Taken together, these results suggest that CD4_Tfh_CXCL13 might be a potential favorable factor for tumor regression during NAIC, which needs further validation.

This study has several limitations. The single-center design limits the generalization of the findings. The relatively small sample size and single-arm design underscore the exploratory nature of this study, making it insufficient to draw definitive conclusions. Although the NAIC regimen showed encouraging pathological responses, long-term survival outcomes require further follow-up. While MPR and pCR are regarded as surrogate endpoints for survival outcomes in lung cancer

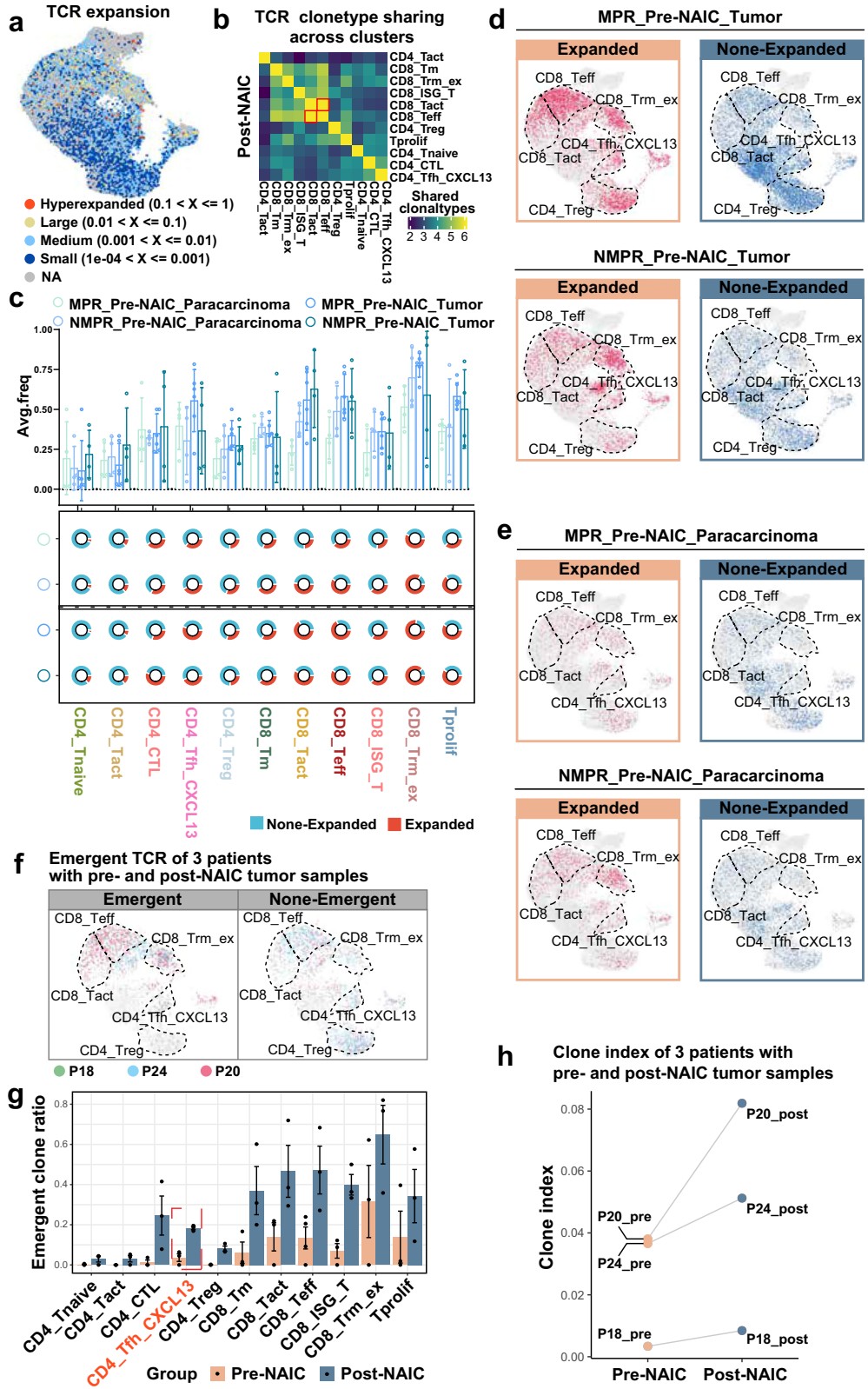

**Fig. 6 | Characterization of TCR repertoire before and after NAIC. a** Landscape of TCR expansion. **b** TCR clonotype sharing in post-NAIC tumor tissue. **c** The percentages of clonal T cells in Pre-NAIC_Paracarcinoma (MPR_Pre-NAIC_Paracarcinoma, *n* = 4; NMPR_Pre-NAIC_Paracarcinoma, *n* = 4) and Pre-NAIC_Tumor samples (MPR_Pre-NAIC_Tumor, *n* = 6; NMPR_Pre-NAIC_Tumor, *n* = 4), top is Column chart, bottom is Pie chart, data are presented as mean values with SD, the dot represents an individual data point. **d, e** Projection of TCRs onto the UMAP embeddings of T cells in MPR_Pre-NAIC_Tumor and NMPR_Pre-NAIC_Tumor samples (**d**) and MPR_Pre-NAIC_ paracarcinoma and NMPR_Pre-NAIC_ paracarcinoma (**e**) samples. **f** Projection of emergent TCR onto the UMAP embeddings of T cells in paired pre- and post-NAIC tumor tissues (P18, P20, P24). **g** Column chart of emergent TCRs post-NAIC in (**f**) (Pre-NAIC, *n* = 3; Post-NAIC, *n* = 3), data are presented as mean values with SD, the dot represents an individual data point. **h** Clone index changes in paired pre- and post-NAIC tumor tissues (P18, P20, P24).

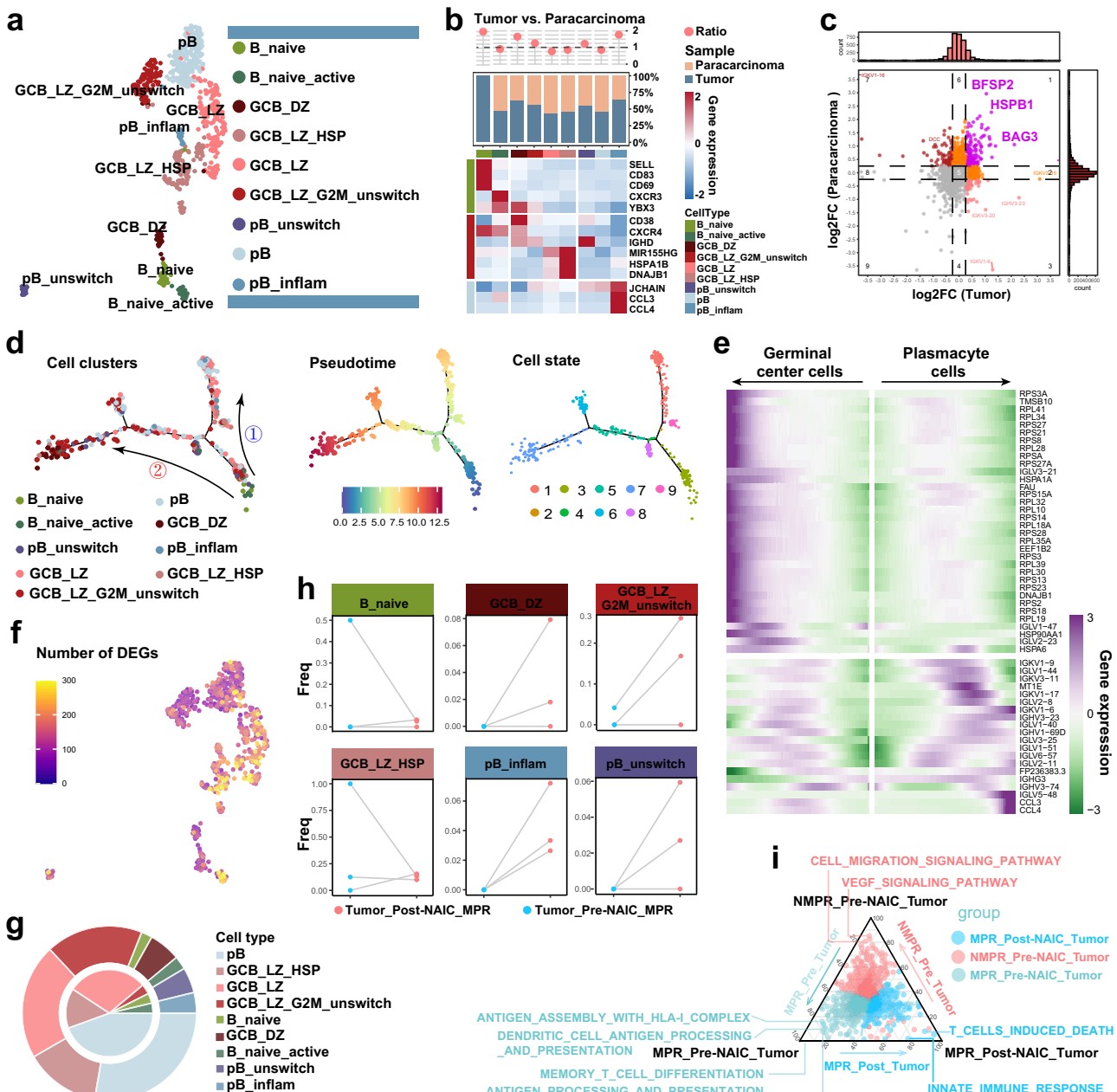

**Fig. 7 | Increased infiltration of germinal center B cells after NAIC. a** Sub-clustering of B cells, colored and labeled by subtypes. **b** Representative marker genes for each cell cluster and fractions of all B cell subtypes in paracarcinoma and tumor tissues before NAIC. **c** Differentially expressed genes (DEGs) of pre-MPR and pre-NMPR between both paracarcinoma and tumor tissues. **d** Pseudotime trajectory analyses of all B cell subtypes in tumor tissue by Monocle2. **e** Heatmap showing scaled expression of DEGs across the pseudotime trajectory in (**d**). **f** UMAP color-coded of DEGs in each B cell subtype between pre-MPR and pre-NMPR in tumor tissues. **g** Pie chart of changes in B cell subtype composition before and after NAIC, outer represents post-NAIC, inner represents pre-NAIC. **h** Changes in infiltration of each B cell subtype for paired pre- and post-NAIC tumor tissues (P18, P20, P24) (Tumor_Pre-NAIC_MPR, *n* = 3; Tumor_Post-NAIC_MPR, *n* = 3). **i** Specific pathways in MPR_Pre-NAIC_Tumor, NMPR_Pre-NAIC_Tumor and MPR_Post-NAIC_Tumor samples. MPR, major pathologic response; NMPR, non-major pathologic response; GCB_DZ, germinal center B cells in the dark zone; GCB_LZ, germinal center B cells in the light zone.

and other solid tumors[48,49], the correlation between pathological response and survival benefits in HNSCC (including OSCC) remains debated. Our findings are preliminary and require validation in future larger-scale studies. In addition, this study cannot determine the specific immunostimulatory or immunosuppressive effects of each agent in the NAIC regimen, nor how to optimize these effects. Further investigation is required to clarify the exact role of each agent in modulating immune response.

In conclusion, neoadjuvant camrelizumab plus nab-paclitaxel and cisplatin demonstrated a promising pathological response and acceptable safety profile in patients with LA-OSCC. Further exploratory

analysis indicated CD4_Tfh_CXCL13 cells as an important predictive biomarker for NAIC, which may help identify patients who are most likely to benefit from the treatment regimen. Our findings may provide clinical and theoretical insights for further validation in phase III trials.

## Methods

### Study design and participants
This single-arm, phase II trial was conducted at West China Hospital, Sichuan University and was approval by the ethics committee of our center. The trial was registered with the Chinese Clinical Trial Registry

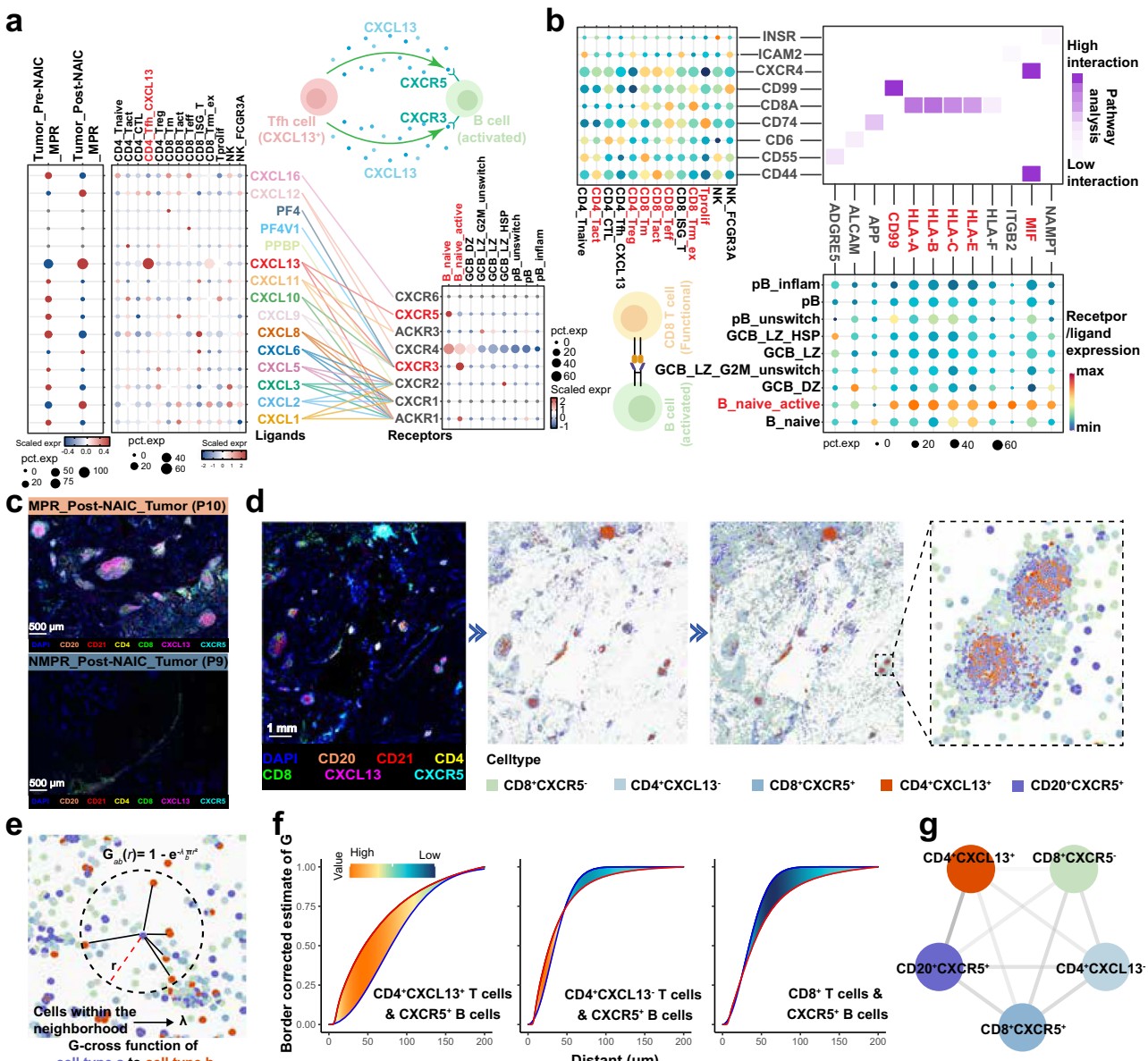

**Fig. 8 | Cell-cell communication and spatial analysis. a** Ligand-receptor analysis based on Cellchat; expression heatmap of chemokine genes in T cell clusters (left) and B clusters (right), and interactions are connected by colored lines (Cells figures were created in Adobe Illustrator 2024). **b** Ligand-receptor analysis based on Cellchat; dot plots show the top predicted ligands in B cells (bottom) and their targets in T cells (left); the heatmap shows the potential interaction (middle) (Cells figures were created in Adobe Illustrator 2024). **c** Comparision of mature pheno-types of tertiary lymphoid structures validated by mIF between MPR_Post-NAIC_Tumor (n = 20) and NMPR_Post-NAIC_Tumor (n = 9). The representative image was from patients P10 and P9, respectively. **d** Spatial analysis based on mIF staining slide (n = 3). The representative image was from patient P10. **e**, **f** G-cross function analysis of spatial distributions of CD8+CXCR5-, CD4+CXCL13-, CD8+CXCR5+, CD4+CXCL13+, and CD20+CXCR5+ cells. **g** Jaccard index analysis of CD8+CXCR5-, CD4+CXCL13-, CD8+CXCR5+, CD4+CXCL13+, and CD20+CXCR5+ cells. NAIC, neoadjuvant immunochemotherapy; MPR, major pathologic response; NMPR, non-major pathologic response.

(ChiCTR2200066119) on November 24, 2022, and adhered to the Declaration of Helsinki and Good Clinical Practice guidelines. Written consent was secured from all participants prior to their enrollment in this study. The authors confirm that explicit authorization to repro-duce the visual data presented in Fig. 2C was obtained from the involved research subjects.

Inclusion criteria included: aged 18–75 years; newly diagnosed, histologically confirmed, resectable stage III-IVB OSCC according to the 8th Union for International Cancer Control and American Joint Committee on Cancer (stage IVB patients were eligible only if they had no skull base invasion, no internal carotid artery encasement, and were resectable after strict evaluation by surgeons before enrollment); Eastern Cooperative Oncology Group performance status of 0 to 1; and

adequate organ function. Exclusion criteria included: distant metas-tasis; refusal of surgery; intent for palliative treatment; previous treatments with chemotherapy, radiotherapy, targeted therapy, anti-PD-L1/anti-PD1 agents or surgery for primary or metastatic nodes (except biopsy); previous malignancy; pregnancy or lactation; severe comorbidities; autoimmune diseases; and intolerance to immunotherapy.

## Procedures

Eligible patients received 2 cycles of NAIC, with camrelizumab 200 mg and nab-paclitaxel 260 mg/m² administered intravenously on days 1 and 22 and cisplatin 75 mg/m² (divided into 3 days) on days 1–3 and days 22-24. Surgery was performed within 2–4 weeks after

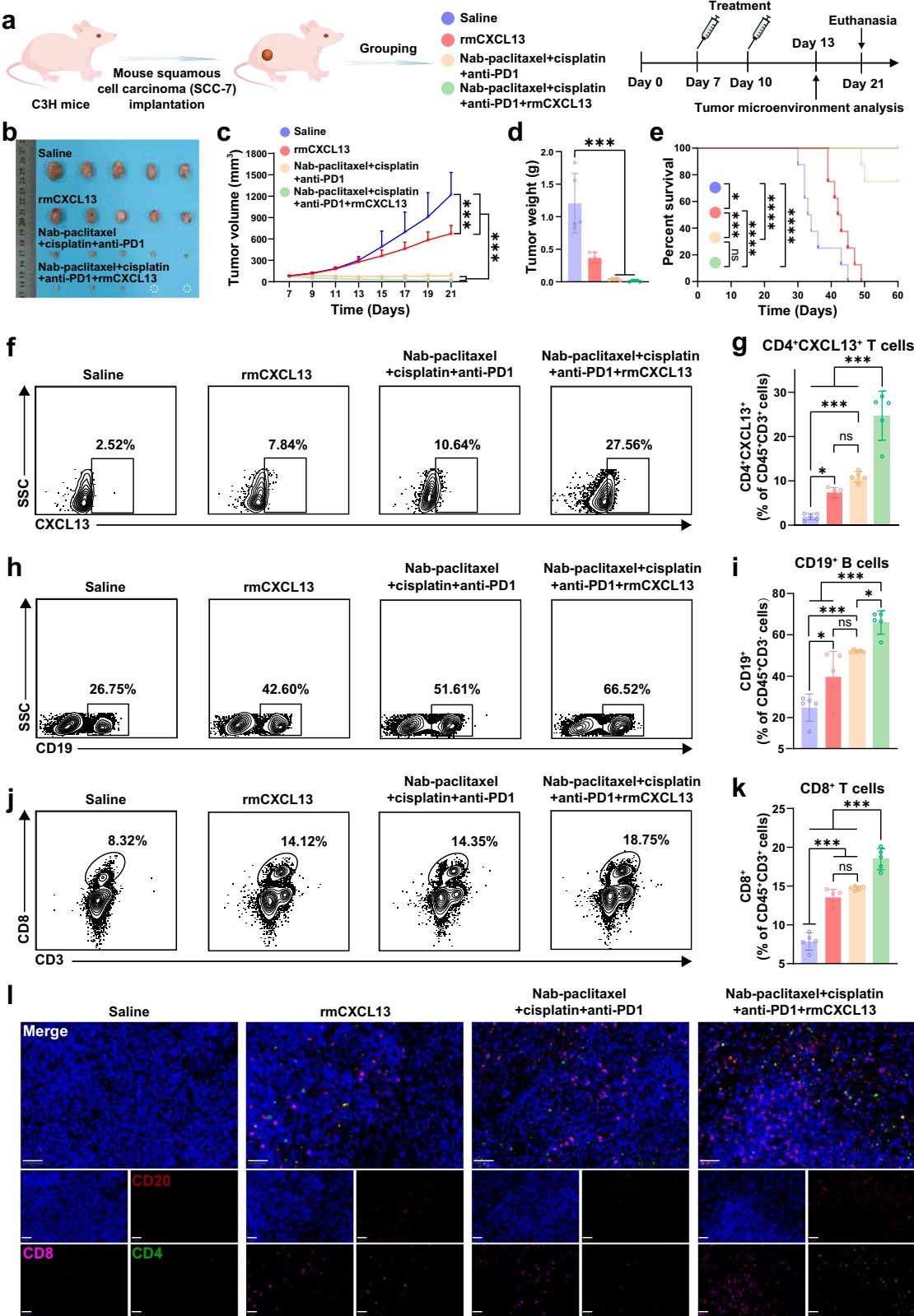

NAIC. Adjuvant radiotherapy or chemoradiotherapy was initiated within 6 weeks after surgery according to the National Comprehensive Cancer Network Clinical Practice Guidelines[50]. Maintenance immunotherapy with camrelizumab 200 mg for 6 cycles was planned after surgery. Each treatment cycle was 21 days. Dose delay or termination of camrelizumab were permitted to manage adverse events (AEs), but dose reduction of camrelizumab was not allowed.

Surgery was carried out according to the standard procedures. Before NAIC, the primary tumor range was recorded by radiographic examinations and photographs. After NAIC and before surgery, tumor characteristics were recorded by the same methods. Surgical resection was determined by pre-NAIC tumor range, regardless of post-NAIC lesion shrinkage. The safety margin and marginal biopsy was 1.0−1.5 cm away from the pre-NAIC recorded lesion. Pedicle or free flap

**Fig. 9 | CXCL13 augmented the anti-tumor immunity of immunochemotherapy in vivo. a** Schematic illustration of the experimental design (Created in Adobe Illustrator 2024). **b** Representative tumor images 14 days after treatment; white dotted circles indicate no residual tumor. **c** Tumor growth curves ($n = 5$ in each group), data are presented as mean values with SD. Two-way ANOVA with Tukey's multiple comparisons test was used for statistical analysis. **d** Tumor weights measured 14 days after treatment ($n = 5$ in each group), data are presented as mean values with SD, the dot represents an individual data point. One-way ANOVA with Tukey's multiple comparisons test was used for statistical analysis. **e** Mice survival curves. Log-rank test was used for statistical analysis. **f–k** Ratios of CD4+CXCL13+ T cells (**f, g**), CD3-CD19+ B cells (**h, i**), and CD3+CD8+ B cells (**j, k**) in tumors, analyzed by flow cytometry ($n = 5$ in each group), data are presented as mean values with SD, the dot represents an individual data point. One-way ANOVA with Tukey's multiple comparisons test was used for statistical analysis. **l** Multiplex immunofluorescence analysis of CD4+, CD8+ T cells and CD20+ B cells in tumor tissues; scale bar = 50 μm. ns, not significant; *$p < 0.05$, ***$p < 0.001$. Source data and $p$-values are presented as a Source Data file.

reconstruction technology was performed to ensure radical resection. We strictly enrolled patients with resectable stage IVB (T4bN0-3M0) lesions without skull base invasion and internal carotid artery encasement and tried to achieve the same resection range as much as possible. For cervical lymph nodes, radical dissection is a classic procedure, and we strictly followed this standard procedure. The dissection of ipsilateral or bilateral neck lymph node mainly depended on the N stage before NAIC. Meanwhile, the primary lesion and lymph nodes were all resected according to the same scope stated above, regardless of radiographic evaluation outcomes (e.g., CR).

Resected tumors and neck lymph nodes were fixed in a 4% formaldehyde solution for hematoxylin and eosin (H&E) staining. Following established methods[51], complete cross-sections of the longest dimensions of the tumor bed were embedded in paraffin and processed into H&E slides for pathological response evaluation. The %RVT was calculated by two experienced pathologists (each with over 10 years of experience) in a blinded manner. In detail, for gross specimen handling, a complete cross-section from the longest dimension of the tumor was obtained for paraffin embedding and slide preparation, with additional sections taken at 1 cm intervals from the remaining specimen. Continuous paraffin-embedded tumor slides were stained with H&E staining, and %RVT was calculated as "RVT surface area/total tumor bed surface area × 100". The whole tumor bed surface area included RVT, necrosis, tumor-associated stroma, and the tumor regression bed. Specimens were scored as 0%, 0< and <10%, 10% RVT, and increasing 10% increase. %RVT was calculated by "the whole RVT surface area in all slides/the whole tumor bed surface area in all slides". The pathological response of cervical lymph nodes was assessed by the same calculation. Radiographic response was evaluated based on the MRI scanning pre- and post- NAIC according to the Response Evaluation Criteria in Solid Tumors version 1.1 (RECIST v1.1)[52]. scRNA-seq and scTCR-seq were conducted on fresh primary tumor and paracarcinoma tissues before (biopsy) and after (surgery) NAIC. Formaldehyde-fixed primary tumor tissues before and after NAIC were also prepared for mIF staining.

## Outcomes

The primary endpoints were MPR (defined as %RVT ≤ 10%) rate and safety. The secondary endpoints included pCR (defined as no tumor cells on H&E slides) rate, objective response rate (ORR), disease-free survival (DFS; defined as the time from surgery to the first documented disease recurrence or death from any cause, with patients having no recurrence or death censored on their last disease evaluation date), and overall survival (defined as the time from initial treatment to death from any cause, with patients having no death censored on their last known survival date). AEs were graded by the Common Terminology Criteria for Adverse Events version 5.0[53]. Surgery-related complications were graded by the Clavien-Dindo system[54]. Exploratory endpoints were comprehensive marker analysis using scRNA-seq, scTCR-seq, WES, and mIFs.

## Immunohistochemistry

PD-L1 in tumors was estimated by CPS based on the PD-L1 immunohistochemistry (IHC) 22C3 pharmDx assay (Dako North America, Carpinteria CA). CPS was calculated as the percentage of PD-L1 positive tumor cells, lymphocytes, and macrophages in all viable tumor cells.

## Multiplex immunfluorescence staining

The 7-color mIF staining kit (Akoya OPAL Polaris 7-Color Automation IHC kit) was used to evaluate the infiltration of T and B lymphocytes, macrophages, NK cells, and myeloid-derived suppressor cells (MDSCs). Three antibodies panels were performed. Panel 1: CD3 (CST, dilution 1:200, catalog no.85061), CD8 (CST, dilution 1:200, catalog no.85336), CD4 (CST, dilution 1:100, catalog no.48274), CD20 (Maixin, dilution 1:200, catalog no. kit-0001), FoxP3 (CST, dilution 1:100, catalog no.98377), Pan-CK (Abcam, dilution 1:200, catalog no.7753). Panel 2: CD11c (CST, dilution 1:200, catalog no.45581), CD68 (CST, dilution 1:200, catalog no.76437), CD163 (Maixin, dilution 1:100, catalog no. MAB-0869), CD56 (Maixin, dilution 1:100, catalog no. MAB-0743), Gr−1 (Novus, dilution 1:100, catalog no. NBP2-00441), Pan-CK (Abcam, dilution 1:200, catalog no.7753). Panel 3: CXCR 5 (Abclonal, dilution 1:200, catalog no.A8950), CXCL13 (HUABIO, dilution 1:200, catalog no.HA722117), CD 20 (Maixin, dilution 1:200, catalog no. kit-0001), CD21 (Maixin, dilution 1:200, catalog no. RMA-0811), CD8 (CST, dilution 1:200, catalog no.853365), CD4 (CST, dilution 1:100, catalog no.48274). DAPI (Sigma, dilution 1:1000, catalog no.D9542) was used for nuclei visualization. Formalin-fixed, paraffinembedded (FFPE) tumor slides were deparaffinized. The first antibodies were added to the slides in panel 1-3 and incubated for one hour. Subsequently, the second antibodies were added, followed by the addition of Fluorophore-conjugated TSA® Plus Amplification Reagent. The stained mIF slides were scanned by using Vectra Polaris Software (Akoya Biosciences). QuPath-0.4.3 software was used for image reading and cell analysis, including cell type, quantity, and location.

## Whole exome sequencing and genetic analysis

Genomic DNA was extracted from FFPE slices using the HiPure FFPE DNA Kit reagent kit (Magen). Sequencing libraries were prepared using the VAHTS Universal Pro DNA Library Prep Kit reagent kit (Vazyme) according to the manufacturer's instructions. Hybridization enrichment was performed using customized xGen locking probes (integrated DNA technology) targeting cancer-related genes. Capture reactions used the xGen™ Exome Hyb Panel v2 locked hybridization and washing reagent kit (IDT). A captured library was constructed using ProFlex PCR System (Thermofisher) primers for bead PCR amplification on KAPA Library Amplification Primer Mix, followed by purification using DNA clean beads. Library quantification was performed using the KAPA HiFi HotStart Real-Time Library Amplification Kit (Roche) via quantitative PCR. Library fragment size was determined using a biological analyzer Qseq400 (BiOptic Inc.). Then, the enriched library was sequenced on Novaseq Xplus (illumina) according to the manufacturer's instructions. The average coverage depth of the tumor sample was 158x.

## Single-cell RNA-seq data pre-processing

The FASTQ files were processed and aligned to the GRCh38 human reference genome using Cell Ranger software (version 6.0.1) from 10x Genomics, with unique molecular identifier (UMI) counts summarized for each barcode. The UMI count matrix was then analyzed using Seurat[55] (version 4.0.0) R package. To remove low-quality cells and likely multiple captures, a set of criteria were conducted: Cells were filtered by (1) gene numbers (gene numbers < 200), (2) UMI (UMI < 1000),

(3) log10GenesPerUMI (log10GenesPerUMI < 0.7), (4) percentage of mitochondrial RNA UMIs (proportion of UMIs mapped to mitochondrial genes > 10%) and (5) percentage of hemoglobin RNA UMIs (proportion of UMIs mapped to hemoglobin genes > 5%). Subsequently, the DoubletFinder[56] package (version 2.0.3) was used to identify potential doublets. To obtain the normalized gene expression data, library size normalization was processed using the NormalizeData function. Specifically, the global-scaling normalization method "LogNormalize" normalized the gene expression measurements for each cell by the total expression, multiplied by a scaling factor (10,000 by default), and log-transformed the results.

Highly variable genes (HVGs) were calculated using the Seurat function FindVariableGenes (mean.function=FastExpMean, dispersion.function= FastLogVMR). Principal-component analysis (PCA) was performed using the RunPCA function to reduce the dimensionality. Graph-based clustering was performed to cluster cells according to their gene expression profile with the FindClusters function. Cells were visualized using a 2-dimensional Uniform Manifold Approximation and Projection (UMAP) algorithm with the RunUMAP function. The FindAllMarkers function (test.use = presto) was used to identify marker genes of each cluster. The detailed methods of pathway analysis, pseudotime analysis[57], cell-cell communication analysis, and scTCR-seq data analysis are shown in supplementary materials.

### Cell culture
The mouse squamous cell carcinoma cell line (SCC7) was kindly provided by the Department of Head and Neck Oncology, West China Hospital of Stomatology, State Key Laboratory of Oral Diseases, National Clinical Research Center for Oral Diseases, Sichuan University[58]. The SCC7 cells were cultured in dulbecco's modified eagle medium with 10% fetal bovine serum at 37 °C with 5% $CO_2$.

### Animal experiments
C3H mice (male, 6–8 weeks) were purchased from the BEIJING HFK BIOSCIENCE CO., LTD. Animals were housed in the SPF laboratory animal room of the State Key Laboratory of Biotherapy of Sichuan University (Chengdu, China). All mice were acclimated for 7 days before the experiment. Mice were kept in pathogen-free cages with well-ventilated shelves with a controlled temperature (22–24 °C) and light cycle (12 h light, 12 h dark) and were supplied with sufficient food and water. Animal experiments were approved by the Institutional Animal Care and Treatment Committee of Sichuan University (Chengdu, China). The detailed experiments were displayed in the supplementary information file.

### Statistical analysis
We used a Simon two-stage optimal design with an one-sided α value of 0.05 and a power of 80% to assess the efficacy of the new NAIC regimen of camrelizumab plus nab-paclitaxel and cisplatin. We assumed that the new NAIC regimen could increase the MPR to 55% compared to a historical control of 27.7% with docetaxel 75 mg/m², cisplatin 75 mg/m², and fluorouracil 750 mg/m²[25]. In the first stage, 9 patients were required. If more than 3 patients achieved MPR, recruitment would proceed to the second stage with an additional 19 patients. The total sample size was up to 28 patients in this two-stage phase 2 study. Considering a dropout rate of 10%[10,22,59], a total of 31 patients were needed in the study. If more than 11 patients achieved MPR among 28 patients eligible for the MPR evaluation, this indicated that the new regimen was worth for further exploration.

All statistical analysis were performed using R (version 4.1.3) and GraphPad Prism (version 8.0.2). For scatter plots and box plots, the intermediate line upper and lower boundaries represented the mean, 25%, and 75% percentiles, respectively. The 95% CIs were calculated by the Clopper-Pearson method for MPR rate, pCR rate, and ORR, and were calculated by the Greenwood method for survival. Spearman's correlation coefficients were used to evaluate the correlations between potential markers and %RVT. Kaplan-Meier method was used to estimate OS and DFS probabilities.

### Reporting summary
Further information on research design is available in the Nature Portfolio Reporting Summary linked to this article.

## Data availability
The trial protocol is available as a Supplementary Note in the Supplementary Information file. The anonymized patient data supporting this study's findings, including clinical demographics, tumor characteristics, safety assessments, and radiographic/pathological evaluations from neoadjuvant therapy, are provided within the article and supplementary materials. The raw sequence data targeted WES and scRNA-seq/scTCR-seq have been deposited in the Genome Sequence Archive in BIG Data Center, Beijing Institute of Genomics (BIG, http://bigd.big.ac.cn/gsa-human/) with Project Accession No. PRJCA034157 and GSA Accession No. HRA009876. The raw sequencing data contain information unique to individuals are available under controlled access, authorization by the Clinical Research Ethics Board is required as per institutional policy prior to any disclosure of participants' genomic data and individual-level clinical data, which will be made available for scholarly research objectives upon submission of a justified application to the corresponding author via liuleihx@gmail.com. The TCGA database is available at https://portal.gdc.cancer.gov/. The raw flow cytometry data and multiplex immunofluorescence staining images data generated in this study have been deposited in the Zenodo under accession https://doi.org/10.5281/zenodo.14993113. The other data are available within the manuscript, Supplementary Information, and Source file. Source data are provided in this paper.

## Code availability
R codes used to analyze data and generate figures are available at Github (https://github.com/sdzxzh/SC-neoadjuvant)[60]. The scRNA data were processed using Cell Ranger v.6.1.2 (https://www.10xgenomics.com/) and analyzed with the R package Seurat v.4.0.1 (https://satijalab.org/ seurat/). The R packages clusterProfiler v3.18.1 (https://github.com/YuLab-SMU/clusterProfiler) and msigdbr v.7.5.1 (https://github.com/igordot/msigdbr) were used to perform the GSEA.

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

## Acknowledgements

We thank all the patients who participated in this trial and their families, and the investigators and staff. This work was supported by Clinical Research Incubation Project of West China Hospital of Sichuan University (2022HXFH025 to L.L.), Sichuan Natural Science Foundation of China (2023NSFSC0706 to L.L.), the National Science Foundation of China (81472195 to L.L.), 73rd batch of general funding from China Postdoctoral Science Foundation (2023M732460 to Z.X.). Camrelizumab and nab-paclitaxel used in this clinical trial was supported by Hengrui Pharmaceuticals Co. Ltd, China. We thank Huijiao Chen (Department of Pathology, West China Hospital, Sichuan University, Chengdu, China) for her guidance on the initial pathological diagnosis of patients. We thank Bowen Zhang for his biopsy operation on the tumor (Department of Head and Neck Oncology, West China Hospital of Stomatology, Sichuan University, Chengdu, China). We thank Li Li, Fei Chen, and Chunjuan Bao (Institute of Clinical Pathology, West China Hospital, Sichuan University) for processing histological staining. We also thank Hao Zeng (Department of Biotherapy, Cancer Center & State Key Laboratory of Biotherapy, West China Hospital, Sichuan University, Chengdu, China.) for his guidance on the bioinformatic analysis.

## Author contributions

L.L. was the chief investigator of the trial. L.L., C.L., Y.L., and Z.X. conceived and designed the study. L.L., Z.X., X.W., C.T., and M.H. treated the patients. Y.L., C.L., Z.Z., and G.Z. performed the surgical procedure. X.W. and Y.Z. performed the biological samples and prepared for sequencing or pathology procedure. Y.T. analyzed the pathological response evaluation. Z.X., Y.T., and J.W. carried out tumor immune microenvironment analyses. L.C. performed all the bioinformatic analysis of multiomics sequencing data. L.C. and Z.X. developed the tables and figures. Z.X. and X.W. conducted the literature research and manuscript writing. Z.D. and N.X. followed up with patients. Z.X., X.W., Z.Z., Y.T., and L.C. contributed equally to this paper. All authors reviewed and approved the final manuscript.

## Competing interests

The authors declare no competing interests.
