## [Transparent Peer Review file · Nature Communications]

Efficacy, safety and single-cell analysis of neoadjuvant immunochemotherapy in locally advanced oral squamous cell carcinoma: a phase II trial

Corresponding Author: Dr Lei Liu

Version 0:

Reviewer comments:

Reviewer #1

(Remarks to the Author)

Report on the manuscript titled "A phase II trial and bio-radiomic analysis of neoadjuvant immunochemotherapy in locally advanced oral squamous cell carcinoma" (NCOMMS-24-45477-T):

The authors conducted a phase II trial to evaluate the efficacy and safety of neoadjuvant PD-1 inhibitor camrelizumab with nab-paclitaxel and cisplatin chemotherapy in patients with locally advanced OSCC. The primary endpoint was major pathological response (MPR). And they found that the MPR and pCR rate was 69% and 41%, respectively. Grade 3-4 neoadjuvant therapy-related AEs occurred in 2 patients. Although the aim of this study is good, there are several issues require further attention.

1. There was a problem with clinical staging, as presented in Table 1, there were 5 IVB patients, but the number of IVB patients was inconsistent with the TNM staging in Supplementary Table 5. It is necessary to explain why include stage IVB patients in this study? Is their efficacy consistent with III/IVA?
2. The protocol of surgical procedure should be described in detail, what is the principle of primary lesion resection, the principle of lymph node dissection has not been explained, what is the surgical margin? Especially when the patients archived CR, how to determine the surgical margin? For the IVB patients, how to set the scope for surgical resection?
3. On the aspect of trial protocol, there is no detail description of pathological response evaluation in the protocol, including primary tumor site and lymph node metastasis. Did you perform continuous pathological sections?
4. There were a lot of logical errors in the Supplementary Table 5.
5. The follow-up period was too short to assess tumor recurrence and survival data. Extended follow-up is recommended.
6. In Supplementary Figures, the authors used RTV (residual tumor volume) rather than RVT (Residual viable tumor) to evaluate treatment responsiveness, while in the main manuscript, the author used RVT. It is suggested to provide a specific explanation.
7. The authors mentioned CD4+CXCL13+ cells at baseline could serve as a predictive marker, however, no prediction analysis was shown in the manuscript. It is necessary to provide the specific basis for using this cell population as a predictive marker.

Reviewer #2

(Remarks to the Author)

This manuscript reports a 29 patient experience with Neoadjuvant, short course of a PD 1inhibitor camrelizumab lab plus platinum and taxane chemotherapy. The average time between treatment to surgery was 21 days. The regimen was tolerable and associated with a strong pathologic response. The authors analyze pre-and post-treatment tissue specimens for lymphocyte changes in number and phenotype. Unfortunately, after pathologic or radiographic response and surgery at 3 to 4 weeks, further survival data are not available and follow up is only a median of 10 months. This is unfortunate since it significantly reduces the novelty and information regarding durability of any immune responses. Furthermore, the single arm nature of the trial limits the changes that can be interpreted as related to PD1 inhibition versus one or the other chemotherapy or the combination of two or three agents , which all patients received. These aspects severely dampens the generalized ability and translational value to know which is the immunostimulatory or immunosuppressive aspect of this

combination or how to improve upon it. Furthermore RNA sequencing studies are descriptive and relatively observational. Others has associated TLS formation with PD-1 responsiveness. Is it prognostic or predictive, and of which agent?

Unfortunately, a major distraction in this manuscript is the frequent and repeated overstatement and over interpretation of things that are only bioinformatic associations. The authors regularly confuse changes in RNA/gene expression with functional capacities and causation, whereas there are no functional studies performed. For instance, they state that one thing led to an effect or cause in the functional activity of another cell population, without basis or validation functionally. Many of the statements are overinterpretations and may be inappropriate conclusions. The manuscript is also difficult to read from an English and syntax standpoint.

Several minor points to be mentioned include figure 1 B where the waterfall plot portrays the deep pathologic responses to the left instead of to the right, which is more standard. Similarly in Figure 2 H panel shows the major responders first then the non-responders, which is usually reversed, and the axis are very small and difficult to read from the dense figures. A native English speaker should read at line by line and correct the sentences and words which are in some cases garbled or inappropriately used.

In addition, the post treatment neoadjuvant staging seems to be missing. In table 1, we see the pre-neoadjuvant stage but not the post. Also, the primary tumor site of tongue should separately state oral or base of tongue primary tumour. They also did not seem to test for human papillomavirus, which is essential in the oropharynx. Lastly they list 31 patients in table 1 but indicate treating 29 patients. A consort diagram with patient allocation, and how patients are not listed in the response and efficacy should be stated.

Reviewer #3

(Remarks to the Author)

In this clinical and translational manuscript by Xiang et al, the investigators conduct a phase 2 Neo-adjuvant immune-chemo (NAIC) trial on 30+ patients with HNSCC and report their clinical data as well a number of well-executed translational parameters to identify novel biomarkers of response. The clinical trial is an important contribution to the literature (although I never like overstatements such as targets study etc etc), and it appears likely that the field is moving towards this approach in advanced head and neck cancer. These studies are not easy to conduct and my congratulations to the authors for completing the clinical trial in such a short period of time with remarkable tolerance and dropout rates. I have a number of queries and issues that I feel can be addressed through a revised version of the manuscript and the are divided into the clinical and translational aspects of the study:

1. Clinical:

- a. I did not see a proper consort diagram of the trial, can this be included in the main manuscript (including number of patients screened etc)
- b. although it was suggested that there was a stage reduction, was this limited to T or N stage? this was not shown
- c. for pathological assessment, were the pathologists blinded to each others review of residual tumor etc
- d. there were 2 surgical drop outs- any idea as to their response to NAIC?
- e. for the Radiologic correlation with pathological responses, was this based on CT/MRI?
- f. What was the Median time from diagnosis to surgery and did this compared to usual practice in their centre?

2. Exploratory:

- a. The figures showed the genomic data first but it was only discussed later. This somewhat disrupts the readers flow and I wonder if it can be re-arranged. Also it would be useful to show the TMB data in the co-mutation plot for easier reading.
- b. I find it odd that the investigators only used 1 radiomic feature. how was this one feature selected? it seems an odd plugin and somewhat non-intuitive to a paper that was solely about genomics and transcriptomics to add this bit? either expand on this or I am ok for this section to be removed completely as it is deceptively focussed without any basis.
- c. Minor typo in the label of figure 3A which should read paracancerous or para carcinoma but perhaps call it near adjacent 'normal'?
- d. scRNAseq data suggests that overall higher lymphocyte infiltrates = better response..can this be seen in simple h and e staining?
- e. the CD4_{tfh} population seems exceedingly small, less than 5% of the total population? is this correct? can this be identified through other techniques such as the mIF used? similarly, can mIF be used to look at the proximities of the cd4_{tfh} population with the TLS structures?
- f. This is my major issues with this paper; the whole idea of using monocle for CD4 lineage discussion is completely flawed as these cell types DO NOT transition to one another. CD4 maturation occurs in the thymus and Tregs do not become teh's in the tumors. This entire analyses and presentation needs to be re-examined! The use of monocle in B cells and CD8 is logical as these cell states/lineage transitions happen in the tumor but CD4 does not!
- g. the authors have performed some interesting scTCRseq but I do not see much in the way of the post-treatment data being shows. the claim of 6B of being post is unclear as there are no label on the axes. What are you showing? and 6i shows jus 3 tumors or 3 clones where there is expansion? what about the other data/tumors? this selective display of data is misleading.

Discussion:

1. there have been larger studies performed on just ICI alone with pretty impressive response rates and these should not be excluded just to make the authors response rates look better. the references appear to be there but discussion omits these.
2. the concept of TLS being increased after chemo or ICI is not novel. the authors should discuss the existing data on this,

and perhaps offer a view as to whether this is a cause or merely consequence of tumor regression.

Reviewer #4

(Remarks to the Author)

Abstract

Should spell out pCR?

The results of exploratory analysis take a majority, more than reporting the primary and secondary outcomes. So, this manuscript mainly focuses on bio-radiomic analysis? If yes, may revise the manuscript title.

Follow-up of long-term survival outcomes is ongoing. How long is the study period for this Phase 2 trial?

Lines 140-141: the order of ORR and pCR are inconsistent with these two in the other sections.

Line 216: Simon two-stage design is a one-sided test rather than two-sided one.

Line 222: 10% drop-out was used. Any references to justify this choice?

Lines 229-230: There are so many exploratory endpoints. What's the significance level? Should consider False Discovery Rate?

Lines 231-232: Spearman correlation coefficient should be used for two continuous variables. The term "response" in "pathological response" may be misleading since pCT and ORR are categorical values. In the efficacy section and Figures, pathological response is continuous variable, e.g., RTV%. As such, the statistical section should be clearly defined.

Lines 258-259; What does "immature" mean? Is the study ongoing? How long is the study period?

Table 1:

Any missing value for age and scanning?

The label of age "range" but the right side uses \pm which indicates standard deviation in general;

Per footnote, suggest using "age (years)" instead;

Two rows of male and female has the different format as other characteristics. Suggest adding a row of gender, and present male and female under the new row.

Version 1:

Reviewer comments:

Reviewer #1

(Remarks to the Author)

Report on the manuscript titled "Efficacy, safety and single-cell analysis of neoadjuvant immunochemotherapy in locally advanced oral squamous cell carcinoma: a phase II trial" (NCOMMS-24-45477A):

The authors have revised the manuscript according to the comments, and the quality has been significantly improved. The authors have also answered my comments, and only a few suggestions need to be further improved.

In addition to pathological evaluation for neoadjuvant immunochemotherapy, clinical stage before surgery by image evaluation should also be described for the efficacy evaluation of primary lesions and lymph node lesions. For the two patients refusing surgery, subsequent treatment and outcomes should be described in the section of Results. There are too many pictures in the body of revision, and the number of pictures needs to be reduced according to the requirements of the journal.

(Remarks on code availability)

Reviewer #3

(Remarks to the Author)

Thank you for your response. I believe all my queries have been responded to satisfactorily. Congratulations on the study.

(Remarks on code availability)

Not qualified to assess

Reviewer #4

(Remarks to the Author)

Abstract

1. Lines 37-38: "The primary endpoint was met," is confusing. "meet some criteria" or what? The primary endpoint of MPR is a proportion and thus has nothing to "meet". Or does it refer to reject the null hypothesis?
2. Line 42: "rates" is inappropriate when time-to-event analysis were used to calculate survival probabilities at some time points. If my understanding is correct, they should be "overall survival and disease-free survival probabilities at 18-month were..."

Outcomes

1. Lines 155: One (MPR) or two primary endpoints (MPR and safety)?
2. Lines 158-160: the definitions of DFS and OS are inaccurate. How about the censored patients?
3. Lines 156-157: pCR rate is before ORR. However, ORR was reported before pCR in the Efficacy sections, e.g., Lines 270.
4. Line 234: Is Simon two-stage optimal and MiniMax design?
5. Line 240: 10% drop-out was used. The references to justify this choice should be added.
6. Line 240: Suggest adding a sentence-"The total sample size is up to 28 patients in this two-stage Phase 2 study".
7. Line 241: "If more than 11 patients achieved MPR," should be among 28 patients eligible for the MPR evaluation.
8. Line 242: Same comment as 1.
9. Line 247: ORR is a proportion and "was described as frequency and percentage" does not make sense.
10. Lines 250, 283-284: "rates" is inappropriate when time-to-event analysis is used.
11. Lines 260-261: 28 patients were proposed in Simon two-stage design. However, 29 patients were included in the analysis. Even if more patients were preferred in a statistical analysis, "If more than 11 patients achieved MPR" as a criterion cannot be used in this Simon two-stage design with 29 patients since ≥ 11 are for 28 patients rather than 29 patients.

Table 1:

1. Any missing value for age? Only mean and std were shown in the first row.

(Remarks on code availability)

POINT-BY-POINT RESPONSE TO THE REVIEWER'S COMMENTS

Reviewer #1 (Remarks to the Author):

Report on the manuscript titled "A phase II trial and bio-radiomic analysis of neoadjuvant immunochemotherapy in locally advanced oral squamous cell carcinoma" (NCOMMS-24-45477-T):

The authors conducted a phase II trial to evaluate the efficacy and safety of neoadjuvant PD-1 inhibitor camrelizumab with nab-paclitaxel and cisplatin chemotherapy in patients with locally advanced OSCC. The primary endpoint was major pathological response (MPR). And they found that the MPR and pCR rate was 69% and 41%, respectively. Grade 3-4 neoadjuvant therapy-related AEs occurred in 2 patients. Although the aim of this study is good, there are several issues require further attention.

- 1. There was a problem with clinical staging, as presented in Table 1, there were 5 IVB patients, but the number of IVB patients was inconsistent with the TNM staging in Supplementary Table 5. It is necessary to explain why include stage IVB patients in this study? Is their efficacy consistent with III/IVA?**
- 2. The protocol of surgical procedure should be described in detail, what is the principle of primary lesion resection, the principle of lymph node dissection has not been explained, what is the surgical margin? Especially when the patients archived CR, how to determine the surgical margin? For the IVB patients, how to set the scope for surgical resection?**
- 3. On the aspect of trial protocol, there is no detail description of pathological response evaluation in the protocol, including primary tumor site and lymph node metastasis. Did you perform continuous pathological sections?**
- 4. There were a lot of logical errors in the Supplementary Table 5.**
- 5. The follow-up period was too short to assess tumor recurrence and survival data. Extended follow-up is recommended.**
- 6. In Supplementary Figures, the authors used RTV (residual tumor volume)**

rather than RVT (Residual viable tumor) to evaluate treatment responsiveness, while in the main manuscript, the author used RVT. It is suggested to provide a specific explanation.

7. The authors mentioned CD4+CXCL13+ cells at baseline could serve as a predictive marker, however, no prediction analysis was shown in the manuscript. It is necessary to provide the specific basis for using this cell population as a predictive marker.

Response to reviewer #1:

Dear reviewer,

Thank you for your decision and constructive comments on our manuscript. We have tried our best to improve and made some changes in the manuscript. Our response to the questions is shown below.

1. There was a problem with clinical staging, as presented in Table 1, there were 5 IVB patients, but the number of IVB patients was inconsistent with the TNM staging in Supplementary Table 5. It is necessary to explain why include stage IVB patients in this study? Is their efficacy consistent with III/IVA?

Response: Thank you for your valuable comment. A total of 31 patients (5 with stage IVB and 26 with stage III/IVA) received neoadjuvant immunochemotherapy (NAIC). Among them, 2 patients with clinical stage IVB refused surgery for personal reasons after NAIC, and 29 patients underwent surgery. Consequently, the pathological staging was performed for the 29 patients who underwent surgery. All 31 patients who received NAIC were included in **Table 1** (baseline characteristics). The original **Supplementary Table 5** (clinical to pathological downstage after neoadjuvant immunochemotherapy) only included 29 patients who underwent surgery. To avoid any confusion, we have now added the data for the 2 clinical stage IVB patients who refused surgery for personal reasons in the revised version, and the original **Supplementary Table 5** has been revised as **Supplementary Table 1** for

organizational need.

The study population in this study is locally advanced oral squamous cell carcinoma (OSCC), including resectable stage IVB (T4bN0-3M0) lesions without skull base invasion and internal carotid artery encasement. Generally, surgery is not recommended for stage T4bN0-3M0 lesions, but the NCCN Guidelines (Head and Neck Cancers, Version 2.2022 - April 26, 2022, www.nccn.org/patients) recommend that clinical trial can be also preferred. In this study, the enrolled patients were highly selected and all lesions were resectable after strict evaluation by surgeons before enrollment. The clinical stages of the 5 stage IVB patients were cT4aN3aM0, cT3N3bM0, cT3N3bM0, cT4bN1M0 (masticator space invasion), and cT4bN2aM0 (masticator space invasion). The primary tumors and cervical lymph nodes were all resectable. We noted that stage IVB patients were also included in similar studies, such as NCT04826679 (*Nat Commun*, 2024, 15(1):2177, doi: 10.1038/s41467-024-46444-z), ChiCTR1900025303 (*Clin Cancer Res*, 2022, 28(15):3268-3276, doi: 10.1158/1078-0432.CCR-22-0666), etc. **We have added detailed inclusion criteria for IVB patients in the revised manuscript (Lines: 98-101).**

For efficacy, among the 5 stage IVB patients, 3 (cT4aN3aM0, cT3N3bM0, cT3N3bM0) underwent surgery after NAIC and all achieved major pathological response (MPR) for both the primary tumor and cervical lymph nodes. The other 2 patients (cT4bN1M0 and cT4bN2aM0) refused surgery for personal reasons and withdrew from this trial. For the 26 III/IVA patients, 17 achieved MPR at the primary tumor lesions (including 11 with pCR), 14 achieved MPR at the cervical lymph nodes (including 8 with pCR). Both stage III/IVA and IVB patients showed good pathological responses for NAIC. However, since only 3 stage IVB patients underwent surgery, the exact efficacy differences between stage III/IVA and IVB patients need further investigation. **We have added these contents in the Results section (Lines: 276-280) and Supplementary Table 1 in the revised manuscript.**

Supplementary Table 1. Clinical to pathological downstage after neoadjuvant immunochemotherapy

Patient number	Clinical stage (Pre-NAIC)	Pathological stage (Post-NAIC)	Residual viable tumor (%RVT)		Radiographic response (%)	Clinical to pathological downstage	
			Primary tumor	Lymph node		Primary tumor	Lymph node
P1	cT3N0M0 (stage III)	ypT1N0M0 (stage I)	MPR (%RVT < 10%)	-	PR (-67.2%)	Yes	-
P2	cT4aN2cM0 (stage IVA)	ypT4aN2bM0 (stage IVA)	NMPR (%RVT = 90%)	NMPR (%RVT = 80%)	PR (-37.5%)	No	Yes
P3	cT4aN2cM0 (stage IVA)	ypT1N1M0 (stage III)	MPR (%RVT < 10%)	MPR (%RVT < 10%)	PR (-52.60%)	Yes	Yes
P4	cT3N2cM0 (stage IVA)	ypT2N2cM0 (stage IVA)	NMPR (%RVT = 40%)	NMPR (%RVT = 50%)	SD (-29.83%)	Yes	No
P5	cT3N1M0 (stage III)	ypT1N1M0 (stage III)	NMPR (%RVT = 30%)	NMPR (%RVT = 60%)	PR (-64.50%)	Yes	No
P6	cT3N2bM0 (stage IVA)	ypT3N2bM0 (stage IVA)	NMPR (%RVT = 70%)	NMPR (%RVT = 80%)	SD (-28.9%)	No	No
P7	cT3N1M0 (stage III)	ypT0N0M0 (-)	pCR (%RVT = 0)	pCR (%RVT = 0)	PR (-70.57%)	Yes	Yes
P8	cT3N1M0 (stage III)	ypT3N1M0 (stage III)	NMPR (%RVT = 80%)	NMPR (%RVT = 90)	SD (-9.96%)	No	No
P9	cT4aN2aM0 (stage IVA)	ypT1N2aM0 (stage IVA)	NMPR (%RVT = 40%)	NMPR (%RVT = 70%)	PR (-51.72%)	Yes	No
P10	cT4aN3aM0 (Stage IVB)	ypT1N0M0 (stage I)	MPR (%RVT < 10%)	pCR (%RVT = 0)	PR (-50.78%)	Yes	Yes
P11	cT4aN2aM0 (stage IVA)	ypT1N0M0 (stage IVA)	MPR (%RVT < 10%)	pCR (%RVT = 0)	PR (-57.55%)	Yes	Yes

P12	cT2N2aM0 (stage IVA)	ypT1N2aM0 (stage IVA)	NMPR (%RVT = 40%)	NMPR (%RVT = 60%)	SD (-21.16%)	Yes	No
P13	cT3N2aM0 (stage IVA)	ypT1N2aM0 (stage IVA)	MPR (%RVT < 10%)	MPR (%RVT < 10%)	PR (-54.83%)	Yes	No
P14	cT3N2aM0 (stage IVA)	ypT3N2aM0 (stage IVA)	NMPR (%RVT = 95%)	NMPR (%RVT = 80%)	PR (-35.11%)	No	No
P15	cT4aN0M0 (stage IVA)	ypT0N0M0 (-)	pCR (%RVT = 0)	-	PR (-43.43%)	Yes	-
P16	cT3N3bM0 (stage IVB)	ypT0N1M0 (stage III)	pCR (%RVT = 0)	MPR (%RVT < 10%)	PR (-53.96%)	Yes	Yes
P17	cT4aN0M0 (stage IVA)	ypT4aN0M0 (stage IVA)	NMPR (%RVT = 30%)	-	SD (-4.24%)	No	-
P18	cT3N3bM0 (stage IVB)	ypT1N3bM0 (stage IVB)	MPR (%RVT < 10%)	MPR (%RVT < 10%)	PR (-40.43%)	Yes	No
P19	cT3N2bM0 (stage IVA)	ypT0N2bM0 (stage IVA)	pCR (%RVT = 0)	MPR (%RVT < 10%)	PR (-61.60%)	Yes	No
P20	cT3N1M0 (stage III)	ypT0N0M0 (-)	pCR (%RVT = 0)	pCR (%RVT = 0)	PR (-66.67%)	Yes	Yes
P21	cT3N1M0 (stage III)	ypT0N0M0 (-)	pCR (%RVT = 0)	pCR (%RVT = 0)	PR (-61.94%)	Yes	Yes
P22	cT3N2aM0 (stage IVA)	ypT0N0M0 (-)	pCR (%RVT = 0)	pCR (%RVT = 0)	CR (-100%)	Yes	Yes
P23	cT4aN2bM0 (stage IVA)	ypT0N0M0 (-)	pCR (%RVT = 0)	pCR (%RVT = 0)	CR (-100%)	Yes	Yes
P24	cT3N2aM0 (stage IVA)	ypT0N0M0 (-)	pCR (%RVT = 0)	pCR (%RVT = 0)	CR (-100%)	Yes	Yes
P25	cT3N2aM0 (stage IVA)	ypT0N0M0 (-)	pCR (%RVT = 0)	pCR (%RVT = 0)	CR (-100%)	Yes	Yes
P26	cT3N0M0 (stage III)	ypT0N0M0 (-)	pCR (%RVT =	-	CR (-100%)	Yes	-

P27	cT4aN2cM0 (stage IVA)	ypT0N1M0 (stage III)	pCR (%RVT = 0)	MPR (%RVT < 10%)	CR (-100%)	Yes	Yes
P28	cT4aN2cM0 (stage IVA)	ypT1N1M0 (stage III)	MPR (%RVT < 10%)	MPR (%RVT < 10%)	PR (-82.56%)	Yes	Yes
P29	cT3N1M0 (stage III)	ypT1N1M0 (stage III)	MPR (%RVT < 10%)	MPR (%RVT < 10%)	CR (-100%)	Yes	No
P30	cT4bN1M0 (stage IVB)	-	-	-	-	-	-
P31	cT4bN2aM0 (stage IVB)	-	-	-	-	-	-

NAIC, neoadjuvant immunochemotherapy; MPR, major pathologic response; NMPR, non-major pathologic response; CPS, combined positive score; CR, complete response; PR, partial response; SD, stable disease; PD, progressive disease. %RVT, the percent of residual viable tumor. Note, P30 and P31 refused surgical resection and imaging examination for personal reasons after receiving 2 cycles of NAIC, thus the pathological stage, %RVT and radiographic response of these two patients were unevaluable.

2. *The protocol of surgical procedure should be described in detail, what is the principle of primary lesion resection, the principle of lymph node dissection has not been explained, what is the surgical margin? Especially when the patients archived CR, how to determine the surgical margin? For the IVB patients, how to set the scope for surgical resection?*

Response: Thank you for your valuable suggestion. Surgery was carried out according to the standard procedures. Before NAIC, the primary tumor range was recorded by radiographic examinations and photographs. After NAIC and before surgery, tumor characteristics were recorded by the same methods. Surgical resection was determined by pre-NAIC tumor range, regardless of post-NAIC lesion shrinkage. The safety margin and marginal biopsy was 1.0-1.5 cm away from pre-NAIC recorded lesion. Pedicle or free flap reconstruction technology was performed to ensure radical resection. We strictly enrolled patients with resectable stage IVB (T4bN0-3M0) lesions without skull base invasion and internal carotid artery encasement, and tried to achieve the same resection range as much as possible. For cervical lymph nodes, radical dissection is a classic procedure, and we strictly followed this standard procedure. The dissection of ipsilateral or bilateral neck lymph node mainly depended on the N stage before NAIC. Meanwhile, the primary lesion and lymph nodes were all resected according to the same scope stated above, regardless of radiographic evaluation outcomes (e.g., CR). We have added the detailed surgical procedure in the revised manuscript (Methods section: Lines 117-130).

3. *On the aspect of trial protocol, there is no detail description of pathological response evaluation in the protocol, including primary tumor site and lymph node metastasis. Did you perform continuous pathological sections?*

Response: Thank you for your insightful comment. We performed the pathological response evaluation according to a previously reported paper that recommends a pan-tumor (including head and neck squamous cell carcinoma) pathologic scoring method of response to PD-(L)1 blockade (*Clin Cancer Res*, 2020, 26(3):545-551, doi: 10.1158/1078-0432.CCR-19-2379). In detail, for gross specimen

handling, a complete cross-section from the longest dimension of the tumor was obtained for paraffin embedding and slide preparation, with additional sections taken at 1-cm intervals from the remaining specimen. Continuous paraffin-embedded tumor slides were stained with H&E staining, and percent residual viable tumor (%RVT) was calculated as “RTV surface area/total tumor bed surface area \times 100”. The whole tumor bed surface area included RVT, necrosis, tumor-associated stroma, and the tumor regression bed. Specimens were scored as 0%, 0< and <10%, 10% RVT, and increasing 10% increase. %RVT was calculated by “the whole RTV surface area in all slides/the whole tumor bed surface area in all slides”. The pathological response of cervical lymph nodes was assessed by the same calculation. We have added the related contents in the revised manuscript (Methods section: Lines 136-147).

4. There were a lot of logical errors in the Supplementary Table 5.

Response: Thank you for your careful review. We have rechecked the contents in the original **Supplementary Table 5**, and made some corrections. The original Supplementary Table 5 has been revised as Supplementary Table 1 for the organizational need. To make the contents more logically, we divided the “Clinical to pathological downstage” into 2 columns, respectively: “Primary tumor” and “Lymph node”, as shown in Supplementary Table 1. Meanwhile, the “%RVT = 0” was replaced by “pCR”, “None-MPR” was replaced by “NMPR”.

5. The follow-up period was too short to assess tumor recurrence and survival data. Extended follow-up is recommended.

Response: According to your constructive advice, we have extended the data cut-off on November 30, 2024. As of the data cutoff on November 30, 2024, the median follow-up duration for 31 treated patients was 18.6 months (range: 12.6-22.2). For all 31 treated patients, the 18-month OS rate was 96.77% (95%CI: 79.23%-99.54%) (**Supplementary Fig. 1**). For 29 patients who underwent surgery, the 18-month disease-free survival rate was 85.71% (95%CI: 53.95%-96.22%) (**Supplementary Fig. 2**). These have been added to the results section (Lines 281-285)

and supplemented materials (Supplementary Fig. 1, Supplementary Fig. 2). The long survival follow-up is still ongoing.

Supplementary Figure 1. Kaplan-Meier analysis of overall survival for all 31 treated patients. Source data are presented as a Source Data file.

Supplementary Figure 2. Kaplan-Meier analysis of disease-free survival for 29 patients who received neoadjuvant immunochemotherapy and surgery. Source data are presented as a Source Data file.

6. In Supplementary Figures, the authors used RTV (residual tumor volume) rather than RVT (Residual viable tumor) to evaluate treatment responsiveness, while in the main manuscript, the author used RVT. It is suggested to provide a specific explanation.

Response: Thank you for your careful review. We apologize for the typing error in the Supplementary Figures, where the incorrect acronym (RTV) was used to represent 'the percent of residual viable tumor.' The correct acronym is RVT, and the data labeled as RTV actually correspond to RVT. This error has been carefully checked and corrected.

7. The authors mentioned $CD4^+CXCL13^+$ cells at baseline could serve as a predictive marker, however, no prediction analysis was shown in the manuscript. It is necessary to provide the specific basis for using this cell population as a predictive marker.

Response: Thank you for your constructive suggestion. To evaluate the predictive value of $CD4^+CXCL13^+$ cells at baseline, we performed receiver operating characteristic curve (ROC) analysis, as shown in **Figure 5j**. The area under the curve (AUC) was 0.9251 ($p < 0.0001$). We also analyzed the survival outcomes between high and low CXCL13 expression populations from TCGA data. The results showed that head and neck carcinoma patients with high CXCL13 expression had improved overall survival and progression-free survival than those with low CXCL13 expression, as displayed in **Figure 5k, l (Lines: 372-376)**.

Figure 5. j, Receiver operating characteristic curve analysis for baseline $CD4^+CXCL13^+$ cell density to predict the percentage of residual viable tumor (%RVT)

after NAIC. k and l, Kaplan-Meier analysis of overall survival (k) and progression free survival (l) in patients with high versus low CXCL13 expression in the TCGA head and neck cancer cohort. AUC, area under curve.

We further explored the role of CXCL13 in the anti-tumor effect of immunochemotherapy *in vivo*. As shown in lines 452-465 in the revised manuscript, “SCC7-bearing mice were treated with saline, rmCXCL13, immunochemotherapy, and rmCXCL13 plus immunochemotherapy (combination) groups (Fig. 9a). Compared with the saline and rmCXCL13 groups, the combination group showed a greater inhibitory effect on tumor growth, as well as a longer mice survival time. The inhibitory effect on tumor growth was more obvious in the combination group than in the immunochemotherapy group, and two of five tumors in the combination group completely regressed (Fig. 9b-e). Flow cytometry analyses revealed that the percentages of CD4⁺CXCL13⁺, CD3⁺CD19⁺ and CD3⁺CD8⁺ cells were significantly increased in the rmCXCL13 group compared with the saline group, and ratios of these cells in the combination group were all higher than single reCXCL13 and immunochemotherapy group (Fig. 9f-k). Meanwhile, mIF confirmed the increase percentages of CD4⁺, CD8⁺ and CD20⁺ cells in the combination group (Fig. 9l). These results indicated the potential of CXCL13 in improving the anti-tumor effect of immunochemotherapy.”

Taken together, these results suggest that baseline CD4⁺CXCL13⁺ cells might be a potential predictive biomarker of efficacy.

[FIGURE REDACTED]

Figure 9. CXCL13 augments the anti-tumor immunity of immunochemotherapy *in vivo*. a, Schematic illustration of the experimental design. b, Representative tumor images 14 days after treatment; white dotted circles indicate no residual tumor. c, Tumor growth curves. d, Tumors weights measured 14 days after treatment. e, Mice

survival curves. f-k, Ratios of CD4⁺CXCL13⁺ T cells (f-g), CD3⁻CD19⁺ B cells (h-i), and CD3⁺CD8⁺ B cells (j-k) in tumors, analyzed by flow cytometry. l, Multiplex immunofluorescence analysis of CD4⁺, CD8⁺ T cells and CD20⁺ B cells in tumor tissue; scale bar = 50 μ m. *ns*, not significant, * $p < 0.05$, *** $p < 0.001$.

Reviewer #2 (Remarks to the Author):

This manuscript reports a 29 patient experience with Neoadjuvant, short course of a PD 1inhibitor camrelizumab lab plus platinum and taxane chemotherapy. The average time between treatment to surgery was 21 days. The regimen was tolerable and associated with a strong pathologic response. The authors analyze pre-and post-treatment tissue specimens for lymphocyte changes in number and phenotype. Unfortunately, after pathologic or radiographic response and surgery at 3 to 4 weeks, further survival data are not available and follow up is only a median of 10 months. This is unfortunate since it significantly reduces the novelty and information regarding durability of any immune responses. Furthermore, the single arm nature of the trial limits the changes that can be interpreted as related to PD1 inhibition versus one or the other chemotherapy or the combination of two or three agents, which all patients received. These aspects severely dampens the generalized ability and translational value to know which is the immunostimulatory or immunosuppressive aspect of this combination or how to improve upon it. Furthermore RNA sequencing studies are descriptive and relatively observational. Others has associated TLS formation with PD-1 responsiveness. Is it prognostic or predictive, and of which agent?

Unfortunately, a major distraction in this manuscript is the frequent and repeated overstatement and over interpretation of things that are only bioinformatic associations. The authors regularly confuse changes in RNA/gene expression with functional capacities and causation, whereas there are no functional studies performed. For instance, they state that one thing led to an effect or cause in the functional activity of another cell population, without basis or validation functionally. Many of the statements are overinterpretations and may be inappropriate conclusions. The manuscript is also difficult to read from an English and syntax standpoint.

Several minor points to be mentioned include figure 1 B where the waterfall plot

portrays the deep pathologic responses to the left instead of to the right, which is more standard. Similarly in Figure 2 H panel shows the major responders first then the non-responders, which is usually reversed, and the axis are very small and difficult to read from the dense figures. A native English speaker should read at line by line and correct the sentences and words which are in some cases garbled or inappropriately used.

In addition, the post treatment neoadjuvant staging seems to be missing. In table 1, we see the pre-neoadjuvant stage but not the post. Also, the primary tumor site of tongue should separately state oral or base of tongue primary tumour. They also did not seem to test for human papillomavirus, which is essential in the oropharynx. Lastly they list 31 patients in table 1 but indicate treating 29 patients. A consort diagram with patient allocation, and how patients are not listed in the response and efficacy should be stated.

Response to reviewer #2:

Dear reviewers,

Thank you very much for giving us the opportunity to revise our manuscript. Your comments and suggestions on our manuscript are also greatly appreciated. All of these comments are valuable and helpful in revising and improving our paper, as well as providing important guidance for our research. We have studied your comments carefully and have made correction which we hope meet with approval. Please find the following detailed responses to your comments and suggestions.

1. Unfortunately, after pathologic or radiographic response and surgery at 3 to 4 weeks, further survival data are not available and follow up is only a median of 10 months. This is unfortunate since it significantly reduces the novelty and information regarding durability of any immune responses.

Response: Thanks for your constructive advice. A total of 31 patients received neoadjuvant immunochemotherapy (NAIC). Among them, 2 patients refused surgery

for personal reasons after NAIC, and 29 patients underwent surgery. We have extended the data cut-off on November 30, 2024, and performed Kaplan-Meier analyses for both all 31 patients who received NAIC with or without surgery and 29 patients who underwent surgery after NAIC. As of the updated data cut-off, the median follow-up duration for 31 treated patients was 18.6 months (range: 12.6-22.2). For all 31 treated patients, the 18-month OS rate was 96.77% (95%CI: 79.23%-99.54%) (**Supplementary Fig. 1**). For 29 patients who underwent surgery, the 18-month disease-free survival rate was 85.71% (95%CI: 53.95%-96.22%) (**Supplementary Fig. 2**). These have been added to the results section (Lines 281-285) and supplemented materials (Supplementary Fig. 1, Supplementary Fig. 2). The long survival follow-up is still ongoing.

Supplementary Figure 1. Kaplan-Meier analysis of overall survival for all 31 treated patients. Source data are presented as a Source Data file.

Supplementary Figure 2. Kaplan-Meier analysis of disease-free survival for 29 patients who received neoadjuvant immunochemotherapy and surgery. Source data are presented as a Source Data file.

2. Furthermore, the single arm nature of the trial limits the changes that can be interpreted as related to PD1 inhibition versus one or the other chemotherapy or the combination of two or three agents , which all patients received. These aspects severely dampens the generalized ability and translational value to know which is the immunostimulatory or immunosuppressive aspect of this combination or how to improve upon it.

Response: Thank you for your insightful comments. ICIs-based neoadjuvant treatment has been a research highlight for OSCC and HNSCC, including mono ICIs, dual ICIs, ICIs plus targeted therapy, ICIs plus radiotherapy, and ICIs plus chemotherapy, etc. Previous studies reported the following pathologic response in locally advanced head and neck squamous cell carcinoma: neoadjuvant mono/dual ICB (MPR rate: 5.9% - 35%; pCR rate: 0 - 10%), ICB combined with targeted therapy ± chemotherapy (MPR rate: 40% - 60%; pCR rate: 5% - 40%), ICB combined with radiotherapy ± chemotherapy (MPR rate: 43.0% - 60%; pCR rate: 20.0% - 57.0%; including p16 positive patients), and ICB combined with chemotherapy (MPR rate: 27.8% - 74.1%; pCR rate: 16.7% - 55.6%), etc ^[1-19]. Compared with these results, the regimen of camrelizumab plus nab-paclitaxel and cisplatin in this study appeared

to demonstrate a relatively higher pathologic response, with an MPR rate of 69% and a pCR rate of 41.4%. However, there is currently no strong evidence to identify which treatment strategy is more effective. From the data currently reported, compared with mono/dual ICIs, ICIs-based combined treatment has shown promising major pathological response and/or pathologic complete response rates. Among these combined strategies, ICIs plus chemotherapy (NAIC) appears particularly effective. Therefore, we reported the preliminary results of this phase II trial.

Our single-arm exploratory study has some weaknesses and limitations, especially in terms of the immunostimulatory or immunosuppressive aspect of this strategy or how to improve upon it. These limitations have been added in the Discussion section (**Lines 551-555**), shown as “In addition, this study cannot determine the specific immunostimulatory or immunosuppressive effects of each agent in the NAIC regimen, nor how to optimize these effects. Further investigation is required to clarify the exact role of each agent in modulating immune response.”

In addition, to further explore these aspects, we performed mice experiments to preliminarily identify which agent in the regimen acted as immunostimulatory or immunosuppressive role. As shown in **Supplementary figure 7**, chemotherapy alone did not induce a favorable anti-tumor immune response, but significantly enhanced the ICIs-triggered anti-tumor immunity. We have added these findings in the Results section (**Lines 434-447**) as follows:

“Chemotherapy enhanced the anti-tumor immune response to ICIs *in vivo*

To identify which drug in this immunochemotherapy regimen functioned as an immunostimulatory or immunosuppressive role, we performed *in vivo* experiments on SCC7-bearing mice. Mice were treated with saline, cisplatin, nab-paclitaxel, nab-paclitaxel+cisplatin, anti-PD1, or nab-paclitaxel+cisplatin+anti-PD1 when the tumor volume reached about 80 mm³ (**Supplementary Fig. 7a**). We found anti-PD1+chemotherapy significantly suppressed tumor growth and prolonged mice survival compared to other groups (**Supplementary Fig. 7b-e**). Meanwhile, flow cytometry analyses illustrated that the percentages of CD4⁺CXCL13⁺, CD3⁺CD19⁺

and CD3⁺CD8⁺ cells were significantly increased in the anti-PD1 and nab-paclitaxel+cisplatin+anti-PD1 groups compared to other groups (**Supplementary Fig. 7f-k**). These results suggest that, among those agents, anti-PD-1 shown a more favourable immune activation function, and anti-PD1 combined with chemotherapy further enhanced the anti-tumor immunity”.

[FIGURE REDACTED]

Supplementary Figure 7. Exploration of the immune function of each agent in the immunochemotherapy regimen *in vivo*. a, Schematic illustration of the experimental design. b, Representative tumors after 14 days of treatment. c, Tumor growth curves after different treatments. d, Tumor weights after 14 days of treatment. e, Survival curves of mice after different treatments. f and g, CD4⁺CXCL13⁺ T cell ratios in tumors after different treatments by flow cytometry analysis. h and i, CD3⁺CD19⁺ B cell ratios in tumors after different treatments by flow cytometry

analysis. j and k, CD3⁺CD8⁺ T cell ratios in tumors after different treatments by flow cytometry analysis. *ns*, not significant, * $p < 0.05$, ** $p < 0.01$, *** $p < 0.001$.

References

[1] Schoenfeld, J. D. et al. Neoadjuvant Nivolumab or Nivolumab Plus Ipilimumab in Untreated Oral Cavity Squamous Cell Carcinoma: A Phase 2 Open-Label Randomized Clinical Trial. *JAMA. Oncol.* **6**, 1563-1570 (2020).

[2] Uppaluri, R. et al. Neoadjuvant and Adjuvant Pembrolizumab in Resectable Locally Advanced, Human Papillomavirus-Unrelated Head and Neck Cancer: A Multicenter, Phase II Trial. *Clin. Cancer. Res* **26**, 5140-5152 (2020).

[3] Knochelmann, H. M. et al. Neoadjuvant presurgical PD-1 inhibition in oral cavity squamous cell carcinoma. *Cell. Rep. Med.* **2**, 100426 (2021).

[4] Ferris, R. L. et al. Neoadjuvant nivolumab for patients with resectable HPV-positive and HPV-negative squamous cell carcinomas of the head and neck in the CheckMate 358 trial. *J. Immunother. Cancer.* **9**, e002568 (2021).

[5] Vos, J. L. et al. Neoadjuvant immunotherapy with nivolumab and ipilimumab induces major pathological responses in patients with head and neck squamous cell carcinoma. *Nat. Commun* **12**, 7348 (2021).

[6] Ferrarotto, R. et al. Impact of Neoadjuvant Durvalumab with or without Tremelimumab on CD8⁺ Tumor Lymphocyte Density, Safety, and Efficacy in Patients with Oropharynx Cancer: CIAO Trial Results. *Clin. Cancer. Res.* **26**, 3211-3219 (2020).

[7] Ferris, R. L. et al. Neoadjuvant nivolumab alone or in combination with relatlimab or ipilimumab in resectable head and neck squamous cell carcinoma (HNSCC) [J/OL]. *J. Clin. Oncol.* **41**, 6018 (2023).

[8] Ju, W.T. et al. A pilot study of neoadjuvant combination of anti-PD-1 camrelizumab and VEGFR2 inhibitor apatinib for locally advanced resectable oral squamous cell carcinoma. *Nat. Commun.* **13**, 5378 (2022).

[9] Winston, Wong. et al. Neoadjuvant cemiplimab with platinum-doublet chemotherapy and cetuximab to de-escalate surgery and omit adjuvant radiation in locoregionally advanced head & neck squamous cell carcinoma (HNSCC). *J. Clin. Oncol.* **41**, 6019 (2023).

[10] Leidner, R. et al. Neoadjuvant immunoradiotherapy results in high rate of complete pathological response and clinical to pathological downstaging in locally

advanced head and neck squamous cell carcinoma. *J. Immunother. Cancer*. **9**, e002485 (2021).

[11] Liu, Z. et al. Neoadjuvant low-dose radiotherapy, tislelizumab, combined with albumin-bound paclitaxel and cisplatin in resectable locally advanced head and neck squamous cell carcinoma (NeoRTPC02): the first-stage result from an open label, single-arm, stage two, phase II clinical trial[J/OL]. *J. Clin. Oncol.* **41**, 607 (2023).

[12] Patel, S. A. et al. A phase 2 study of neoadjuvant chemotherapy plus durvalumab in resectable locally advanced head and neck squamous cell carcinoma. *Cancer* **129**, 3381-3389 (2023).

[13] Wu, D. et al. Neoadjuvant chemo-immunotherapy with camrelizumab plus nab-paclitaxel and cisplatin in resectable locally advanced squamous cell carcinoma of the head and neck: a pilot phase II trial. *Nat. Commun.* **15**, 2177 (2024).

[14] Zhang, Z. et al. Neoadjuvant Chemoimmunotherapy for the Treatment of Locally Advanced Head and Neck Squamous Cell Carcinoma: A Single-Arm Phase 2 Clinical Trial. *Clin. Cancer Res.* **28**, 3268-3276 (2022).

[15] Wang, K. et al. Efficacy and safety of pembrolizumab with preoperative neoadjuvant chemotherapy in patients with resectable locally advanced head and neck squamous cell carcinomas. *Front. Immunol.* **14**, 1189752 (2023).

[16] Rosenberg, A. et al. Neoadjuvant nivolumab, paclitaxel, and carboplatin followed by response-stratified chemoradiation in locoregionally advanced HPV negative head and neck squamous cell carcinoma (HNSCC): The DEPEND trial. *J. Clin. Oncol.* **41**, 6007 (2023).

[17] Huang, X. et al. Neoadjuvant toripalimab combined with gemcitabine and cisplatin in resectable locally advanced head and neck squamous cell carcinoma (NeoTGP01): An open label, single-arm, phase Ib clinical trial. *J. Exp. Clin. Cancer Res* **41**, 300 (2022).

[18] Zinner, R. et al. Neoadjuvant nivolumab (N) plus weekly carboplatin (C) and paclitaxel (P) in resectable locally advanced head and neck cancer. *J. Clin. Oncol* **38**, 6583 (2020).

[19] Huang, Y. et al. Neoadjuvant immunochemotherapy for locally advanced resectable oral squamous cell carcinoma: a prospective single-arm trial (Illuminate Trial). *Int. J. Surg.* **109**, 2220-2227 (2023).

3. Furthermore RNA sequencing studies are descriptive and relatively observational. Others has associated TLS formation with PD-1 responsiveness. Is it prognostic or predictive, and of which agent?

Response: Thank you for your constructive comments. According to your valuable comments, we have realized the limitation of scRNA-seq and our exaggerated statements in our initial submitted manuscript. In the revised version, we have reorganized the English writing and supplemented clinical analysis and mice experiments to further explore these above concerns. In multiplex immunofluorescence analysis, the difference in TLS numbers between MPR and NMMPR patients was not significant before NAIC; however, after NAIC, the TLS density statistically significantly increased in MPR patients compared with NMMPR populations (**Figure 3a-d**). Additionally, TLS density showed negative correlation with %RVT after NAIC (**Supplementary Figure 3b**). Therefore, the enrichment of TLS significantly related with the efficacy of NAIC, and was a potential predictive marker for anti-PD1 responsiveness.

To identify which agent in NAIC promote the formation of TLS, we performed mice experiments. As shown in **Supplementary Figure 7**, ICIs alone promoted the increased ratios of CD4⁺CXCL13⁺ T, CD3⁻CD19⁺ B cells, and CD3⁺CD8⁺ T cells in tumors. However, neither single or double chemotherapy agent induced a high ratio of these cells in tumors. When chemotherapy was combined with ICIs, a more favorable tumor immune microenvironment was triggered compared with other single agent, indicating that the induction of TLS mainly depends on the function of ICIs, while ICIs plus chemotherapy further activated the anti-tumor immunity.

Figure 3. Tertiary lymphoid structures and TME features in MPR and NMMPR patients before and after NAIC. a, H&E staining, TLSs were marked by blue circles. b and c, Multiplex immunofluorescence (mIF) staining for TLS and TLS-related markers. d, e, f, Quantification of TLSs, CD8⁺ T cells, and CD20⁺ B cells based on mIF staining analysis in MPR (n = 20) and NMMPR (n = 9) patients before and after

NAIC. g, Representative mIF images showing dendritic cells (CD11c⁺), natural killer cells (CD56⁺), and M2 type tumor associated macrophages (CD68⁺CD163⁺). h-j, Quantification of CD56⁺, CD11c⁺, and CD68⁺CD163⁺ cells based on mIF staining analysis in MPR and NMPR patients before and after NAIC. k, Heatmaps illustrating the proportions of the above immune cells. TLS, tertiary lymphoid structures; NAIC, neoadjuvant immunochemotherapy; MPR, major pathologic response; NMPR, non-major pathologic response.

Supplementary Figure 3. Correlation between different factors and %RVT. a, Correlation between PD-L1 (CPS) and %RVT. b, Correlation between TLS and %RVT after NAIC. c, Correlation between CD8⁺ T cells density and %RVT after NAIC. d, Correlation between CD20⁺ cells and %RVT after NAIC. %RVT, percent of residual viable tumor; TLS, tertiary lymphoid structures; NAIC, neoadjuvant immunochemotherapy; CPS, combined positive score.

[FIGURE REDACTED]

Supplementary Figure 7. Exploration of the immune function of each agent in the immunochemotherapy regimen *in vivo*. a, Schematic illustration of the experimental design. b, Representative tumors after 14 days of treatment. c, Tumor growth curves after different treatments. d, Tumor weights after 14 days of treatment. e, Survival curves of mice after different treatments. f and g, CD4⁺CXCL13⁺ T cell ratios in tumors after different treatments by flow cytometry analysis. h and i, CD3⁻CD19⁺ B cell ratios in tumors after different treatments by flow cytometry analysis. j and k, CD3⁺CD8⁺ T cell ratios in tumors after different treatments by flow cytometry analysis. *ns*, not significant, * $p < 0.05$, ** $p < 0.01$, *** $p < 0.001$.

4. Unfortunately, a major distraction in this manuscript is the frequent and repeated overstatement and over interpretation of things that are only bioinformatic associations. The authors regularly confuse changes in RNA/gene expression with functional capacities and causation, whereas there are no functional studies performed. For instance, they state that one thing led to an effect or cause in the functional activity of another cell population, without basis or validation functionally. Many of the statements are overinterpretations and may be inappropriate conclusions.

Response: Thank you for your thoughtful and constructive comments. According to your suggestion, we have been aware of the repeated overstatement and over interpretation of the results, and carefully revised these inappropriate statements in the revised manuscript. Thanks again for your advice, which can significantly improve the quality of our manuscript. Also, we have added functional validation experiments.

Firstly, we explore the association between CD4_Tfh_CXCL13 cells and TLS by adding spatial analysis based on patients multiplex immunofluorescence staining slides. As shown in **lines 417-432 (Figure 8)** in the revised manuscript:

“Association between CD4_Tfh_CXCL13 cells and TLS

Cell-cell communication analysis revealed that CXCL13 (ligand) on CD4_Tfh_CXCL13 cells interacted with CXCR5 (receptor) on B_naïve and CXCR3 (receptor) on B_naïve_active cells (**Fig. 8a**). The activated naive B cells promoted reciprocal signaling with CD8 T cell subsets (CD8_Teff, CD8_Tm, CD8_Trm_ex) via MHC I pathway (**Fig. 8b**). These results were supported by mIF analysis, which showed that post-NAIC, MPR patients had an increased number of TLS with mature phenotypes (reticulate FDCs and germinal center) and widespread distribution of CXCL13 and CXCR5 positive cells (**Fig. 8c, Supplementary Fig. 6**). Moreover, in germinal center, CD4⁺CXCL13⁺ cells were obviously adjacent to CD20⁺CXCR5⁺ cells on multiplex immunofluorescence staining slides (**Fig. 8d**), further spatial analysis by G-cross function and Jaccard index analyses showed that CD4⁺CXCL13⁺ cells were most frequently co-localized with CD20⁺CXCR5⁺ cells in the TLS, whereas other CD4⁺CXCL13⁻ cells presented with a poor co-localization to CD20⁺CXCR5⁺ cells

(Fig. 8e-g). These data suggest that interaction between CD4_Tfh_CXCL13 cells and B_naïve cells might be a potential mechanism for the immune response during NAIC.”

Figure 8. Cell-cell communication and spatial analysis. a, Ligand-receptor analysis based on Cellchat; expression heatmap of chemokine genes in T cell clusters (left) and B clusters (right), and interactions are connected by colored lines. b, Ligand-receptor analysis based on Cellchat; dot plots show the top predicted ligands in B cells (bottom) and their targets in T cells (left); the heatmap shows the potential interaction (middle). c, Mature phenotypes of tertiary lymphoid structures validated by mIF. The representative image was from patient P9. d, Spatial analysis based on mIF staining slide. The representative image was from patient P9. e and f, G-cross function analysis of spatial distributions of CD8+CXCR5+, CD4+CXCL13+, CD8+CXCR5+, CD4+CXCL13+, and CD20+CXCR5+ cells. g, Jaccard index analysis of CD8+CXCR5+,

CD4⁺CXCL13⁻, CD8⁺CXCR5⁺, CD4⁺CXCL13⁺, and CD20⁺CXCR5⁺ cells. NAIC, neoadjuvant immunochemotherapy; MPR, major pathologic response; NMPR, non-major pathologic response.

Then, we performed mice experiment to further validate the function of CXCL13 during the treatment of immunochemotherapy. As shown in Results section (Lines 449-465, Figure 9) in the revised manuscript:

“CXCL13 augmented the anti-tumor effect of immunochemotherapy *in vivo*

Considering the association between CXCL13 and pathological response, we explored the role of CXCL13 in the anti-tumor effect of immunochemotherapy *in vivo*. SCC7-bearing mice were divided into saline, rmCXCL13, immunochemotherapy, and rmCXCL13 plus immunochemotherapy (combination) groups (**Fig. 9a**). Compared with the saline and rmCXCL13 groups, the combination group showed a greater inhibitory effect on tumor growth and longer survival duration. The inhibitory effect on tumor growth was more obvious in the combination group than in the immunochemotherapy group, and two of five tumors in the combination group completely regressed (**Fig. 9b-e**). Flow cytometry analyses revealed that the percentages of CD4⁺CXCL13⁺, CD3⁻CD19⁺ and CD3⁺CD8⁺ cells were significantly increased in the rmCXCL13 group compared with the saline group, and ratios of these cells in the combination group were all higher than single reCXCL13 and immunochemotherapy group (**Fig. 9f-k**). Meanwhile, mIF confirmed the increase percentages of CD4⁺, CD8⁺ and CD20⁺ cells in the combination group (**Fig. 9l**). These results indicated the potential of CXCL13 in improving the anti-tumor effect of immunochemotherapy.”

[FIGURE REDACTED]

Figure 9. CXCL13 augmented the anti-tumor immunity of immunotherapy *in vivo*. a, Schematic illustration of the experimental design. b, Representative tumor images 14 days after treatment; white dotted circles indicate no residual tumor. c, Tumor growth curves. d, Tumors weights measured 14 days after

treatment. e, Mice survival curves. f-k, Ratios of CD4⁺CXCL13⁺ T cells (f-g), CD3⁺CD19⁺ B cells (h-i), and CD3⁺CD8⁺ B cells (j-k) in tumors, analyzed by flow cytometry. l, Multiplex immunofluorescence analysis of CD4⁺, CD8⁺ T cells and CD20⁺ B cells in tumor tissue; scale bar = 50 μm. *ns*, not significant, * $p < 0.05$, *** $p < 0.001$.

5. The manuscript is also difficult to read from an English and syntax standpoint.

Response: Thank you very much for your comments. We have invited a native English speaker to carefully checked the whole manuscript and revised the unintelligible English and syntax by a series of checks. if these revisions were not appropriate, we can revise again.

6. Several minor points to be mentioned include figure 1 B where the waterfall plot portrays the deep pathologic responses to the left instead of to the right, which is more standard. Similarly in Figure 2 H panel shows the major responders first then the non-responders, which is usually reversed, and the axis are very small and difficult to read from the dense figures. A native English speaker should read at line by line and correct the sentences and words which are in some cases garbled or inappropriately used.

Response: Thank you for your constructive comments and suggestions. We have rearranged the displays of “pathologic responses”, “major responders” and “non-responders” parts in all the figures according to your advice, including the presentation orders and axis notes. We have invited a native English speaker to carefully checked the whole manuscript, and revised the garbled or inappropriately used contents. Thanks again for your kind reminder. For the consideration of organizational need, Figure 1 and Figure 2 in the initially submitted manuscript have been rearranged as Figure 2 and Figure 3, respectively, in the revised manuscript.

Figure 2. Clinical responses to NAIC. a, Clinical characteristics and pathologic response. b, Radiographic response per RECIST v1.1 criteria. c, Representative images of patients who achieved MPR after NAIC. d, Spearman correlation analysis between radiographic response and percent of residual viable tumor (%RVT). NAIC, neoadjuvant immunochemotherapy; MPR, major pathologic response; NMPR, non-major pathologic response; CPS, combined positive score; CR, complete response; PR, partial response; SD, stable disease; PD, progressive disease. Source data are presented as a Source Data file.

Figure 3. Tertiary lymphoid structures and TME features in MPR and NMMP patients before and after NAIC. a, H&E staining, TLSs were marked by blue circles. b and c, Multiplex immunofluorescence (mIF) staining for TLSs and TLS-related markers. d, e, f, Quantification of TLSs, CD8⁺ T cells, and CD20⁺ B cells based on mIF staining analysis in MPR (n = 20) and NMMP (n = 9) patients before and after

NAIC. g, Representative mIF images showing dendritic cells (CD11c⁺), natural killer cells (CD56⁺), and M2 type tumor associated macrophages (CD68⁺CD163⁺). h-j, Quantification of CD56⁺, CD11c⁺, and CD68⁺CD163⁺ cells based on mIF staining analysis in MPR and NMPR patients before and after NAIC. k, Heatmaps illustrating the proportions of the above immune cells. TLSs, tertiary lymphoid structures; NAIC, neoadjuvant immunochemotherapy; MPR, major pathologic response; NMPR, non-major pathologic response. The dot represents an individual data point. *ns*, not significant; *** $P < 0.001$, **** $P < 0.0001$. Source data are presented as a Source Data file.

7. In addition, the post treatment neoadjuvant staging seems to be missing. In table 1, we see the pre-neoadjuvant stage but not the post.

Response: Thank you for your comment. In our initially submitted version, the post treatment neoadjuvant staging was displayed in the **Supplementary Table 5**. In the revised manuscript, we have rearranged “**Supplementary Table 5**” as “**Supplementary Table 1**” and added pre- and post-neoadjuvant stage. In this revised table, the column titled “Pathological stage” refers to “post-neoadjuvant stage”.

Supplementary Table 1. Clinical to pathological downstage after neoadjuvant immunochemotherapy

Patient number	Clinical stage (Pre-NAIC)	Pathological stage (Post-NAIC)	Residual viable tumor (%RVT)		Radiographic response (%)	Clinical to pathological downstage	
			Primary tumor	Lymph node		Primary tumor	Lymph node
P1	cT3N0M0 (stage III)	ypT1N0M0 (stage I)	MPR (%RVT < 10%)	-	PR (-67.2%)	Yes	-
P2	cT4aN2cM0 (stage IVA)	ypT4aN2bM0 (stage IVA)	NMPR (%RVT = 90%)	NMPR (%RVT = 80%)	PR (-37.5%)	No	yes
P3	cT4aN2cM0 (stage IVA)	ypT1N1M0 (stage III)	MPR (%RVT < 10%)	MPR (%RVT < 10%)	PR (-52.60%)	Yes	Yes

P4	cT3N2cM0 (stage IVA)	ypT2N2cM0 (stage IVA)	10%) NMPR (%RVT = 40%)	10%) NMPR (%RVT = 50%)	SD (-29.83%)	Yes	No
P5	cT3N1M0 (stage III)	ypT1N1M0 (stage III)	30%) NMPR (%RVT = 30%)	60%) NMPR (%RVT = 60%)	PR (-64.50%)	Yes	No
P6	cT3N2bM0 (stage IVA)	ypT3N2bM0 (stage IVA)	70%) NMPR (%RVT = 70%)	80%) NMPR (%RVT = 80%)	SD (-28.9%)	No	No
P7	cT3N1M0 (stage III)	ypT0N0M0 (-)	0) pCR (%RVT = 0)	0) pCR (%RVT = 0)	PR (-70.57%)	Yes	Yes
P8	cT3N1M0 (stage III)	ypT3N1M0 (stage III)	80%) NMPR (%RVT = 80%)	90%) NMPR (%RVT = 90%)	SD (-9.96%)	No	No
P9	cT4aN2aM0 (stage IVA)	ypT1N2aM0 (stage IVA)	40%) NMPR (%RVT = 40%)	70%) NMPR (%RVT = 70%)	PR (-51.72%)	Yes	No
P10	cT4aN3aM0 (Stage IVB)	ypT1N0M0 (stage I)	10%) MPR (%RVT < 10%)	0) pCR (%RVT = 0)	PR (-50.78%)	Yes	Yes
P11	cT4aN2aM0 (stage IVA)	ypT1N0M0 (stage IVA)	10%) MPR (%RVT < 10%)	0) pCR (%RVT = 0)	PR (-57.55%)	Yes	Yes
P12	cT2N2aM0 (stage IVA)	ypT1N2aM0 (stage IVA)	40%) NMPR (%RVT = 40%)	60%) NMPR (%RVT = 60%)	SD (-21.16%)	Yes	No
P13	cT3N2aM0 (stage IVA)	ypT1N2aM0 (stage IVA)	10%) MPR (%RVT < 10%)	10%) MPR (%RVT < 10%)	PR (-54.83%)	Yes	No
P14	cT3N2aM0 (stage IVA)	ypT3N2aM0 (stage IVA)	95%) NMPR (%RVT = 95%)	80%) NMPR (%RVT = 80%)	PR (-35.11%)	No	No
P15	cT4aN0M0 (stage IVA)	ypT0N0M0 (-)	0) pCR (%RVT = 0)	- - (%RVT = 0)	PR (-43.43%)	Yes	-
P16	cT3N3bM0 (stage IVB)	ypT0N1M0 (stage III)	0) pCR (%RVT = 0)	10%) MPR (%RVT < 10%)	PR (-53.96%)	Yes	Yes
P17	cT4aN0M0 (stage IVA)	ypT4aN0M0 (stage IVA)	30%) NMPR (%RVT = 30%)	- - (%RVT = 30%)	SD (-4.24%)	No	-
P18	cT3N3bM0	ypT1N3bM0	MPR	MPR	PR (-40.43%)	Yes	No

	(stage IVB)	(stage IVB)	(%RVT < 10%)	(%RVT < 10%)			
P19	cT3N2bM0 (stage IVA)	ypT0N2bM0 (stage IVA)	pCR (%RVT = 0)	MPR (%RVT < 10%)	PR (-61.60%)	Yes	No
P20	cT3N1M0 (stage III)	ypT0N0M0 (-)	pCR (%RVT = 0)	pCR (%RVT = 0)	PR (-66.67%)	Yes	Yes
P21	cT3N1M0 (stage III)	ypT0N0M0 (-)	pCR (%RVT = 0)	pCR (%RVT = 0)	PR (-61.94%)	Yes	Yes
P22	cT3N2aM0 (stage IVA)	ypT0N0M0 (-)	pCR (%RVT = 0)	pCR (%RVT = 0)	CR (-100%)	Yes	Yes
P23	cT4aN2bM0 (stage IVA)	ypT0N0M0 (-)	pCR (%RVT = 0)	pCR (%RVT = 0)	CR (-100%)	Yes	Yes
P24	cT3N2aM0 (stage IVA)	ypT0N0M0 (-)	pCR (%RVT = 0)	pCR (%RVT = 0)	CR (-100%)	Yes	Yes
P25	cT3N2aM0 (stage IVA)	ypT0N0M0 (-)	pCR (%RVT = 0)	pCR (%RVT = 0)	CR (-100%)	Yes	Yes
P26	cT3N0M0 (stage III)	ypT0N0M0 (-)	pCR (%RVT = 0)	-	CR (-100%)	Yes	-
P27	cT4aN2cM0 (stage IVA)	ypT0N1M0 (stage III)	pCR (%RVT = 0)	MPR (%RVT < 10%)	CR (-100%)	Yes	Yes
P28	cT4aN2cM0 (stage IVA)	ypT1N1M0 (stage III)	MPR (%RVT < 10%)	MPR (%RVT < 10%)	PR (-82.56%)	Yes	Yes
P29	cT3N1M0 (stage III)	ypT1N1M0 (stage III)	MPR (%RVT < 10%)	MPR (%RVT < 10%)	CR (-100%)	Yes	No
P30	cT4bN1M0 (stage IVB)	-	-	-	-	-	-
P31	cT4bN2aM0 (stage IVB)	-	-	-	-	-	-

NAIC, neoadjuvant immunochemotherapy; MPR, major pathologic response; NMPR, non-major pathologic response; CPS, combined positive score; CR, complete response; PR, partial response; SD, stable disease; PD, progressive disease. %RVT, the percent of residual viable tumor. Note, P30 and P31 refused surgical resection and

imaging examination for personal reasons after receiving 2 cycles of NAIC, thus the pathological stage, %RVT and radiographic response of these two patients were evaluable.

8. Also, the primary tumor site of tongue should separately state oral or base of tongue primary tumour. They also did not seem to test for human papillomavirus, which is essential in the oropharynx. Lastly they list 31 patients in table 1 but indicate treating 29 patients.

Response: Thank you for your constructive comment. In this study, we enrolled patients with locally advanced oral squamous cell carcinoma, with oral tongue (but not base of tongue) included as one of the subtypes enrolled. According to your comments and to avoid confusion, we have now replaced "tongue" with "oral tongue" throughout the manuscript and tables. Human papillomavirus has been identified closely correlated with oropharynx carcinoma^[20-22]. However, human papillomavirus is uncommonly detected in oral cavity squamous cell carcinoma, and the relationship between human papillomavirus and oral tongue still remains unclear^[23-25]. Therefore, we did not perform detection of human papillomavirus in our study.

In this study, a total of 31 patients received NAIC. All 31 patients who received NAIC were included in table 1. Among 31 treated patients, 2 refused surgical resection and imaging examination for personal reasons after receiving 2 cycles of NAIC. Therefore, 29 patients completed the NAIC treatment and subsequently underwent radical surgery. To avoid any confusion, we have added the TNM stages of the 2 patients (P30, P31) who did not undergo surgery after NAIC in Supplementary Table 1 in the revised manuscript.

Supplementary Table 1. Clinical to pathological downstage after neoadjuvant immunochemotherapy

Patient number	Clinical stage (Pre-NAIC)	Pathological stage (Post-NAIC)	Residual viable tumor (%RVT)		Radiographic response (%)	Clinical to pathological downstage	
			Primary tumor	Lymph node		Primary tumor	Lymph node
P1	cT3N0M0 (stage III)	ypT1N0M0 (stage I)	MPR (%RVT < 10%)	-	PR (-67.2%)	Yes	-

P2	cT4aN2cM0 (stage IVA)	ypT4aN2bM0 (stage IVA)	NMPR (%RVT = 90%)	NMPR (%RVT = 80%)	PR (-37.5%)	No	Yes
P3	cT4aN2cM0 (stage IVA)	ypT1N1M0 (stage III)	MPR (%RVT < 10%)	MPR (%RVT < 10%)	PR (-52.60%)	Yes	Yes
P4	cT3N2cM0 (stage IVA)	ypT2N2cM0 (stage IVA)	NMPR (%RVT = 40%)	NMPR (%RVT = 50%)	SD (-29.83%)	Yes	No
P5	cT3N1M0 (stage III)	ypT1N1M0 (stage III)	NMPR (%RVT = 30%)	NMPR (%RVT = 60%)	PR (-64.50%)	Yes	No
P6	cT3N2bM0 (stage IVA)	ypT3N2bM0 (stage IVA)	NMPR (%RVT = 70%)	NMPR (%RVT = 80%)	SD (-28.9%)	No	No
P7	cT3N1M0 (stage III)	ypT0N0M0 (-)	pCR (%RVT = 0)	pCR (%RVT = 0)	PR (-70.57%)	Yes	Yes
P8	cT3N1M0 (stage III)	ypT3N1M0 (stage III)	NMPR (%RVT = 80%)	NMPR (%RVT = 90)	SD (-9.96%)	No	No
P9	cT4aN2aM0 (stage IVA)	ypT1N2aM0 (stage IVA)	NMPR (%RVT = 40%)	NMPR (%RVT = 70%)	PR (-51.72%)	Yes	No
P10	cT4aN3aM0 (Stage IVB)	ypT1N0M0 (stage I)	MPR (%RVT < 10%)	pCR (%RVT = 0)	PR (-50.78%)	Yes	Yes
P11	cT4aN2aM0 (stage IVA)	ypT1N0M0 (stage IVA)	MPR (%RVT < 10%)	pCR (%RVT = 0)	PR (-57.55%)	Yes	Yes
P12	cT2N2aM0 (stage IVA)	ypT1N2aM0 (stage IVA)	NMPR (%RVT = 40%)	NMPR (%RVT = 60%)	SD (-21.16%)	Yes	No
P13	cT3N2aM0 (stage IVA)	ypT1N2aM0 (stage IVA)	MPR (%RVT < 10%)	MPR (%RVT < 10%)	PR (-54.83%)	Yes	No
P14	cT3N2aM0 (stage IVA)	ypT3N2aM0 (stage IVA)	NMPR (%RVT = 95%)	NMPR (%RVT = 80%)	PR (-35.11%)	No	No
P15	cT4aN0M0 (stage IVA)	ypT0N0M0 (-)	pCR (%RVT = 0)	-	PR (-43.43%)	Yes	-
P16	cT3N3bM0 (stage IVB)	ypT0N1M0 (stage III)	pCR (%RVT = 0)	MPR (%RVT < 0)	PR (-53.96%)	Yes	Yes

P17	cT4aN0M0 (stage IVA)	ypT4aN0M0 (stage IVA)	NMPR (%RVT = 0)	- 10%)	SD (-4.24%)	No	-
P18	cT3N3bM0 (stage IVB)	ypT1N3bM0 (stage IVB)	MPR (%RVT < 10%)	MPR (%RVT < 10%)	PR (-40.43%)	Yes	No
P19	cT3N2bM0 (stage IVA)	ypT0N2bM0 (stage IVA)	pCR (%RVT = 0)	MPR (%RVT < 10%)	PR (-61.60%)	Yes	No
P20	cT3N1M0 (stage III)	ypT0N0M0 (-)	pCR (%RVT = 0)	pCR (%RVT = 0)	PR (-66.67%)	Yes	Yes
P21	cT3N1M0 (stage III)	ypT0N0M0 (-)	pCR (%RVT = 0)	pCR (%RVT = 0)	PR (-61.94%)	Yes	Yes
P22	cT3N2aM0 (stage IVA)	ypT0N0M0 (-)	pCR (%RVT = 0)	pCR (%RVT = 0)	CR (-100%)	Yes	Yes
P23	cT4aN2bM0 (stage IVA)	ypT0N0M0 (-)	pCR (%RVT = 0)	pCR (%RVT = 0)	CR (-100%)	Yes	Yes
P24	cT3N2aM0 (stage IVA)	ypT0N0M0 (-)	pCR (%RVT = 0)	pCR (%RVT = 0)	CR (-100%)	Yes	Yes
P25	cT3N2aM0 (stage IVA)	ypT0N0M0 (-)	pCR (%RVT = 0)	pCR (%RVT = 0)	CR (-100%)	Yes	Yes
P26	cT3N0M0 (stage III)	ypT0N0M0 (-)	pCR (%RVT = 0)	-	CR (-100%)	Yes	-
P27	cT4aN2cM0 (stage IVA)	ypT0N1M0 (stage III)	pCR (%RVT = 0)	MPR (%RVT < 10%)	CR (-100%)	Yes	Yes
P28	cT4aN2cM0 (stage IVA)	ypT1N1M0 (stage III)	MPR (%RVT < 10%)	MPR (%RVT < 10%)	PR (-82.56%)	Yes	Yes
P29	cT3N1M0 (stage III)	ypT1N1M0 (stage III)	MPR (%RVT < 10%)	MPR (%RVT < 10%)	CR (-100%)	Yes	No
P30	cT4bN1M0 (stage IVB)	-	-	-	-	-	-
P31	cT4bN2aM0 (stage IVB)	-	-	-	-	-	-

NAIC, neoadjuvant immunochemotherapy; MPR, major pathologic response; NMPR, non-major pathologic response; CPS, combined positive score; CR, complete response; PR, partial response; SD, stable disease; PD, progressive disease. %RVT, the percent of residual viable tumor.

Note, P30 and P31 refused surgical resection and imaging examination for personal reasons after receiving 2 cycles of NAIC, thus the pathological stage, %RVT and radiographic response of these two patients were evaluable.

References

- [20] Chaturvedi AK. et al. Human Papillomavirus and Rising Oropharyngeal Cancer Incidence in the United States. *J. Clin. Oncol.* **41**, 3081-3088 (2023).
- [21] Damgacioglu H. et al. Oropharyngeal Cancer Incidence and Mortality Trends in All 50 States in the US, 2001-2017. *JAMA. Otolaryngol. Head. Neck. Surg.* **148**, 155-165 (2022).
- [22] Kang JJ. et al. Consensuses, controversies, and future directions in treatment deintensification for human papillomavirus-associated oropharyngeal cancer. *CA. Cancer. J. Clin.* **73**, 164-197 (2023).
- [23] Li H. et al. Association of human papillomavirus status at head and neck carcinoma subsites with overall survival. *JAMA. Otolaryngol.-- Head. Neck. Surg.* **144**,519-525 (2018).
- [24] Zafereo ME. et al. Squamous cell carcinoma of the oral cavity often overexpresses p16 but is rarely driven by human papillomavirus. *Oral. Oncol.* **56**, 47-53 (2016).
- [25] Sahovaler A. et al. Survival outcomes in human papillomavirus-associated nonoropharyngeal squamous cell carcinomas: a systematic review and meta-analysis. *JAMA. Otolaryngol.-- Head. Neck. Surg.* **146**, 1158-1166 (2020).

9. *A consort diagram with patient allocation, and how patients are not listed in the response and efficacy should be stated.*

Response: Thank you for your advice about the consort diagram. In the initial submission, the consort diagram was displayed in the “Supplementary Materials”. According to your advice, we have moved this consort diagram to the main manuscript, as shown in **Figure 1** in the revised manuscript. We have also added the patient numbers for efficacy evaluation and safety evaluation in the consort diagram (**revised version**).

Figure 1. Study design. a, Trial schema. Eligible patients received 2 cycles of NAIC with camrelizumab plus nab-paclitaxel and cisplatin (every 3 weeks). Surgery was performed within 2-4 weeks after NAIC. Radiotherapy was performed within 6 weeks after surgery. Maintenance immunotherapy with 6 cycles of camrelizumab (every 3 weeks) was started simultaneously with radiotherapy. b, Patient flowchart. c, Treatment and follow-up status for each patient. NAIC, neoadjuvant immunochemotherapy; MPR, major pathologic response; NMPR, non-major pathologic response. Source data are presented as a Source Data file.

Note, P30 and P31 refused surgical resection and imaging examination for personal reasons after receiving 2 cycles of NAIC. The pathological stage, %RVT and radiographic response of these two patients were evaluable.

Reviewer #3 (Remarks to the Author):

In this clinical and translational manuscript by Xiang et al, the investigators conduct a phase 2 Neo-adjuvant immune-chemo (NAIC) trial on 30+ patients with HNSCC and report their clinical data as well a number of well-executed translational parameters to identify novel biomarkers of response. The clinical trial is an important contribution to the literature (although I never like overstatements such as targets study etc etc), and it appears likely that the field is moving towards this approach in advanced head and neck cancer. These studies are not easy to conduct and my congratulations to the authors for completing the clinical trial in such a short period of time with remarkable tolerance and dropout rates. I have a number of queries and issues that I feel can be addressed through a revised version of the manuscript and the are divided into the clinical and translational aspects of the study:

1. Clinical:

- a. I did not see a proper consort diagram of the trial, can this be included in the main manuscript (including number of patients screened etc)**
- b. although it was suggested that there was a stage reduction, was this limited to T or N stage? this was not shown**
- c. for pathological assessment, were the pathologists blinded to each others review of residual tumor etc**
- d. there were 2 surgical drop outs- any idea as to their response to NAIC?**
- e. for the Radiologic correlation with pathological responses, was this based on CT/MRI?**
- f. What was the Median time from diagnosis to surgery and did this compared to usual practice in their centre?**

2. Exploratory:

- a. The figures showed the genomic data first but it was only discussed later. This**

somewhat disrupts the readers flow and I wonder if it can be re-arranged. Also it would be useful to show the TMB data in the co-mutation plot for easier reading.

b. I find it odd that the investigators only used 1 radiomic feature. how was this one feature selected? it seems an odd plugin and somewhat non-intuitive to a paper that was solely about genomics and transcriptomics to add this bit? either expand on this or I am ok for this section to be removed completely as it is deceptively focussed without any basis.

c. Minor typo in the label of figure 3A which should read paracancerous or paracarcinoma but perhaps call it near adjacent 'normal'?

d. scRNAseq data suggests that overall higher lymphocyte infiltrates = better response..can this be seen in simple h and e staining?

e. the CD4_tfh population seems exceedingly small, less than 5% of the total population? is this correct? can this be identified through other techniques such as the mIF used? similarly, can mIF be used to look at the proximities of the cd4-tfh population with the TLS structures?

f. This is my major issues with this paper; the whole idea of using monocle for CD4 lineage discussion is completely flawed as these cell types DO NOT transition to one another. CD4 maturation occurs in the thymus and Tregs do not become tef's in the tumors. This entire analyses and presentation needs to be re-examined! The use of monocle in B cells and CD8 is logical as these cell states/lineage transitions happen in the tumor but CD4 does not!

g. the authors have performed some interesting scTCRseq but I do not see much in the way of the post-treatment data being shown. the claim of 6B of being post is unclear as there are no label on the axes. What are you showing? and 6i shows just 3 tumors or 3 clones where there is expansion? what about the other data/tumors? this selective display of data is misleading.

Discussion:

1. there have been larger studies performed on just ICI alone with pretty impressive response rates and these should not be excluded just to make the

authors response rates look better. the references appear to be there but discussion omits these.

2. the concept of TLS being increased after chemo or ICI is not novel. the authors should discuss the existing data on this, and perhaps offer a view as to whether this is a cause or merely consequence of tumor regression.

Response to reviewer #3:

Dear reviewer,

Thank you very much for giving us the opportunity to revise our manuscript. All of your comments are valuable and helpful in revising and improving our paper, as well as providing important guidance for our research. We have studied comments carefully and have made correction which we hope meet with approval. Please find the following detailed responses to your comments and suggestions.

1. Clinical:

a. I did not see a proper consort diagram of the trial, can this be included in the main manuscript (including number of patients screened etc)

Response: Thank you for your comment. In the initial submission, the consort diagram was displayed in the “Supplementary Materials”. According to your advice, we have moved this consort diagram to the main manuscript (**Figure 1**). A total of 35 patients were screened. Of these, two patients refused tumor biopsy, and two patients withdrew consent before starting NAIC. Thus, a total of 31 patients received NAIC. Among the 31 patients treated with NAIC, two refused surgical resection and imaging examination due to personal reasons. Consequently, 29 patients underwent radical surgery. All these 29 patients subsequently received postoperative radiotherapy or chemoradiotherapy and immunotherapy as maintenance therapy.

Figure 1. Study design. a, Trial schema. Eligible patients received 2 cycles of NAIC with camrelizumab plus nab-paclitaxel and cisplatin (every 3 weeks). Surgery was performed within 2-4 weeks after NAIC. Radiotherapy was performed within 6 weeks after surgery. Maintenance immunotherapy with 6 cycles of camrelizumab (every 3 weeks) was started simultaneously with radiotherapy. b, Patient flowchart. c, Treatment and follow-up status for each patient. NAIC, neoadjuvant immunochemotherapy; MPR, major pathologic response; NMPR, non-major pathologic response. Source data are presented as a Source Data file.

Note, P30 and P31 refused surgical resection and imaging examination for personal reasons after receiving 2 cycles of NAIC. The pathological stage, %RVT and radiographic response of these two patients were evaluable.

b. although it was suggested that there was a stage reduction, was this limited to T or N stage? this was not shown

Response: Thank you for your constructive comment. Based on your suggestion, we have added and divided the “Residual viable tumor (%RVT)” and “Clinical to pathological downstage” into 2 columns, respectively: “Primary tumor”, and “Lymph node” , as shown in Supplementary Table 1 in the revised manuscript.

Supplementary Table 1. Clinical to pathological downstage after neoadjuvant immunochemotherapy

Patient number	Clinical stage (Pre-NAIC)	Pathological stage (Post-NAIC)	Residual viable tumor (%RVT)		Radiographic response (%)	Clinical to pathological downstage	
			Primary tumor	Lymph node		Primary tumor	Lymph node
P1	cT3N0M0 (stage III)	ypT1N0M0 (stage I)	MPR (%RVT < 10%)	-	PR (-67.2%)	Yes	-
P2	cT4aN2cM0 (stage IVA)	ypT4aN2bM0 (stage IVA)	NMPR (%RVT = 90%)	NMPR (%RVT = 80%)	PR (-37.5%)	No	Yes
P3	cT4aN2cM0 (stage IVA)	ypT1N1M0 (stage III)	MPR (%RVT < 10%)	MPR (%RVT < 10%)	PR (-52.60%)	Yes	Yes
P4	cT3N2cM0 (stage IVA)	ypT2N2cM0 (stage IVA)	NMPR (%RVT = 40%)	NMPR (%RVT = 50%)	SD (-29.83%)	Yes	No
P5	cT3N1M0 (stage III)	ypT1N1M0 (stage III)	NMPR (%RVT = 30%)	NMPR (%RVT = 60%)	PR (-64.50%)	Yes	No
P6	cT3N2bM0 (stage IVA)	ypT3N2bM0 (stage IVA)	NMPR (%RVT = 70%)	NMPR (%RVT = 80%)	SD (-28.9%)	No	No
P7	cT3N1M0 (stage III)	ypT0N0M0 (-)	pCR (%RVT = 0)	pCR (%RVT = 0)	PR (-70.57%)	Yes	Yes
P8	cT3N1M0 (stage III)	ypT3N1M0 (stage III)	NMPR (%RVT =	NMPR (%RVT =	SD (-9.96%)	No	No

P9	cT4aN2aM0 (stage IVA)	ypT1N2aM0 (stage IVA)	NMPR (%RVT = 80%)	NMPR (%RVT = 90%)	PR (-51.72%)	Yes	No
P10	cT4aN3aM0 (Stage IVB)	ypT1N0M0 (stage I)	MPR (%RVT < 10%)	pCR (%RVT = 0)	PR (-50.78%)	Yes	Yes
P11	cT4aN2aM0 (stage IVA)	ypT1N0M0 (stage IVA)	MPR (%RVT < 10%)	pCR (%RVT = 0)	PR (-57.55%)	Yes	Yes
P12	cT2N2aM0 (stage IVA)	ypT1N2aM0 (stage IVA)	NMPR (%RVT = 40%)	NMPR (%RVT = 60%)	SD (-21.16%)	Yes	No
P13	cT3N2aM0 (stage IVA)	ypT1N2aM0 (stage IVA)	MPR (%RVT < 10%)	MPR (%RVT < 10%)	PR (-54.83%)	Yes	No
P14	cT3N2aM0 (stage IVA)	ypT3N2aM0 (stage IVA)	NMPR (%RVT = 95%)	NMPR (%RVT = 80%)	PR (-35.11%)	No	No
P15	cT4aN0M0 (stage IVA)	ypT0N0M0 (-)	pCR (%RVT = 0)	-	PR (-43.43%)	Yes	-
P16	cT3N3bM0 (stage IVB)	ypT0N1M0 (stage III)	pCR (%RVT = 0)	MPR (%RVT < 10%)	PR (-53.96%)	Yes	Yes
P17	cT4aN0M0 (stage IVA)	ypT4aN0M0 (stage IVA)	NMPR (%RVT = 30%)	-	SD (-4.24%)	No	-
P18	cT3N3bM0 (stage IVB)	ypT1N3bM0 (stage IVB)	MPR (%RVT < 10%)	MPR (%RVT < 10%)	PR (-40.43%)	Yes	No
P19	cT3N2bM0 (stage IVA)	ypT0N2bM0 (stage IVA)	pCR (%RVT = 0)	MPR (%RVT < 10%)	PR (-61.60%)	Yes	No
P20	cT3N1M0 (stage III)	ypT0N0M0 (-)	pCR (%RVT = 0)	pCR (%RVT = 0)	PR (-66.67%)	Yes	Yes
P21	cT3N1M0 (stage III)	ypT0N0M0 (-)	pCR (%RVT = 0)	pCR (%RVT = 0)	PR (-61.94%)	Yes	Yes
P22	cT3N2aM0 (stage IVA)	ypT0N0M0 (-)	pCR (%RVT = 0)	pCR (%RVT = 0)	CR (-100%)	Yes	Yes
P23	cT4aN2bM0	ypT0N0M0	pCR	pCR	CR (-100%)	Yes	Yes

	(stage IVA)	(-)	(%RVT = 0)	(%RVT = 0)			
P24	cT3N2aM0 (stage IVA)	ypT0N0M0 (-)	pCR (%RVT = 0)	pCR (%RVT = 0)	CR (-100%)	Yes	Yes
P25	cT3N2aM0 (stage IVA)	ypT0N0M0 (-)	pCR (%RVT = 0)	pCR (%RVT = 0)	CR (-100%)	Yes	Yes
P26	cT3N0M0 (stage III)	ypT0N0M0 (-)	pCR (%RVT = 0)	-	CR (-100%)	Yes	-
P27	cT4aN2cM0 (stage IVA)	ypT0N1M0 (stage III)	pCR (%RVT = 0)	MPR (%RVT < 10%)	CR (-100%)	Yes	Yes
P28	cT4aN2cM0 (stage IVA)	ypT1N1M0 (stage III)	MPR (%RVT < 10%)	MPR (%RVT < 10%)	PR (-82.56%)	Yes	Yes
P29	cT3N1M0 (stage III)	ypT1N1M0 (stage III)	MPR (%RVT < 10%)	MPR (%RVT < 10%)	CR (-100%)	Yes	No
P30	cT4bN1M0 (stage IVB)	-	-	-	-	-	-
P31	cT4bN2aM0 (stage IVB)	-	-	-	-	-	-

NAIC, neoadjuvant immunochemotherapy; MPR, major pathologic response; NMPR, non-major pathologic response; CPS, combined positive score; CR, complete response; PR, partial response; SD, stable disease; PD, progressive disease. %RVT, the percent of residual viable tumor. Note, P30 and P31 refused surgical resection and imaging examination for personal reasons after receiving 2 cycles of NAIC, thus the pathological stage, %RVT and radiographic response of these two patients could not be evaluate.

c. for pathological assessment, were the pathologists blinded to each others review of residual tumor etc

Response: Thank you for your comment. Yes, the percent residual viable tumor (%RVT) evaluation for pathological assessment was conducted by two experienced pathologists (each with over 10 years of experience) in a blinded manner. We have stated this content in the revised manuscript (Lines: 134-136).

d. there were 2 surgical drop outs- any idea as to their response to NAIC?

Response: Thank you for your comment. 2 patients refused surgery after NAIC for personal reason and declined further treatments or radiographic examinations in our center. Through follow-up, both patients reported significant symptoms relief, such as increased mouth opening and reduced pain, etc. They also said the tumor lesions significantly shrunk after NAIC. 1 patient (cT4bN2aM0) died 12.6 months after the initial treatment. The other patient (cT4bN1M0) underwent MRI reexamination at a local hospital after NAIC. With the patient's consent, we obtained the MRI images, which showed that the primary lesion in oral tongue had almost completely disappeared after NAIC (see below image). This patient then received radiotherapy at the local hospital, but suffered neck lymph node metastasis 14 months after the initial treatment. The post-NAIC MRI reexamination images are illustrated as follows.

MRI imaging pictures of the patient (cT4bN1M0, stage IVB) who did not undergo surgery before (A, at our center) and after (B, at the patient's local

hospital) NAIC.

e. for the Radiologic correlation with pathological responses, was this based on CT/MRI?

Response: Thank you for your comment. Radiographic response was evaluated based on the enhanced MRI scanning pre- and post-NAIC according to the Response Evaluation Criteria in Solid Tumors version 1.1 (*Eur J Cancer*, 2016, 62:132-137, doi: 10.1016/j.ejca.2016.03.081). Pathological response was assessed by calculating the residual viable tumor (%RVT) from the resected tumor on H&E staining slides according to a previously reported study (*Clin Cancer Res*, 2020, 26(3):545-551, doi: 10.1158/1078-0432.CCR-19-2379). **As shown in lines 147-149.**

f. What was the Median time from diagnosis to surgery and did this compared to usual practice in their centre?

Response: Thank you for your comment. In this study, the median time from diagnosis to surgery was 44 days (range: 36-52). We reviewed the usual practice in our centre, where the patient number was extremely large, and found that the usual time from diagnosis to surgery was 42 days (range: 28-69). Compared with the usual interval time, NAIC in this study did not delay the surgery.

2. Exploratory:

a. The figures showed the genomic data first but it was only discussed later. This somewhat disrupts the readers flow and I wonder if it can be re-arranged. Also it would be useful to show the TMB data in the co-mutation plot for easier reading.

Response: Thank you for your kind comment. Based on your suggestion, we have rearranged the figures, and added the TMB data in the co-mutation plot (**Supplementary Figure 4**) in the revised manuscript.

Supplementary Figure 4. Gene characteristics. a, Waterfall plot of WES in baseline primary tumor tissue. Each column represents a single patient, with gene mutation frequencies shown on the left and percentages displayed on the right. The bottom bars illustrate clinical and pathological characteristics. b, Comparison of percent of residual viable tumor (%RTV) in TP53(+) (n = 21) and TP53(-) (n = 8) populations. c, Comparison of tumor mutation burden in MPR (n = 20) and NMPR (n = 9) population. TMB, tumor mutation burden; MPR, major pathologic response; NMPR, none-major pathologic response; CPS, combined positive score; CR, complete response; PR, partial response; SD, stable disease; PD, progressive disease. The dot represents an individual data point. *ns*, not significant, * $P < 0.05$.

b. I find it odd that the investigators only used 1 radiomic feature. how was this one feature selected? it seems an odd plugin and somewhat non-intuitive to a paper that was solely about genomics and transcriptomics to add this bit? either expand on this or I am ok for this section to be removed completely as it is deceptively focussed without any basis.

Response: Thank you for your comment. Radiomic feature based on scillating-gradient spin-echo (OGSE) sequence in this study is a newly developed MRI scanning sequence, has been utilized in preclinical studies to distinguish cell populations within the tumor microenvironment by calculating mean cell diameters (*J Immunother Cancer*, 2023, 11(3):e006092, doi: 10.1136/jitc-2022-006092). Therefore, in this study, we performed OGSE scanning before and after NAIC to explore the potential new radiomic-based predictive markers under the NAIC setting. We agree with your point and have completely removed this section.

c. Minor typo in the label of figure 3A which should read paracancerous or paracarcinoma but perhaps call it near adjacent 'normal'?

Response: Thank you for your comment. According to your suggestion, we have used the “paracarcinoma” to replaced “normal” in all figures and the whole manuscript. For the consideration of the organizational need, the initially submitted **“Figure 3”** has been rearranged as **“Figure 4”** in the revised manuscript.

Figure 4. Single-cell RNA-seq analyses of tumor and paracarcinoma tissues. a, Study workflow. b, UMAP illustrations of all cell clusters. c, Representative marker genes for each cell cluster and cell distribution characteristics in paracarcinoma and tumor tissues. d, UMAP illustrations of the density of cell subtypes in MPR_pre-NAIC_Tumor and NMMPR_pre-NAIC_Tumor samples. e, Proportions of cell clusters among different groups. f, UMAP illustrations of changes in the density of all cell subtypes and cell numbers comparison in paired pre- and post-NAIC tumor tissues (P18, P20, P24). MPR, major pathologic response; NMMPR, non-major pathologic response. Source data are presented as a Source Data file.

d. scRNAseq data suggests that overall higher lymphocyte infiltrates = better response. can this be seen in simple h and e staining?

Response: Thank you for your comment. As shown in **Figure 3a** in the revised manuscript, tertiary lymphoid structure composed of infiltrated immune cells were observed on the H&E staining slides. Compared with NMPR patients, MPR patients exhibited increased number of TLS after NAIC on the H&E staining slides. However, the anti- and pro-tumor phenotypes of the immune cells cannot be directly distinguished on H&E staining slides. Therefore, we conducted multiple immunofluorescence staining (**Figure 3b-j**), and found that higher infiltration of CD8⁺ and CD20⁺ lymphocytes was associated with better pathological responses (**Supplementary Figure 3c,d**).

Figure 3. Tertiary lymphoid structures and TME features in MPR and NMMPR patients before and after NAIC. a, H&E staining, TLSs were marked by blue circles. b and c, Multiplex immunofluorescence (mIF) staining for TLSs and TLS-related markers. d, e, f, Quantification of TLSs, CD8⁺ T cells, and CD20⁺ B cells based on mIF staining analysis in MPR (n = 20) and NMMPR (n = 9) patients before and after

NAIC. g, Representative mIF images showing dendritic cells (CD11c⁺), natural killer cells (CD56⁺), and M2 type tumor associated macrophages (CD68⁺CD163⁺). h-j, Quantification of CD56⁺, CD11c⁺, and CD68⁺CD163⁺ cells based on mIF staining analysis in MPR and NMPR patients before and after NAIC. k, Heatmaps illustrating the proportions of the above immune cells. TLSs, tertiary lymphoid structures; NAIC, neoadjuvant immunochemotherapy; MPR, major pathologic response; NMPR, non-major pathologic response. The dot represents an individual data point. *ns*, not significant; *** $P < 0.001$, **** $P < 0.0001$. Source data are presented as a Source Data file.

Supplementary Figure 3. Correlation between different factors and %RVT. a, Correlation between PD-L1 (CPS) and %RVT. b, Correlation between TLS and %RVT after NAIC. c, Correlation between CD8⁺ T cells density and %RVT after NAIC. d, Correlation between CD20⁺ cells and %RVT after NAIC. %RVT, percent of residual viable tumor; TLS, tertiary lymphoid structures; NAIC, neoadjuvant immunochemotherapy; CPS, combined positive score.

e. the CD4_{tfh} population seems exceedingly small, less than 5% of the total population? is this correct? can this be identified through other techniques such as the mIF used? similarly, can mIF be used to look at the proximities of the cd4-tfh population with the TLS structures?

Response: Thank you for your valuable comments on the CD4_{tfh} population. To fully understand the ratio of CD4_{tfh} population in tumor microenvironment, we reviewed the related published studies and performed validation by mIF. Most studies reported the ratio of CXCL13⁺CD4⁺ cells, and we also calculated this ratio based on our data. The mean ratio of CXCL13⁺CD4⁺ cells for OSCC in our research, based on scRNA-seq analysis, was 23.35% (range: 6.36%-41.38%), and the ratio validated by mIF was 20.53% (range: 6.84%-34.93%). The ratios vary significantly across different tumors types: approximately 0-20% in triple-negative breast cancer (*Cancer Cell*, 2021, 39(12):1578-1593.e8, doi:10.1016/j.ccell.2021.09.010), 0.67-16.7% in ovarian cancer, (*JCI Insight*, 2022, 7(12):e157215, doi: 10.1172/jci.insight.157215), 7-55% in melanoma (*Cancer Cell*, 2022, 40(4): 393-409.e9, doi:10.1016/j.ccell.2022.03.006), and 3.84-9.08% in nasopharyngeal carcinoma (*J Immunother Cancer*, 9(7):e002101, doi: 10.1136/jitc-2020-002101), etc.

Meanwhile, we performed spatial analysis based on multiplex immunofluorescence staining slides to explore the co-localization of cd4-tfh and TLS. We observed that, in germinal center, CD4⁺CXCL13⁺ cells were obviously adjacent to CD20⁺CXCR5⁺ cells on multiplex immunofluorescence staining slides (**Fig. 8d**), G-cross function and Jaccard index analyses further found that CD4⁺CXCL13⁺ cells were most frequently co-localized with CD20⁺CXCR5⁺ cells in the TLS, whereas other CD4⁺CXCL13⁻ cells presented with a poor co-localization to CD20⁺CXCR5⁺ cells (**Fig. 8e-g**). These data suggest that interaction between CD4_{Tfh}_CXCL13 cells and B_{naïve} cells might be a potential mechanism for the immune response during NAIC (lines 417-432, revised version).

Figure 8. Cell-cell communication and spatial analysis. a, Ligand-receptor analysis based on Cellchat; expression heatmap of chemokine genes in T cell clusters (left) and B clusters (right), and interactions are connected by colored lines. b, Ligand-receptor analysis based on Cellchat; dot plots show the top predicted ligands in B cells (bottom) and their targets in T cells (left); the heatmap shows the potential interaction (middle). c, Mature phenotypes of tertiary lymphoid structures validated by mIF. The representative image was from patient P9. d, Spatial analysis based on mIF staining slide. The representative image was from patient P9. e and f, G-cross function analysis of spatial distributions of CD8⁺CXCR5⁻, CD4⁺CXCL13⁺, CD8⁺CXCR5⁺, CD4⁺CXCL13⁺, and CD20⁺CXCR5⁺ cells. g, Jaccard index analysis of CD8⁺CXCR5⁻, CD4⁺CXCL13⁺, CD8⁺CXCR5⁺, CD4⁺CXCL13⁺, and CD20⁺CXCR5⁺ cells. NAIC, neoadjuvant immunochemotherapy; MPR, major pathologic response; NMPR, non-major pathologic response.

f. This is my major issues with this paper; the whole idea of using monocle for CD4 lineage discussion is completely flawed as these cell types DO NOT transition to one another. CD4 maturation occurs in the thymus and Tregs do not become T_H1s in the tumors. This entire analyses and presentation needs to be re-examined! The use of monocle in B cells and CD8 is logical as these cell states/lineage transitions happen in the tumor but CD4 does not!

Response: Thank you for your constructive comments. We totally agree with your points. We have removed the contents related to the monocle for CD4 lineage analysis in tumor from the manuscript. Thanks again for your kind reminder. According to your comments, we have had a deeper understand of the CD4 maturation process.

g. the authors have performed some interesting scTCRseq but I do not see much in the way of the post-treatment data being shown. the claim of 6B of being post is unclear as there are no label on the axes. What are you showing? and 6i shows just 3 tumors or 3 clones where there is expansion? what about the other data/tumors? this selective display of data is misleading.

Response: Thank you for your constructive comments. It's our oversight that we did not label the axes in **Figure. 6b and 6i** in our initially submitted manuscript. The initially submitted **Figure. 6b** was actually the CD4 and CD8 TCR clonotype sharing condition across clusters after NAIC. We have added the label "Post-NAIC" on the axes. In addition, the initially submitted **Figure. 6i** illustrated the clone index changes in three paired tumor tissues (P18, P20, P24) before and after NAIC. Due to the highly heterogeneous TCR repertoire among different individuals, and to reduce bias caused by these individualized differences, we only included samples with paired sequencing data (n=3) before and after treatment in the TCR clone index analysis. Thanks again for your valuable comments.

Figure 6. Characterization of TCR repertoire before and after NAIC. a, Landscape of TCR expansion. b, TCR clonotype sharing in post-NAIC tumor tissue. c, The percentages of clonal T cells in Pre-NAIC_Paracarcinoma and Pre-NAIC_Tumor

samples. Top is Column chart, bottom is Pie chart. d and e, Projection of TCRs onto the UMAP embeddings of T cells in MPR_Pre-NAIC_Tumor and NMPR_Pre-NAIC_Tumor samples (d) and MPR_Pre-NAIC_ paracarcinoma and NMPR_Pre-NAIC_ paracarcinoma (e) samples. f, Projection of emergent TCR onto the UMAP embeddings of T cells in paired pre- and post-NAIC tumor tissues (P18, P20, P24). g, Column chart of emergent TCRs post NAIC in (f). h, Clone index changes in paired pre- and post-NAIC tumor tissues (P18, P20, P24).

Discussion:

1. there have been larger studies performed on just ICI alone with pretty impressive response rates and these should not be excluded just to make the authors response rates look better. the references appear to be there but discussion omits these.

Response: Thank you for your constructive comments. We have added the related discussion in the revised manuscript as follows: “Previous studies reported the following pathological responses in locally advanced HNSCC. For instance, the neoadjuvant mono/dual ICB (MPR rate: 5.9% - 35%; pCR rate: 0 - 10%), ICB combined with targeted therapy ± chemotherapy (MPR rate: 40% - 60%; pCR rate: 5% - 40%), ICB combined with radiotherapy ± chemotherapy (MPR rate: 43.0% - 60%; pCR rate: 20.0% - 57.0%; including p16 positive patients), and ICB combined with chemotherapy (MPR rate: 27.8% - 74.1%; pCR rate: 16.7% - 55.6%)^{[9-27]}” (**Lines: 474-480, revised version**).

2. the concept of TLS being increased after chemo or ICI is not novel. the authors should discuss the existing data on this, and perhaps offer a view as to whether this is a cause or merely consequence of tumor regression.

Response: Thank you for your constructive comments. We have added the discussion contents about TLS as follows: “Recent studies have identified TLS as intratumoral loci for initiating antitumor immune response^{[44, 50]}. TLS have been reported as potential predictive biomarkers in several solid tumors^{[51]}, including esophageal cancer^{[52]}, clear cell renal cell cancer^{[53]}, and non-small cell lung cancer^{[54]}.

Our study revealed that there was no significantly statistical difference of the density of TLS between MPR and NMPR tissues before NAIC, and an obvious enrichment of TLS was only observed in the MPR tissue after NAIC. Thus, the accumulation of TLS after NAIC might be closely related with the tumor regression. Therefore, TLS might be the cause of the tumor regression, which needs to be further exploration.” (Lines 515-524, revised version).

Reviewer #4 (Remarks to the Author):

Abstract

Should spell out pCR?

The results of exploratory analysis take a majority, more than reporting the primary and secondary outcomes. So, this manuscript mainly focuses on bio-radiomic analysis? If yes, may revise the manuscript title.

Follow-up of long-term survival outcomes is ongoing. How long is the study period for this Phase 2 trial?

Lines 140-141: the order of ORR and pCR are inconsistent with these two in the other sections.

Line 216: Simon two-stage design is a one-sided test rather than two-sided one.

Line 222: 10% drop-out was used. Any references to justify this choice?

Lines 229-230: There are so many exploratory endpoints. What's the significance level? Should consider False Discovery Rate?

Lines 231-232: Spearman correlation coefficient should be used for two continuous variables. The term "response" in "pathological response" may be misleading since pCT and ORR are categorical values. In the efficacy section and Figures, pathological response is continuous variable, e.g., RTV%. As such, the statistical section should be clearly defined.

Lines 258-259; What does "immature" mean? Is the study ongoing? How long is the study period?

Response to reviewer #4:

Dear reviewer,

Thank you very much for your careful review of our manuscript. we are very grateful to you for the time you have given in the appraisal of our paper and for the constructive feedback. After carefully reviewing the comments, we have modified the manuscript according to your advice. Please find the following detailed responses to your comments and suggestions.

1. Should spell out pCR?

Response: Thank you for your careful review. We have added the full spelling (pathological complete response) of “pCR” in the revised manuscript (**Line 156**).

2.The results of exploratory analysis take a majority, more than reporting the primary and secondary outcomes. So, this manuscript mainly focuses on bio-radiomic analysis? If yes, may revise the manuscript title.

Response: Thank you for your comment. The primary endpoints, and thus the main objectives of this study, were major pathologic response and safety of NAIC. On this basis, we conducted bioinformatics analyses to explore potential response biomarkers and the underlying mechanisms of this treatment. In the initial submitted manuscript, the title was “A phase II trial and bio-radiomic analysis of neoadjuvant immunotherapy in locally advanced oral squamous cell carcinoma”. As the radiomic analysis was not logically correlated with the main objectives of our study, therefore, **in the revised manuscript, we have more paid attention to the clinical and bioinformatics results, and removed the radiomic analysis section. The manuscript title has been revised to “Efficacy, safety and single-cell analysis of neoadjuvant immunochemotherapy in locally advanced oral squamous cell carcinoma: a phase II trial”.** Meanwhile, we further carried out the mechanism experiments *in vivo* to validate the exploratory results. If you find the revised title inappropriate, we are open to make further changes.

3. Follow-up of long-term survival outcomes is ongoing. How long is the study period for this Phase 2 trial?

Response: Thank you for your comment. This phase II clinical study was designed to last 5 years. Patient enrollment was planned for the first year, followed by patient treatment and survival follow-up during the subsequent years. And we have extended the data cut-off on November 30, 2024. And we have also added the current updated survival results in the revised manuscript. As of this data cutoff on November 30, 2024, the median follow-up duration for 31 treated patients was 18.6 months (range: 12.6-22.2). For all 31 treated patients, the 18-month OS rate was 96.77% (95%CI: 79.23%-99.54%) (**Supplementary Fig. 1**). For 29 patients who underwent surgery, the 18-month disease-free survival rate was 85.71% (95%CI: 53.95%-96.22%) (**Supplementary Fig. 2**). We have added these survival data in the results section (**Lines 281-285**) and supplemented materials (**Supplementary Fig. 1, Supplementary Fig. 2**). The long survival follow-up is still going on. The final survival data will be reported in the future.

Supplementary Figure 1. Kaplan-Meier analysis of overall survival for all 31 treated patients. Source data are presented as a Source Data file.

Supplementary Figure 2. Kaplan-Meier analysis of disease-free survival for 29 patients who received neoadjuvant immunochemotherapy and surgery. Source data are presented as a Source Data file.

4. Lines 140-141: the order of ORR and pCR are inconsistent with these two in the other sections.

Response: Thank you very much for your careful review. We have re-arranged the order of ORR and pCR (**Lines 156-157**) in the revised manuscript.

5. Line 216: Simon two-stage design is a one-sided test rather than two-sided one.

Response: Thank you for your kind comment. We have rechecked and confirmed that the sample size in the original submitted manuscript was calculated using a one-sided test based on the Simon two-stage design. The typing error, where “one-sided” was incorrectly written as “two-sided” has been corrected in **line 234** in the revised manuscript.

```
> ph2simon(0.277, 0.55, 0.05, 0.20)

Simon 2-stage Phase II design

Unacceptable response rate: 0.277
Desirable response rate: 0.55
Error rates: alpha = 0.05 ; beta = 0.2

      r1 n1  r  n  EN(p0)  PET(p0)  qLo  qHi
Minimax 3 11  9  21  14.63  0.6368  0.541  1.000
Admissible 2  8  9  22  13.45  0.6104  0.045  0.541
Optimal  3  9  11  28  13.17  0.7804  0.000  0.045

> 28*1.1
[1] 30.8
```

Codes used in the calculation of the study sample size.

6. Line 222: 10% drop-out was used. Any references to justify this choice?

Response: Thank you for your constructive comment. Referring to studies of the same type, the dropout rate is often set at 10-20% (Clin Cancer Res, 2022, 28(15):3268-3276.doi: 10.1158/1078-0432.CCR-22-0666; Clin Cancer Res, 2020, 26(19):5140-5152. doi: 10.1158/1078-0432.CCR-20-1695; Nat Commun, 2024,15(1):5251. doi: 10.1038/s41467-024-49121-3). Based on the actual situation of the multidisciplinary team management for oral cancer patients in our center, a dropout rate of 10% was set after a comprehensive evaluation of the implementation and difficulty of this clinical study.

7. Lines 229-230: There are so many exploratory endpoints. What's the significance level? Should consider False Discovery Rate?

Response: Thank you for your comment. As you pointed out, we cannot exclude the possibility of increasing false positive rates due to multiple comparisons for exploratory endpoints. Our exploratory endpoints, including bioinformatics findings, are preliminary and require validation in future larger-scale studies. These limitations have been acknowledged in the manuscript. In the phase II clinical trial, the sample size is calculated based on the primary endpoint, and the significance level for testing the primary endpoint was set at a one-sided α value of 0.05 with a power of 80%. It is common for secondary and exploratory endpoints not to perform significance level adjustments, as they are primarily intended for hypothesis generation, supporting primary outcomes, or identifying trends for future studies rather than drawing definitive conclusions.

8. Lines 231-232: Spearman correlation coefficient should be used for two continuous variables. The term "response" in "pathological response" may be misleading since pCT and ORR are categorical values. In the efficacy section and Figures, pathological response is continuous variable, e.g., RTV%. As such, the statistical section should be clearly defined.

Response: Thank you for your comments. We agree with your point, and have revised it as follows: “Spearman’s correlation coefficients were used to evaluate the correlations between potential markers and RTV%.” (lines 247-249, revised version).

9. Lines 258-259; What does “immature” mean? Is the study ongoing? How long is the study period?

Response: Thank you for your comments. In our initially submitted manuscript, the follow-up time was only 10 months, thus we did not report the survival results. This phase II clinical study was designed to last 5 years. Patient enrollment was planned for the first year, followed by patient treatment and survival follow-up during the subsequent years. We have extended the survival follow-up, and the corresponding results are included in the revised manuscript (Lines 281-285; Supplementary Fig. 1-2): As of the data cutoff on November 30, 2024, the median follow-up duration for 31 treated patients was 18.6 months (range: 12.6-22.2). For all 31 treated patients, the 18-month OS rate was 96.77% (95%CI: 79.23%-99.54%) (Supplementary Fig. 1). For 29 patients who underwent surgery, the 18-month disease-free survival rate was 85.71% (95%CI: 53.95%-96.22%) (Supplementary Fig. 2). The long survival follow-up is still ongoing. The final survival data will be reported in the future.

Supplementary Figure 1. Kaplan-Meier analysis of overall survival for all 31 treated patients. Source data are presented as a Source Data file.

Supplementary Figure 2. Kaplan-Meier analysis of disease-free survival for 29 patients who received neoadjuvant immunochemotherapy and surgery. Source data are presented as a Source Data file.

Table 1:

1. Any missing value for age and scanning? The label of age “range” but the right side uses \pm which indicates standard deviation in general; Per footnote, suggest using “age (years)” instead;

Response: Thank you for your comment. Based on your suggestion, we have modified the label to “age (years)” in **Table 1**. In the revised version, to maintain the overall logic and readability of the article, we have removed the part of OGSE radiomic analysis and plan to report it in future.

Characteristics	All patients n = 31
Age (years)	54 \pm 11
Gender	
Male	27 (87.10)
Female	4 (12.90)

Data are mean \pm standard deviation or n (%)

2. Two rows of male and female has the different format as other characteristics. Suggest adding a row of gender, and present male and female under the new row.

Response: Thank you for your comment. We have added the row “Gender” as

per your suggestion in **Table 1**.

Table 1. Baseline characteristics

Characteristics	All patients n = 31
Age (years)	54 ± 11
Gender	
Male	27 (87.10)
Female	4 (12.90)

Data are mean ± standard deviation or n (%).

POINT-BY-POINT RESPONSE TO THE REVIEWER'S COMMENTS

Reviewer #1 (Remarks to the Author):

Report on the manuscript titled “Efficacy, safety and single-cell analysis of neoadjuvant immunochemotherapy in locally advanced oral squamous cell carcinoma: a phase II trial” (NCOMMS-24-45477A):

The authors have revised the manuscript according to the comments, and the quality has been significantly improved. The authors have also answered my comments, and only a few suggestions need to be further improved. In addition to pathological evaluation for neoadjuvant immunochemotherapy, clinical stage before surgery by image evaluation should also be described for the efficacy evaluation of primary lesions and lymph node lesions. For the two patients refusing surgery, subsequent treatment and outcomes should be described in the section of Results. There are too many pictures in the body of revision, and the number of pictures needs to be reduced according to the requirements of the journal.

Response to reviewer #1:

Dear reviewer,

Thank you for your positive comments on our revised manuscript. We appreciate your thoughtful suggestions and further made relative changes. Our responses are shown below.

1. In addition to pathological evaluation for neoadjuvant immunochemotherapy, clinical stage before surgery by image evaluation should also be described for the efficacy evaluation of primary lesions and lymph node lesions.

Response: Thank you for your valuable suggestion, we have added the clinical stage by image evaluation before surgery (after NAIC, highlighted in blue) for both primary lesions and lymph node lesions in the revised **Supplementary Table 1**.

Supplementary Table 1. Clinical to pathological downstage after neoadjuvant immunochemotherapy

Patient number	Clinical stage (Pre-NAIC)	Clinical stage (Post-NAIC)	Clinical downstage	
			Primary tumor	Lymph node
P1	cT3N0M0 (stage III)	cT1N0M0 (stage I)	Yes	-
P2	cT4aN2cM0 (stage IVA)	cT4aN2cM0 (stage IVA)	No	No
P3	cT4aN2cM0 (stage IVA)	cT1N1M0 (stage III)	Yes	Yes
P4	cT3N2cM0 (stage IVA)	cT3N2cM0 (stage IVA)	No	No
P5	cT3N1M0 (stage III)	cT1N1M0 (stage III)	Yes	No
P6	cT3N2bM0 (stage IVA)	cT3N2bM0 (stage IVA)	No	No
P7	cT3N1M0 (stage III)	cT1N0M0 (stage I)	Yes	Yes
P8	cT3N1M0 (stage III)	cT3N1M0 (stage III)	No	No
P9	cT4aN2aM0 (stage IVA)	cT2N2aM0 (stage IVA)	Yes	No
P10	cT4aN3aM0 (Stage IVB)	cT2N1M0 (Stage III)	Yes	Yes
P11	cT4aN2aM0 (stage IVA)	cT2N1M0 (stage III)	Yes	Yes
P12	cT2N2aM0 (stage IVA)	cT2N2aM0 (stage IVA)	No	No
P13	cT3N2aM0 (stage IVA)	cT1N1M0 (stage III)	Yes	Yes

P14	cT3N2aM0 (stage IVA)	cT3N2aM0 (stage IVA)	No	No
P15	cT4aN0M0 (stage IVA)	cT2N0M0 (stage II)	Yes	-
P16	cT3N3bM0 (stage IVB)	cT1N2aM0 (stage IVA)	Yes	Yes
P17	cT4aN0M0 (stage IVA)	cT4aN0M0 (stage IVA)	No	-
P18	cT3N3bM0 (stage IVB)	cT1N3bM0 (stage IVB)	Yes	No
P19	cT3N2bM0 (stage IVA)	cT1N2bM0 (stage IVA)	Yes	No
P20	cT3N1M0 (stage III)	cT1N0M0 (stage I)	Yes	Yes
P21	cT3N1M0 (stage III)	cT1N0M0 (stage I)	Yes	Yes
P22	cT3N2aM0 (stage IVA)	T0N0M0 (-)	Yes	Yes
P23	cT4aN2bM0 (stage IVA)	T0N0M0 (-)	Yes	Yes
P24	cT3N2aM0 (stage IVA)	T0N0M0 (-)	Yes	Yes
P25	cT3N2aM0 (stage IVA)	T0N0M0 (-)	Yes	Yes
P26	cT3N0M0 (stage III)	T0N0M0 (-)	Yes	-
P27	cT4aN2cM0 (stage IVA)	T0N0M0 (-)	Yes	Yes
P28	cT4aN2cM0 (stage IVA)	T1N0M0	Yes	Yes

		(stage I)		
P29	cT3N1M0 (stage III)	T0N0M0 (-)	Yes	Yes
P30	cT4bN1M0 (stage IVB)	-	-	-
P31	cT4bN2aM0 (stage IVB)	-	-	-

NAIC, neoadjuvant immunochemotherapy; MPR, major pathologic response; NMPR, non-major pathologic response; CPS, combined positive score; CR, complete response; PR, partial response; SD, stable disease; PD, progressive disease. %RVT, the percent of residual viable tumor. Note, P30 and P31 refused surgical resection and imaging examination for personal reasons after receiving 2 cycles of NAIC, thus the pathological stage, %RVT and radiographic response of these two patients were unevaluable.

2. *For the two patients refusing surgery, subsequent treatment and outcomes should be described in the section of Results.*

Response: Thank you for your constructive advice. We have added the subsequent treatment and outcomes of the two patients refusing surgery after NAIC in the results section of the manuscript (**Lines 121-126**) as follows: “For the two patients who refused surgery after NAIC, one (cT4bN2aM0) did not receive any subsequent treatment and died 12.6 months after the initial study treatment. The other patient (cT4bN1M0) underwent MRI reexamination at a local hospital after NAIC, suggesting a complete response. The patient then received radiotherapy at the local hospital but suffered neck lymph node metastasis 14 months after the initial study treatment.”

3. *There are too many pictures in the body of revision, and the number of pictures needs to be reduced according to the requirements of the journal.*

Response: Thank you for your comment. According to your advice, we moved the Fig 5k,l in the last version of manuscript to Supplementary Information file as Supplementary Fig. 6a,b in this revised version. Meanwhile, there are 9 figures and 1 table in the main body of the manuscript, which meets the requirements that a total of

10 display items (Figures or Tables) in the main article are accommodated according to the “article structure guidance” of the journal. If the number of figures needs adjust, we are open to make further changes. We thank again for your careful review.

Reviewer #3 (Remarks to the Author):

Thank you for your response. I believe all my queries have been responded to satisfactorily. Congratulations on the study.

Response: Thank you for your patient review and positive feedback.

Reviewer #3 (Remarks on code availability):

Not qualified to assess

Reviewer #4 (Remarks to the Author):

Abstract

1. Lines 37-38: “The primary endpoint was met,” is confusing. “meet some criteria” or what? The primary endpoint of MPR is a proportion and thus has nothing to “meet”. Or does it refer to reject the null hypothesis?
2. Line 42: “rates” is inappropriate when time-to-event analysis were used to calculate survival probabilities at some time points. If my understanding is correct, they should be “overall survival and disease-free survival probabilities at 18-month were...”

Outcomes

1. Lines 155: One (MPR) or two primary endpoints (MPR and safety)?
2. Lines 158-160: the definitions of DFS and OS are inaccurate. How about the censored patients?
3. Lines 156-157: pCR rate is before ORR. However, ORR was reported before pCR in the Efficacy sections, e.g., Lines 270.
4. Line 234: Is Simon two-stage optimal and MiniMax design?
5. Line 240: 10% drop-out was used. The references to justify this choice should be added.
6. Line 240: Suggest adding a sentence-“The total sample size is up to 28 patients in this two-stage Phase 2 study”.
7. Line 241: “If more than 11 patients achieved MPR,” should be among 28 patients eligible for the MPR evaluation.
8. Line 242: Same comment as 1.
9. Line 247: ORR is a proportion and “was described as frequency and percentage” does not make sense.
10. Lines 250, 283-284: “rates” is inappropriate when time-to-event analysis is used.
11. Lines 260-261: 28 patients were proposed in Simon two-stage design. However, 29 patients were included in the analysis. Even if more patients were preferred in a statistical analysis, “If more than 11 patients achieved MPR” as a criterion cannot be used in this Simon two-stage design with 29 patients since ≥ 11 are for 28 patients

rather than 29 patients.

Table 1:

1. Any missing value for age? Only mean and std were shown in the first row.

Response to reviewer #4:

Dear reviewer,

We extend our deepest gratitude for your meticulous evaluation and invaluable insights provided during the peer-review process. Your expertise in assessing our work and identifying areas for refinement has profoundly strengthened the rigor and clarity of this research. We have systematically addressed all observations through comprehensive revisions. We consider it a privilege to have benefited from your discerning perspective.

Abstract:

1. Lines 37-38: "The primary endpoint was met," is confusing. "meet some criteria" or what? The primary endpoint of MPR is a proportion and thus has nothing to "meet". Or does it refer to reject the null hypothesis?

Response: Thank you for your careful review. we have deleted "The primary endpoint was met", and revised the statement as "For primary endpoint, the major pathological response (MPR) rate was 69.0% (95% confidence interval (CI): 49.2%-84.7%). The treatment was well-tolerated, with only 2 patients (6.45%) having grade 3 or 4 treatment-related adverse events during neoadjuvant treatment. " in the revised manuscript (**Lines: 37-41**).

2. Line 42: "rates" is inappropriate when time-to-event analysis were used to calculate survival probabilities at some time points. If my understanding is correct, they should be "overall survival and disease-free survival probabilities at 18-month were..."

Response: Thank you for your constructive comment. We have replaced "rates"

with “probabilities” in the revised manuscript (Line 44).

Outcomes:

1. *Lines 155: One (MPR) or two primary endpoints (MPR and safety)?*

Response: Thank you for your careful review. There are two primary endpoints, including MPR and safety. We have replaced “was” with “were” in Line 471 of the revised manuscript.

2. *Lines 158-160: the definitions of DFS and OS are inaccurate. How about the censored patients?*

Response: Thanks for your comment. For DFS, patients without disease recurrence or death were censored on their last disease evaluation date. For OS, patients who had no death were censored on their last known survival date. These have been added to the manuscript (Lines: 473-477).

3. *Lines 156-157: pCR rate is before ORR. However, ORR was reported before pCR in the Efficacy sections, e.g., Lines 270.*

Response: Thank you for careful comment. We have revised the manuscript to ensure that “MPR/pCR” is consistently reported before ORR throughout the manuscript, including the sections you mentioned.

4. *Line 234: Is Simon two-stage optimal and MiniMax design?*

Response: Thank you for comment. Simon two-stage optimal design was used in this phase II trial. This has been added to the revised manuscript (Lines: 569).

5. *Line 240: 10% drop-out was used. The references to justify this choice should be added.*

Response: Thanks for your suggestion. We have added the references for the choice of 10% drop-out in the revised manuscript (Lines: 576).

6. Line 240: Suggest adding a sentence-“The total sample size is up to 28 patients in this two-stage Phase 2 study”.

Response: Thanks for your constructive advice. We have added the sentence, “The total sample size was up to 28 patients in this two-stage phase 2 study” in **Lines 575-576** in the revised manuscript.

7. Line 241: “If more than 11 patients achieved MPR,” should be among 28 patients eligible for the MPR evaluation.

Response: Thanks for your careful comment. We have revised the content to: “If more than 11 patients achieved MPR among 28 patients eligible for the MPR evaluation...” in **Lines 577-578** in the revised manuscript.

8. Line 242: Same comment as 1.

Response: Thanks for your comment. To avoid confusion, we have deleted “the primary endpoint of MPR rate was met” in the revised manuscript (**Line 578**).

9. Line 247: ORR is a proportion and “was described as frequency and percentage” does not make sense.

Response: Thanks for your comment. We have revised it to “The 95% CIs were calculated by Clopper-Pearson method for MPR rate, pCR rate, and ORR, and were calculated by Greenwood method for survival.” in the revised manuscript (**Lines: 582-584**).

10. Lines 250, 283-284: “rates” is inappropriate when time-to-event analysis is used.

Response: Thanks for your comment. We have used “probability” or “probabilities” to replace “rate” or “rates”, respectively, in **Lines 120, 127 and 586** in the revised manuscript.

11. Lines 260-261: 28 patients were proposed in Simon two-stage design. However, 29 patients were included in the analysis. Even if more patients were preferred in a

statistical analysis, “If more than 11 patients achieved MPR” as a criterion cannot be used in this Simon two-stage design with 29 patients since ≥ 11 are for 28 patients rather than 29 patients

Response: Thanks for your comment. In Simon two-stage design, 28 patients were proposed, considering a drop-out rate of 10%, a total of 31 patients were needed in the study. However, after enrollment, 2 patients refused surgery after NAIC and were not eligible for MPR evaluation. Therefore, 29 patients were included in the statistical analysis. And there were 20 patients achieved MPR after NAIC. Among the anterior 28 patients eligible for the MPR evaluation, 19 patients achieved MPR (**Supplementary Table 1**), which also indicates that the NAIC regimen was worth for further exploration.

Table 1:

1. Any missing value for age? Only mean and std were shown in the first row.

Response: Thanks for your comment. Age analysis was available for all patients. We have added n (%) for age (<65 or ≥ 65) in the revised **Table 1**.

Characteristics	All patients n = 31
Age, years	
Mean \pm SD (range)	54 \pm 11 (30-75)
<65	26 (83.87)
≥ 65	5 (16.13)